# The Unseen Threat: Residual Knowledge in Machine Unlearning under Perturbed Samples

**Hsiang Hsu**[1],[*] **Pradeep Niroula**[1]**, Zichang He**[1]
**Ivan Brugere**[2]**, Freddy Lecue**[2]**, and Chun-Fu Chen**[1]
[1]JPMorganChase Global Technology Applied Research
[2]JPMorganChase AI Research

## Abstract

Machine unlearning offers a practical alternative to avoid full model re-training by approximately removing the influence of specific user data. While existing methods certify unlearning via statistical indistinguishability from re-trained models, these guarantees do not naturally extend to model outputs when inputs are adversarially perturbed. In particular, slight perturbations of forget samples may still be correctly recognized by the unlearned model—even when a re-trained model fails to do so—revealing a novel privacy risk: information about the forget samples may persist in their local neighborhood. In this work, we formalize this vulnerability as residual knowledge and show that it is inevitable in high-dimensional settings. To mitigate this risk, we propose a fine-tuning strategy, named RURK, that penalizes the model's ability to re-recognize perturbed forget samples. Experiments on vision benchmarks with deep neural networks demonstrate that residual knowledge is prevalent across existing unlearning methods and that our approach effectively prevents residual knowledge.

## 1 Introduction

The widespread use of user data in training machine learning (ML) models has raised significant privacy concerns, particularly when user data is memorized by models such as deep neural networks, and can later be extracted or reconstructed (Carlini et al., 2023; Li et al., 2024). This memorization violates regulations such as the "Right to be Forgotten" in EU's GDPR (Voigt and Von dem Bussche, 2017), which mandates that, upon a user's removal request, an ML service provider must not only delete the user data from databases but also ensure its removal from the ML models themselves (Shastri et al., 2019). As a result, simply deleting the data is often insufficient; instead, re-training the model from scratch without the specific user data is necessary. However, this re-training process is computationally expensive and impractical in real-time or large-scale settings (Ginart et al., 2019).

To address this challenge, *machine unlearning* has been proposed as a more scalable alternative that *approximately* removes the influence of the specific data (referred to as forget samples) from a pre-trained model, as a substitute for avoiding full re-training (Cao and Yang, 2015; Bourtoule et al., 2021). This approach, known as *approximate unlearning*, is often certified through statistical indistinguishability between the unlearned model and one re-trained without the forget samples (Guo et al., 2020; Chourasia and Shah, 2023).

Theoretical formulations of approximate unlearning (cf. § 2) suggest that an unlearned model should behave similarly to a re-trained model on forget samples, for example, by producing similar predictions or classification accuracy. However, these guarantees typically apply only to the original samples and do not extend to their proximities, especially under imperceptible adversarial perturbations. In complex hypothesis spaces, even minor input changes can cause the unlearned and re-trained models

---

[*]Correspondence to: Hsiang Hsu <`hsiang.hsu@jpmchase.com`>.

39th Conference on Neural Information Processing Systems (NeurIPS 2025).

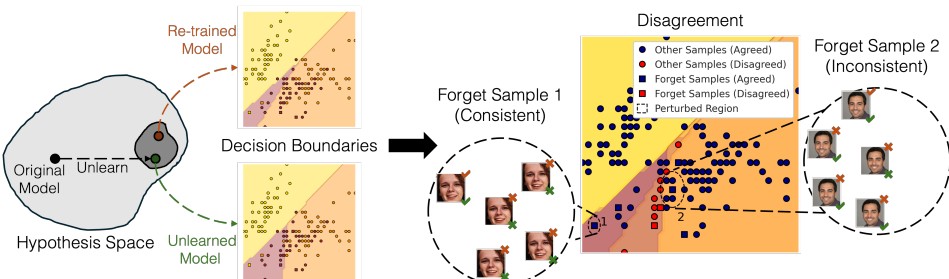

Figure 1: The re-trained (brown) and unlearned (green) models are statistically similar but may have slightly different decision boundaries (**left**), leading to disagreements on forget samples (**right**). Checkmarks and crosses on the images indicate correct and incorrect predictions from the re-trained model (top) and the unlearned model (bottom), respectively. Ideally—as shown with forget sample 1—both models should behave consistently across the original and all perturbed inputs. Residual knowledge in machine unlearning is illustrated by comparing prediction correctness: for forget sample 2, both models agree on the original sample, but the unlearned model correctly predicts more of its perturbed variants. see Appendix B.5 for experimental details.

to *disagree* in their predictions, creating an additional layer of privacy risk. A particularly concerning case is when a slightly perturbed forget sample is still correctly classified by the unlearned model but not by the re-trained one. This reveals the presence of *residual knowledge*—latent traces of the forget samples that still persist in the unlearned model (cf. Figure 1). Residual knowledge is a prevalent issue. For instance, on the CIFAR-10 dataset, when subjected to a small perturbation norm ($\approx 0.03$), over 7% of the forget samples still exhibit residual knowledge (cf. Appendix C.4).

In this paper, we study how adversarially perturbed inputs affect the indistinguishability between unlearned and re-trained models in the classification setting. In § 3, we show that adversarial examples can reliably distinguish between the two models, even under certified approximate unlearning. We formalize this observation by demonstrating that adversarial attacks can induce probabilistically distinguishable outputs. Further, using geometric probability (Talagrand, 1995), we prove that such disagreement is inevitable in high-dimensional input spaces. These findings underscore the need to explicitly incorporate robustness against such local disagreement into unlearning frameworks.

To capture this risk, § 4 introduces the notion of residual knowledge, which quantifies the likelihood that an unlearned model retains predictive traces of forget samples under perturbation. As a more tractable proxy for disagreement, residual knowledge exposes a vulnerability not addressed by current certification methods. To mitigate this, we propose RURK, a fine-tuning strategy for Robust Unlearning that suppresses Residual Knowledge while maintaining accuracy on the rest of the samples. Our approach identifies and penalizes perturbed inputs that the unlearned model still classifies correctly, thus enhancing robustness against residual knowledge. We empirically validate the existence of such local disagreement and residual knowledge across multiple unlearning algorithms in § 5. We further demonstrate that our fine-tuning strategy effectively reduces residual knowledge on standard vision datasets using deep neural networks. To the best of our knowledge, this is the first work to uncover this novel privacy risk and provide a scalable solution to address it. We conclude in § 6 with a discussion of limitations and future research directions.

Omitted proofs, additional explanations and discussions, details on experiment setups and training, and additional experiments are included in Appendices A, B and C, respectively.

## 2 Background and related work

Let $\mathcal{S} \triangleq \{\mathbf{s}_i = (\mathbf{x}_i, y_i)\}_{i=1}^n$ be a training dataset with $\mathbf{x}_i \in \mathcal{X} \subseteq \mathbb{R}^d$ and $y_i \in \mathcal{Y}$. The hypothesis space $\mathcal{H} \triangleq \{h_{\mathbf{w}} : \mathcal{X} \to \mathcal{Y}; \mathbf{w} \in \mathcal{W}\}$ consists of models parameterized by $\mathbf{w}$. We use $h$ and $\mathbf{w}$ interchangeably to refer to a model in $\mathcal{H}$. Let $\ell : \mathcal{W} \times \mathcal{S} \to \mathbf{R}^+$ be the loss function, and define the empirical risk as $L(\mathbf{w}, \mathcal{S}) = \frac{1}{n} \sum_{i=1}^n \ell(\mathbf{w}, \mathbf{s}_i)$. A (randomized) learning algorithm $A : \mathcal{S} \to \mathcal{H}$, such as Stochastic Gradient Descent (SGD), returns a model minimizing $L(\mathbf{w}, \mathcal{S})$ and induces a distribution over $\mathcal{H}$. We denote the $\ell_p$-norm by $\|\mathbf{z}\|_p$, and the $\ell_p$ ball of radius $\tau$ centered at $\mathbf{x}$ by $\mathcal{B}_p(\mathbf{x}, \tau) \triangleq \{\mathbf{z} \in \mathcal{X} : \|\mathbf{z} - \mathbf{x}\|_p \leq \tau\}$. Also, let $\mathbb{1}(\cdot)$ be the indicator function, $\mathrm{surf}(\mathcal{Z})$ the surface area of a set $\mathcal{Z}$ in $\mathbb{R}^d$, and $\mathcal{O}(\cdot)$ the big-O notation.

## 2.1 Machine unlearning

The forget set $\mathcal{S}_f \subseteq \mathcal{S}$ contains training samples to be removed. Machine unlearning is modeled as a randomized mechanism $M(A(\mathcal{S}), \mathcal{S}, \mathcal{S}_f)$ that removes the influence of $\mathcal{S}_f$ from a pre-trained model $A(\mathcal{S})$ (Xu et al., 2023). Like the learning algorithm itself, the output of the unlearning mechanism can be regarded as a random variable. A simple approach adds Gaussian noise to the model weights: $M(\mathbf{w}, \mathcal{S}, \mathcal{S}_f) = \mathbf{w} + \sigma\mathbf{n}$, with $\mathbf{n} \sim \mathcal{N}(0, \mathbf{I}_{|\mathbf{w}|})$ (Golatkar et al., 2020a). However, as $\sigma$ increases, the model becomes increasingly independent of the training data, potentially degrading performance. To preserve utility, unlearning should retain accuracy on the retain set $\mathcal{S}_r \triangleq \mathcal{S} \setminus \mathcal{S}_f$. A naïve yet effective strategy is to re-train the model from scratch on the retain set, i.e., $M(A(\mathcal{S}), \mathcal{S}, \mathcal{S}_f) = A(\mathcal{S}_r)$. While this achieves *exact unlearning*, it is often impractical due to the high computational cost of re-training for each unlearning request. Therefore, the core objective of machine unlearning is to efficiently eliminate the influence of $\mathcal{S}_f$ while preserving utility on $\mathcal{S}_r$, all without full re-training.

This objective has spurred a vast literature on machine unlearning. Here, we outline the broad trends and defer details of specific algorithms to § 5. For comprehensive surveys, see Nguyen et al. (2022); Xu et al. (2023); Wang et al. (2024). Initial efforts focused on image classifiers, aiming to remove the influence of specific training images from a pre-trained model (Ginart et al., 2019; Wu et al., 2020; Neel et al., 2021; Sekhari et al., 2021; Izzo et al., 2021; Fan et al., 2023; Kurmanji et al., 2024; Goel et al., 2022; Zhang et al., 2024; Kodge et al., 2024). This idea was later extended to erasing abstract concepts (Ravfogel et al., 2022; Belrose et al., 2023), as well as adapting unlearning to broader model classes, including image generators (Li et al., 2024; Gandikota et al., 2023) and large language models (Eldan and Russinovich, 2023; Yao et al., 2024; Liu et al., 2025; Jang et al., 2022; Wang et al., 2023). Another line of work develops unlearning methods tailored to specific model families—such as linear classifiers (Guo et al., 2020), kernel methods (Golatkar et al., 2020b; Zhang and Zhang, 2022), and tree-based models (Brophy and Lowd, 2021; Schelter et al., 2021).

## 2.2 Certification of machine unlearning

A valid unlearning mechanism ensures that the unlearned model $M(A(\mathcal{S}), \mathcal{S}, \mathcal{S}_f)$ is statistically similar to a re-trained model $A(\mathcal{S}_r)$. We denote the unlearned and re-trained models as random variables $M$ and $A$, with corresponding distributions $P_M$ and $P_A$, respectively. This requirement is formalized via $(\epsilon, \delta)$-indistinguishability:

**Definition 1** $((\epsilon, \delta)$-indistinguishability). *Let $X$ and $Y$ be two random variables over a domain $\Omega$. $X$ and $Y$ are said to be $(\epsilon, \delta)$-indistinguishable, also denoted as $X \overset{\epsilon,\delta}{\approx} Y$, if for all $\mathcal{T} \subseteq \Omega$*

$$e^{-\epsilon}\left(\Pr[Y \in \mathcal{T}] - \delta\right) \leq \Pr[X \in \mathcal{T}] \leq e^{\epsilon}\Pr[Y \in \mathcal{T}] + \delta. \tag{1}$$

This notion underlies differential privacy (DP) (Dwork and Roth, 2014) when $X$ and $Y$ are outputs on neighboring datasets. It also quantifies reproducibility of empirical findings when applied to models trained on independent samples from the same data distribution (Kalavasis et al., 2023; Impagliazzo et al., 2022; Bun et al., 2023).

Guo et al. (2020) were among the first to introduce certified machine unlearning using $(\epsilon, \delta)$-indistinguishability. An unlearning algorithm $M$ satisfies $(\epsilon, \delta)$-unlearning if $M(A(\mathcal{S}), \mathcal{S}, \mathcal{S}_f) \overset{\epsilon,\delta}{\approx} A(\mathcal{S}_r)$; this reduces to exact unlearning when $\epsilon = \delta = 0$. Indistinguishability can be measured through various probability divergences as well. For instance, Chourasia and Shah (2023) and Chien et al. (2024) proposed $(\alpha, \epsilon)$-Rényi unlearning, which holds when the $\alpha$-Rényi divergence[2] between $P_M$ and $P_A$ is bounded by $\epsilon$, and can be translated into $(\epsilon, \delta)$-unlearning. Indeed, $(\alpha, \epsilon)$-Rényi unlearning can be converted to $(\epsilon + \log(1/\delta)/(\alpha - 1), \delta)$-unlearning, for any $0 < \delta < 1$ (Mironov, 2017). Both frameworks assume uniqueness or a unique stationary distribution of the empirical minimizer, which may limit their applicability in complex model classes.

Instead of comparing full model distributions $P_M$ and $P_A$, a more practical certification framework evaluates unlearned models with readout functions $f : \mathcal{H} \times \mathcal{S} \to \mathbb{R}$ that an adversary might use to distinguish unlearned from re-trained models. Indistinguishability is then assessed via the output distribution of $f$, using divergences such as the Kullback-Leibler (KL) divergence[3]. The certification

---

[2]For $\alpha > 1$, the $\alpha$-Rényi divergence (Rényi, 1961) is defined as $D_\alpha(P|Q) \triangleq \frac{1}{\alpha-1}\log\mathbb{E}_Q\left[\left(\frac{P}{Q}\right)^\alpha\right]$.

[3]The KL divergence (Kullback and Leibler, 1951) is defined as $D_{\mathsf{KL}}(P|Q) \triangleq \mathbb{E}_P[\log(P/Q)]$.

condition requires $D_{\mathsf{KL}}(\Pr[f(M, \mathcal{T})] \| \Pr[f(A, \mathcal{T})] \leq \epsilon$ for any subset $\mathcal{T} \subseteq \mathcal{S}$, such as the forget, retain, or even hold-out test sets (Nguyen et al., 2020; Golatkar et al., 2020a). This formulation flexibly captures different behaviors: $f$ could represent a binary classifier (e.g., for Membership Inference Attack (MIA)) (Fan et al., 2023), utility metrics like accuracy, or re-learning time—the training epochs needed to re-learn $\mathcal{S}_f$ (Golatkar et al., 2020a). Focusing on readout functions offers a more tractable and empirically grounded certification approach, especially for complex models.

## 3   Model indistinguishability and disagreement over sample perturbation

Statistical indistinguishability between unlearned and re-trained models can be certified theoretically—via $(\epsilon, \delta)$- or $(\alpha, \epsilon)$-Rényi unlearning—or empirically using a readout function (cf. §2). This, indeed, ensures similarity in model weights or outputs on forget/retain samples. However, such guarantees do not readily extend to perturbed inputs, even with imperceptible adversarial perturbations.

Several studies have investigated the vulnerability of unlearning algorithms to adversarial manipulation. Marchant et al. (2022) show that adversarial inputs can significantly increase the computational cost of unlearning[4], while Pawelczyk et al. (2025) find that poisoning attacks can obstruct complete forgetting (cf. Appendix B.1). Zhao et al. (2024) further demonstrate that even a small number of malicious unlearning requests can weaken the adversarial robustness of the resulting model. More recently, Xuan and Li (2025) propose an attack that manipulates an unlearned model such that its outputs on forget samples resemble those of the original model (see Proposition 3 and Definition 4 in their paper). However, these works primarily study how adversarial perturbations affect the unlearned model itself, whereas our goal is to evaluate the distinguishability between the unlearned and re-trained models.

This paper intends to explore a new dimension of how adversarial examples affect unlearning—specifically, how such examples may behave differently when fed to an unlearned model versus a re-trained one, even when the two satisfy statistical indistinguishability. Remarkably, despite this indistinguishability, it remains possible to craft adversarial inputs that distinguish between the two, revealing a novel privacy risk. This phenomenon, related to the transferability of adversarial examples (Tramèr et al., 2017), remains largely unexplored in the unlearning literature. The proofs of the propositions in this section are included in Appendix A.

### 3.1   Distinguishability of model output with adversarial examples

We begin by considering adversarial examples generated against either the unlearned or the re-trained model. The process of finding an adversarial example for a given input $\mathbf{x} \in \mathcal{X}$ can be formalized as a read-out function $g_{\mathbf{x}} : \mathcal{H} \to \mathcal{X}$, which may be deterministic or randomized. For instance, the minimum $\ell_2$-norm perturbation for a binary linear classifier, given by $g_{\mathbf{x}}(\mathbf{w}) = \mathbf{x} - (\mathbf{x}^\top \mathbf{x})\mathbf{w}/\|\mathbf{w}\|_2^2$, and the Fast Gradient Sign Method (FGSM) (Goodfellow et al., 2016) are deterministic. In contrast, methods such as Projected Gradient Descent (PGD) (Madry et al., 2018) or random perturbations within $\mathcal{B}_p(\mathbf{x}, \tau)$ are randomized. Since generating adversarial examples can be seen as a post-processing of the model, the indistinguishability of adversarial examples can, in principle, be derived from the indistinguishability of the models themselves. Indeed, $(\epsilon, \delta)$-indistinguishability is preserved under arbitrary post-processing: if two random variables $X$ and $Y$ are $(\epsilon, \delta)$-indistinguishable, then so are $f(X)$ and $f(Y)$ for any (deterministic or randomized) function $f$. However, when considering adversarial examples of specific model realizations drawn from either the unlearning or re-training processes (cf. Lemma A.1), the level of indistinguishability can degrade, as formalized in the following proposition.

**Proposition 1** (Adversarial example on a model is less indistinguishable). *Suppose the unlearned $M(A(\mathcal{S}), \mathcal{S}, \mathcal{S}_f)$ and re-trained $A(\mathcal{S}_r)$ models are $(\epsilon, \delta)$-indistinguishable, and let $\mathbf{x}$ be a fixed sample. Then with probability $2\delta/(1 - e^{-\epsilon})$, the adversarial example $g_{\mathbf{x}}(\cdot)$ found against the models $m \sim M$ or $a \sim A$ satisfies, for all $\mathcal{X}' \subseteq \mathcal{X}$*

$$\Pr[g_{\mathbf{x}}(m) \in \mathcal{X}'] \leq e^{-2\epsilon} \Pr[g_{\mathbf{x}}(a) \in \mathcal{X}'] \text{ or } e^{2\epsilon} \Pr[g_{\mathbf{x}}(a) \in \mathcal{X}'] \leq \Pr[g_{\mathbf{x}}(m) \in \mathcal{X}']. \quad (2)$$

Proposition 1 shows that even if the unlearned and re-trained models satisfy approximate unlearning certified by $(\epsilon, \delta)$-indistinguishability, the adversarial examples $g_{\mathbf{x}}(h)$ can become less indistinguishable. Specifically, given a model $h$, with probability $2\delta/(1 - e^{-\epsilon})$, the distinguishability of the

---

[4]Allouah et al. (2025, Theorem 3) report similar behavior when unlearning out-of-distribution samples.

adversarial examples increases by a factor of two. Moreover, the indistinguishability assumption in Proposition 1 can be readily generalized to $(\alpha, \epsilon)$-Rényi unlearning.

## 3.2 Disagreement on adversarial examples is inevitable

In the previous section, Proposition 1 showed that even when the unlearned and re-trained models satisfy $(\epsilon, \delta)$-indistinguishability, there remains a nonzero probability that their likelihood ratio can still be exploited to distinguish them via adversarial examples. However, evaluating the bound in Eq. (2) requires computing probabilities over the entire hypothesis space $\mathcal{H}$, which is often computationally intractable when $\mathcal{H}$ is large or complex.

To address this challenge, we instead focus on model outputs at individual samples $\mathbf{x}$, which may belong to the forget or retain sets. We introduce a more tractable binary metric called *disagreement*, defined as $k(\mathbf{x}) = \mathbb{1}(M(\mathbf{x}) \neq A(\mathbf{x}))$, where $k(\mathbf{x}) = 1$ indicates that the unlearned and re-trained models produce different predictions at $\mathbf{x}$, and $k(\mathbf{x}) = 0$ otherwise. Disagreement has been widely used in the machine learning literature to study model behavior (Krishna et al., 2025; Uma et al., 2021), as well as prediction stability and bias (Kulynych et al., 2023). Notably, the $(\epsilon, \delta)$-indistinguishability guarantee over distributions can be translated into an upper bound on empirical disagreement across samples. In particular, when $\epsilon = \delta = 0$, perfect indistinguishability (i.e., exact unlearning) implies $k(\mathbf{x}) = 0$ for all $\mathbf{x} \in \mathcal{S}$. Ideally, we seek agreement not only on the training set but across the entire sample space $\mathcal{X}$, including unseen or perturbed data. Yet even small nonzero values of $\epsilon$ and $\delta$ can result in disagreement on certain inputs, particularly under adversarial perturbations, as we demonstrate in the following analysis.

To formalize this, we consider the sample space $\mathcal{X} = \mathbb{S}^{d-1} = \{\mathbf{x} \in \mathbb{R}^d \mid \|\mathbf{x}\|_2 = 1\}$, corresponding to the unit sphere in $\mathbb{R}^d$, where all data points are normalized to unit norm. In this setting, the disagreement function $k(\mathbf{x})$ can be viewed as a binary classifier over $\mathbb{S}^{d-1}$. We aim to bound the probability that disagreement occurs not only at a sample $\mathbf{x}$, but also within its local neighborhood under bounded perturbations, i.e., for $\mathbf{x}' \in \mathcal{B}_p(\mathbf{x}, \tau)$ such that $\|\mathbf{x} - \mathbf{x}'\|_p \leq \tau$. This motivates the need to understand how disagreement can propagate beyond the observed dataset to its local neighborhood in the broader sample space $\mathcal{X}$. A key mathematical tool for this purpose is the isoperimetric inequality (cf. Lemma A.2 and Talagrand (1995)). In probability theory and geometry, the isoperimetric inequality provides a lower bound on how the measure of a set expands when it is slightly "extended." In high-dimensional spaces, particularly under uniform or Gaussian distributions, it formalizes the intuition that if a subset occupies a small volume, its boundary must be relatively large. In our context, this implies that even if two models disagree on only a small region, small perturbations can still cause disagreement to spread over a much larger region in the sample space. Combining this geometric insight with the definition of $(\epsilon, \delta)$-unlearning (cf. Definition 1), we establish the following result:

**Proposition 2** (Inevitable disagreement). *Consider a sample $\mathbf{x} \in \mathbb{S}^{d-1}$. Let $M$ and $A$ denote the unlearned and re-trained models, respectively, satisfying $(\epsilon, \delta)$-unlearning. Suppose[5] the agreement region satisfies $\mathsf{surf}(\{\mathbf{x} \in \mathbb{S}^{d-1} | k(\mathbf{x}) = 0\})/\mathsf{surf}(\mathbb{S}^{d-1}) \leq 1/2$. Then with probability at least*

$$\left(\frac{2\delta}{1 - e^{-\epsilon}}\right) \left\{1 - \mathcal{O}\left[\left(\frac{\pi}{8}\right)^{1/2} \exp\left(-2\epsilon - \frac{d-1}{2}\tau^2\right)\right]\right\}, \tag{3}$$

*either $k(\mathbf{x}) = 1$, or there exist $\mathbf{x}' \in \mathcal{B}_p(\mathbf{x}, \tau)$ such that $k(\mathbf{x}') = 1$, where $p = 2$ or $p = \infty$.*

When $\epsilon = \delta = 0$, the probability in Eq. (3) is zero[6], since the unlearned and re-trained models are identical and cannot disagree. Conversely, as $\epsilon \to \infty$ for a fixed $\delta$—reflecting maximal distinguishability between $M$ and $A$—the probability of disagreement approaches its upper bound of $2\delta$. More generally, for any fixed $(\epsilon, \delta)$, the probability in Eq. (3) depends solely on the perturbation norm $\tau$: as $\tau$ increases, the probability of disagreement rises, as adversarial examples explore a broader neighborhood around the input. Moreover, Proposition 2 holds for both $p = 2$ and $p = \infty$, aligning with the common practice of measuring perturbation size using either the $\ell_2$ (Euclidean) norm or the $\ell_\infty$ (maximum) norm in prior works, e.g., Goodfellow et al. (2014); Madry et al. (2018).

While the assumption that samples lie on a unit sphere is admittedly strong, the core principle of the isoperimetric inequality—that small-volume sets necessarily have large boundaries—extends to

---

[5]This condition can be easily satisfied in multi-class settings, i.e., $|\mathcal{Y}| > 2$.

[6]Let $\delta = k\epsilon$ and by L'Hopital's rule $\lim_{\epsilon \to 0} \frac{2k\epsilon}{1 - e^{-\epsilon}} = \lim_{\epsilon \to 0} \frac{2k}{e^{-\epsilon}} = 0$.

more realistic domains. In particular, Ledoux (2001, Proposition 2.8) generalizes this property to the unit cube, a more suitable setting for image data where pixel values are normalized. Nonetheless, Proposition 2 remains insightful, as it demonstrates that disagreement on perturbed samples can arise even under the stricter unit sphere assumption, thereby strengthening its relevance under looser conditions like the unit cube.

## 4 Removing residual knowledge in unlearned models

Proposition 2 demonstrates that disagreement between the unlearned and re-trained models can persist even when the unlearning algorithm satisfies the $(\epsilon, \delta)$-unlearning constraint. Although the two models may agree on the original input $(\mathbf{x}, y)$, it is often possible to craft adversarial examples with imperceptible perturbations (small $\tau$) that cause them to diverge, revealing a potential privacy risk. This result arises from two sources of randomness: (1) that of the unlearning algorithm itself (e.g., model initialization) and (2) that of perturbations applied to forget samples. While the proposition establishes that such disagreement is inevitable, demonstrating existence alone is insufficient—quantifying its extent in practice is equally essential.

To that end, we fix specific realizations of the unlearned and re-trained models and define as *adversarial disagreement*[7] as $k_\tau(\mathbf{x}') = \mathbb{1}(m(\mathbf{x}') \neq a(\mathbf{x}'))$ for $\mathbf{x}' \sim \mathcal{B}_p(\mathbf{x}, \tau)$. The expected value of this indicator gives the probability of disagreement under perturbation: $\mathbb{E}[k_\tau(\mathbf{x}')] = \Pr[m(\mathbf{x}') \neq a(\mathbf{x}')]$. Adversarial disagreement is challenging to control, especially in multi-class settings, as it involves evaluating probabilities over all possible output combinations of $m(\mathbf{x}')$ and $a(\mathbf{x}')$. To address this complexity, we next introduce a special case of adversarial disagreement—called *residual knowledge*—which offers a more tractable measure with operational meanings.

### 4.1 Connecting disagreement with residual knowledge

We focus on a particularly concerning form of adversarial disagreement, i.e., $\mathbb{1}(m(\mathbf{x}') = y, a(\mathbf{x}') \neq y)$ for a forget sample[8] $(\mathbf{x}, y) \in \mathcal{S}_f$. It implies that a forget sample, intended to be fully erased from the model, can be slightly perturbed such that the unlearned model still correctly classifies it, while the re-trained model does not. This discrepancy suggests that traces of the forget samples may remain embedded in the unlearned model's decision boundary, even when formal indistinguishability guarantees are satisfied. This scenario motivates our definition of *residual knowledge*—the persistence of information about the forget set in the predictive behavior of the unlearned model. The presence of residual knowledge reveals a subtle yet significant vulnerability: it demonstrates how adversarial examples can compromise the effectiveness of unlearning, and underscores the need for stronger robustness guarantees that go beyond conventional $(\epsilon, \delta)$-unlearning.

Consider a forget sample $(\mathbf{x}, y) \in \mathcal{S}_f$, and let $m \sim M(A(\mathcal{S}), \mathcal{S}, \mathcal{S}_f)$ and $a \sim A(\mathcal{S}_r)$ be independently sampled unlearned and re-trained models, respectively. We formally define the residual knowledge around $\mathbf{x}$ as the following non-negative ratio:

$$r_\tau((\mathbf{x}, y)) \triangleq \frac{\Pr[m(\mathbf{x}') = y]}{\Pr[a(\mathbf{x}') = y]}, \quad \mathbf{x}' \overset{i.i.d.}{\sim} \mathcal{B}_p(\mathbf{x}, \tau). \tag{4}$$

This definition can be naturally extended to the entire forget set $\mathcal{S}_f$ by averaging over all forget samples, $r_\tau(\mathcal{S}_f) \triangleq \frac{1}{|\mathcal{S}_f|} \sum_{(\mathbf{x}, y) \in \mathcal{S}_f} r_\tau((\mathbf{x}, y))$. Among the possible cases, $r_\tau(\mathcal{S}_f) > 1$ is especially concerning, as it indicates that the unlearned model $m$ is more likely than the re-trained model $a$ to correctly classify perturbed variants of forget samples. This suggests the presence of residual knowledge, which is the main privacy risk we aim to address in this paper. In this sense, the case where $r_\tau(\mathcal{S}_f) < 1$ is more tolerable, as it implies the unlearned model has lost the ability to recognize the forget samples. The case where $r_\tau(\mathcal{S}_f) = 1$ can only occur when $m = a$, meaning the unlearned model $m$ achieves exact unlearning. In practice, $r_\tau((\mathbf{x}, y))$ can be estimated via Monte

---

[7]The randomness in $k(\mathbf{x})$ arises from the stochasticity of the models, whereas that in $k_\tau(\mathbf{x}')$ arises from sampling perturbed inputs $\mathbf{x}' \sim \mathcal{B}_p(\mathbf{x}, \tau)$. Defining disagreement over both sources of randomness would require characterizing the distributions of $M$ and $A$, an intractable task without strong assumptions (Guo et al., 2020; Chourasia and Shah, 2023; Chien et al., 2024).

[8]Access to forget samples is standard in literature (Appendix B.3) and necessary in classification unlearning, as users must provide or allow retrieval of the data for deletion.

Carlo sampling. Specifically, by drawing $c$ i.i.d. samples $\{\mathbf{x}_i'\}_{i=1}^c$ from $\mathcal{B}_p(\mathbf{x}, \tau)$, we approximate the ratio as $\hat{r}_\tau((\mathbf{x}, y)) = \frac{\sum_{i=1}^c \mathbb{1}(m(\mathbf{x}_i')=y)}{\sum_{i=1}^c \mathbb{1}(a(\mathbf{x}_i')=y)}$.

The notion of residual knowledge is closely tied to adversarial disagreement. In particular, residual knowledge offers both upper and lower bounds on the expected adversarial disagreement between the unlearned and re-trained models (cf. Lemma A.4):

$$r_\tau((\mathbf{x}, y)) \Pr\left[a(\mathbf{x}') = y\right]\left(1 - \Pr\left[a(\mathbf{x}') = y\right]\right) \leq \mathbb{E}\left[k_\tau(\mathbf{x}')\right] \leq 1 - r_\tau((\mathbf{x}, y)) \Pr\left[a(\mathbf{x}') = y\right]^2. \quad (5)$$

As $r_\tau((\mathbf{x}, y)) \to 1$, the expected adversarial disagreement $\mathbb{E}\left[k_\tau(\mathbf{x}')\right]$ approaches zero. In this sense, residual knowledge serves as a practical proxy for estimating the distinguishability between the unlearned and re-trained models, and provides a more tractable means to quantify and control adversarial disagreement.

## 4.2  `RURK`: Robust unlearning against residual knowledge

To address residual knowledge, we propose a fine-tuning objective that simultaneously enforces unlearning and regulates residual knowledge. Ideally, residual knowledge should be close to 1; however, accurately computing $r_\tau(\mathcal{S}_f)$ requires access to a re-trained model $a \sim A(\mathcal{S}_r)$, which is typically unavailable during unlearning. Thus, our objective is to attenuate residual knowledge and ensure that $r_\tau(\mathcal{S}_f) \leq 1$. Recall that the numerator of the residual knowledge in Eq. (4) is $\Pr[m(\mathbf{x}') = y]$; thus, directly minimizing this probability alone can effectively suppress residual knowledge, regardless of the denominator $\Pr[a(\mathbf{x}') = y]$. Based on this insight, we define the set of vulnerable perturbations as $\mathcal{V}((\mathbf{x}, y), \tau) \triangleq \{\mathbf{x}' \in \mathcal{B}_p(\mathbf{x}, \tau) | m(\mathbf{x}') = y\}$, which captures perturbed samples that continue to be associated with the true label by the unlearned model—indicating residual knowledge. In practice, we construct $v$ such samples by adapting adversarial attack methods, such as FGSM or PGD, to solve the constrained minimization problem $\min_{\mathbf{z} \in \mathcal{B}_p(\mathbf{x}, \tau)} \ell(\mathbf{w}, (\mathbf{z}, y))$.

Given the vulnerable set, we formulate the following fine-tuning objective for `RURK` as:

$$L_{\text{RURK}}(\mathbf{w}, \mathcal{S}) = \underbrace{\frac{1}{|\mathcal{S}_r|} \sum_{(\mathbf{x}, y) \in \mathcal{S}_r} \ell(\mathbf{w}, (\mathbf{x}, y))}_{\text{Term (i)}} - \lambda \underbrace{\frac{1}{|\mathcal{S}_f|} \sum_{(\mathbf{x}, y) \in \mathcal{S}_f} \kappa(\mathbf{w}, (\mathbf{x}, y))}_{\text{Term (ii)}}, \quad (6)$$

where $\kappa(\mathbf{w}, (\mathbf{x}, y)) = \frac{1}{v} \sum_{\{\mathbf{x}'\}_{i=1}^v \in \mathcal{V}((\mathbf{x}, y), \tau)} \ell(\mathbf{w}, (\mathbf{x}', y))$, and $\lambda \geq 0$ is the regularization strength.

Term (i) preserves performance on the retain set, following prior work such as Neel et al. (2021); Kurmanji et al. (2024); Chien et al. (2024). Term (ii) serves as a regularization term that penalizes residual knowledge by discouraging the unlearned model from re-identifying vulnerable perturbations, thereby improving robustness with respect to residual knowledge. Explicitly searching for vulnerable perturbations in $\mathcal{V}((\mathbf{x}, y), \tau)$ during each gradient step can be computationally expensive. To mitigate this overhead, we may adopt a more efficient approximation by setting $\mathcal{V}((\mathbf{x}, y), \tau) = \mathcal{B}_p(\mathbf{x}, \tau)$, i.e., removing all label information in the neighborhood of each forget sample. We outline the `RURK` algorithm in Appendix B.2.

Note that as $\tau \to 0$, the objective in Eq. (6) reduces to that used in prior works such as Neel et al. (2021); Chien et al. (2024). When optimized via (Projected) Noisy Gradient Descent (NGD) (Chourasia et al., 2021), the resulting unlearned model satisfies $(\alpha, \epsilon)$-Rényi unlearning, where $\epsilon$ depends on the smoothness and Lipschitz continuity of the loss function $L(\mathbf{w}, \mathcal{S})$. Specifically, less smooth or less Lipschitz losses lead to larger $\epsilon$, making the unlearned model more distinguishable from the re-trained one (Chien et al., 2024, Theorem 3.2). This reveals a fundamental trade-off between model indistinguishability and robustness against residual knowledge: increasing $\tau$ in term (ii) of Eq. (6) enlarges the adversarial perturbation space, which makes the adversarial loss $\kappa(\mathbf{w}, (\mathbf{x}, y))$ less smooth and with a larger Lipschitz constant—ultimately increasing $\epsilon$.

## 5  Empirical Studies

We now evaluate residual knowledge of state-of-the-art unlearning algorithms and its mitigation via `RURK` on three vision benchmarks. We denote `Original` as the model $A(\mathcal{S})$ trained on the full dataset $\mathcal{S}$, and `Re-train` as the ideal (but not viable in real world) model $A(\mathcal{S}_r)$ re-trained from scratch without the forget set $\mathcal{S}_f$, used here as a reference. For experimental details, including dataset settings and hyper-parameter choices, refer to Appendix B.4; additional results are provided in Appendix C.

**Data and unlearning settings.** We evaluate unlearning methods on image classification under three random *sample unlearning* scenarios. The first scenario, small CIFAR-5, follows Golatkar et al. (2020a,b) and uses a reduced CIFAR-10 subset (Krizhevsky et al., 2009) containing 200 training and 200 test images per class from the first five classes, while the second scenario uses the full CIFAR-10 dataset; both employ ResNet-18 (He et al., 2016). The third scenario is based on a larger-scale ImageNet-100, a 100-class subset of ImageNet-1k (Deng et al., 2009) with 1,300 images per class, trained with ResNet-50. In all cases, only 50% of class 0 is unlearned, in contrast to the *class unlearning* setting (Kodge et al., 2024), which aims to remove all samples from a class. To avoid any external knowledge, both `Original` and `Re-train` models are trained from scratch without using any pre-trained weights. See Appendix C.2 for class-unlearning results and Appendix C.3 for ablations on `RURK` hyper-parameters and architectures (e.g., VGG (Simonyan and Zisserman, 2014)).

**Baseline algorithms.** We evaluate three machine unlearning algorithms suited for small CIFAR-5 settings {`CR`, `Fisher`, `NTK`}, and eight methods for the full CIFAR-10 {`GD`, `NGD`, `GA`, `NegGrad+`, `EU-k`, `CF-k`, `SCRUB`, `SSD`}. Certified Removal (`CR`) uses influence functions and a one-step Newton update to estimate and remove the effect of forget samples (Guo et al., 2020). This approach was extended by Golatkar et al. (2020a), who incorporate the Fisher information matrix (`Fisher`) computed on the retain set, and by Golatkar et al. (2020b), who apply Neural Tangent Kernel (`NTK`) linearization of neural networks. These pioneering methods, while foundational, are not scalable to large models due to the computational cost of Hessian-based operations. Gradient Descent (`GD`) fine-tunes `Original` on the retain set $\mathcal{S}_r$ using standard SGD (Neel et al., 2021). Noisy Gradient Descent (`NGD`) simply modifies `GD` by adding Gaussian noise in the gradient update steps for better privacy guarantees (Chourasia and Shah, 2023; Chien et al., 2024). On the other hand, Gradient Ascent (`GA`) removes the influence of $\mathcal{S}_f$ by reversing gradient updates[9] on the forget set (Graves et al., 2021; Jang et al., 2022). `NegGrad+` combines both strategies by applying `GD` to $\mathcal{S}_r$ and `GA` to $\mathcal{S}_f$ simultaneously (Kurmanji et al., 2024). To improve parameter efficiency, Goel et al. (2022) propose two layer-wise methods: Exact Unlearning the last $k$ layers (`EU-k`), which re-trains them from scratch, and Catastrophically Forgetting the last $k$ layers (`CF-k`), which only fine-tunes them on $\mathcal{S}_r$. `SCRUB` casts unlearning as a teacher-student distillation process, where the student selectively learns retain-set knowledge from the teacher (Kurmanji et al., 2024). The final method, `SSD`, selectively dampens model weights by uses Fisher information to estimate the influence of $\mathcal{S}_f$ (Foster et al., 2024), a scalable version of `Fisher`. Further details on these baselines are in Appendix B.3.

**Evaluation metrics.** We adopt five evaluation metrics, as described in § 2.2, to comprehensively assess both the unlearning efficacy and the utility of the resulting models. To capture different dimensions of unlearning effectiveness, we first report Unlearning Accuracy[10], following Fan et al. (2023). Second, we conduct a MIA using a support vector classifier (SVC) trained to distinguish training samples from test samples based on the model's output likelihoods; we report the SVC's attack failure rate (MIA Accuracy) as a measure of privacy protection. Third, Re-learn Time quantifies how easily a model can re-acquire the forget-set information: it is defined as the number of fine-tuning epochs needed for the model $m$ to satisfy $L(m, \mathcal{S}_f) \leq (1 + \eta)L(\texttt{Original}, \mathcal{S}_f)$, with $\eta = 0.05$. To assess model utility, we report classification accuracy on the retain set and the hold-out test set, referred to as Retain Accuracy and Test Accuracy, capturing both performance preservation and generalization. As discussed in § 2.1, a desirable unlearning method should minimize the deviation of performance from `Re-train` in both sample and class unlearning settings. To this end, we compute the Average Gap (Avg. Gap)—the mean absolute gap from `Re-train` across Retain, Unlearn, Test, and MIA accuracies—where a smaller Avg. Gap indicates better unlearning. Notably, methods such as `CR`, `Fisher`, `NTK`, and `NGD` already offer certified unlearning guarantees under frameworks like $(\epsilon, \delta)$-unlearning, Rényi unlearning, or bounded KL divergence; these guarantees do not necessarily imply a small performance gap on accuracy-based metrics.

The performance of all unlearning methods, including `RURK`, is best understood by jointly examining Table 1 and Figure 2. Table 1 presents standard unlearning metrics to ensure each method maintains reasonable test accuracy without excessive forgetting, while Figure 2 evaluates residual knowledge, measuring a model's ability to resist re-identifying perturbed forget samples. Ideally, a properly unlearned model should match a re-trained one—preserving performance on standard metrics while failing to recognize small perturbations of forgotten data.

---

[9]`GA` can also be viewed as learning on randomized labels for forget samples.

[10]Unlearning accuracy is defined as $1-$ forget accuracy, i.e., the classification error on the forget set $\mathcal{S}_f$.

Table 1: Performance summary of various unlearning methods on image classification, including the proposed `RURK`, `Original`, `Re-train`, and 11 baseline approaches, evaluated under two unlearning scenarios: small CIFAR-5 and full CIFAR-10, both using ResNet-18. Results are reported in the format $a_{\pm b}$, indicating the mean $a$ and standard deviation $b$ over 3 independent trials. The absolute performance gap relative to `Re-train` is shown in (blue). For methods that fail to recover the forget-set knowledge within 30 training epochs, the re-learn time is reported as ">30". See Appendix C.1 for a complete table for Small CIFAR-5.

| Datasets | Methods | Evaluation Metrics | | | | | |
| --- | --- | --- | --- | --- | --- | --- | --- |
| | | Retain Acc. (%) | Unlearn Acc. (%) | Test Acc. (%) | MIA Acc. (%) | Avg. Gap | Re-learn Time (# Epoch) |
| Small CIFAR-5 | Original | $99.93_{\pm 0.10}(0.03)$ | $0.00_{\pm 0.00}(8.33)$ | $95.37_{\pm 0.80}(0.57)$ | $4.67_{\pm 3.30}(22.33)$ | 7.82 | - |
| | Re-train | $99.96_{\pm 0.05}(0.00)$ | $8.33_{\pm 3.30}(0.00)$ | $94.80_{\pm 0.85}(0.00)$ | $27.00_{\pm 5.66}(0.00)$ | 0.00 | $3.33_{\pm 0.47}$ |
| | CR | $99.56_{\pm 0.47}(0.40)$ | $14.00_{\pm 5.66}(5.67)$ | $91.80_{\pm 0.99}(3.00)$ | $58.17_{\pm 0.79}(31.17)$ | 10.06 | - |
| | Fisher | $92.67_{\pm 0.63}(7.29)$ | $12.67_{\pm 0.94}(4.34)$ | $88.80_{\pm 1.98}(6.00)$ | $47.33_{\pm 6.13}(20.33)$ | 9.49 | $3.00_{\pm 1.41}$ |
| | NTK | $99.93_{\pm 0.10}(0.03)$ | $7.00_{\pm 0.00}(1.33)$ | $95.37_{\pm 0.80}(0.57)$ | $16.00_{\pm 4.24}(11.00)$ | 3.23 | $4.67_{\pm 0.47}$ |
| | RURK | $99.52_{\pm 0.37}(0.44)$ | $5.67_{\pm 2.36}(2.66)$ | $93.83_{\pm 0.90}(0.97)$ | $33.33_{\pm 12.26}(6.33)$ | 2.60 | $2.00_{\pm 0.00}$ |
| CIFAR-10 | Original | $100.00_{\pm 0.00}(0.00)$ | $0.00_{\pm 0.00}(9.47)$ | $94.76_{\pm 0.05}(1.46)$ | $0.07_{\pm 0.02}(22.43)$ | 8.34 | - |
| | Re-train | $100.00_{\pm 0.00}(0.00)$ | $9.47_{\pm 0.61}(0.00)$ | $93.30_{\pm 0.20}(0.00)$ | $22.50_{\pm 0.60}(0.00)$ | 0.00 | $17.33_{\pm 6.65}$ |
| | GD | $99.98_{\pm 0.01}(0.02)$ | $0.00_{\pm 0.00}(9.47)$ | $94.29_{\pm 0.07}(0.99)$ | $0.10_{\pm 0.00}(22.4)$ | 8.22 | $0.20_{\pm 0.04}$ |
| | NGD | $97.53_{\pm 0.03}(2.47)$ | $10.67_{\pm 0.61}(1.20)$ | $90.70_{\pm 0.11}(2.60)$ | $3.70_{\pm 0.53}(18.80)$ | 6.27 | $20.67_{\pm 14.61}$ |
| | GA | $95.41_{\pm 0.04}(4.59)$ | $61.37_{\pm 0.17}(51.91)$ | $85.98_{\pm 0.11}(7.32)$ | $0.00_{\pm 0.00}(22.5)$ | 21.25 | $1.00_{\pm 0.00}$ |
| | NegGrad+ | $99.28_{\pm 0.01}(0.72)$ | $14.00_{\pm 0.44}(4.53)$ | $92.02_{\pm 0.02}(1.28)$ | $18.18_{\pm 0.43}(4.32)$ | 2.71 | $1.00_{\pm 0.00}$ |
| | EU-k | $99.34_{\pm 0.03}(0.66)$ | $1.12_{\pm 0.08}(8.35)$ | $92.97_{\pm 0.08}(0.33)$ | $0.87_{\pm 0.10}(21.63)$ | 7.74 | $4.33_{\pm 3.40}$ |
| | CF-k | $100.00_{\pm 0.00}(0.00)$ | $0.00_{\pm 0.00}(9.47)$ | $94.38_{\pm 0.04}(1.08)$ | $0.42_{\pm 0.10}(22.08)$ | 8.16 | $1.00_{\pm 0.00}$ |
| | SCRUB | $99.61_{\pm 0.02}(0.39)$ | $12.45_{\pm 0.29}(2.98)$ | $92.70_{\pm 0.03}(0.60)$ | $7.10_{\pm 0.14}(15.40)$ | 4.84 | $> 30$ |
| | SSD | $96.49_{\pm 0.02}(3.51)$ | $66.45_{\pm 0.82}(56.98)$ | $88.59_{\pm 0.06}(4.71)$ | $5.12_{\pm 0.44}(17.38)$ | 20.65 | $7.33_{\pm 0.47}$ |
| | RURK | $99.55_{\pm 0.07}(0.45)$ | $14.63_{\pm 2.17}(5.16)$ | $92.60_{\pm 0.24}(0.70)$ | $18.20_{\pm 2.47}(4.30)$ | 2.65 | $> 30$ |

**Performance comparison.** Table 1 compares the performance of `Original`, the ideal reference `Re-train`, our proposed `RURK`, and 11 baseline unlearning methods across two unlearning scenarios. `RURK` achieves the smallest average performance gap to `Re-train` in both settings. We apply SGD with $\tau = 0.03$ and $v = 1$ in Term (ii) of Eq. (6), making `RURK`'s time complexity comparable to other gradient-based approaches such as `GD`, `NGD`, and `NegGrad+`. In the small CIFAR-5 scenario, `NTK` shows a competitive Avg. Gap but suffers from the highest MIA accuracy, indicating vulnerability to privacy attacks. `Fisher` over-forgets the target samples, significantly degrading utility, as seen in its low retain and test accuracies. For CIFAR-10, `NegGrad+` performs similarly to `RURK` in Avg. Gap but shows a much shorter re-learn time than `Re-train`, implying an implicit retention of forget-set knowledge. `GA` and `SSD` aggressively erase forget-set information, achieving high unlearn accuracy but at the cost of test accuracy dropping below 90%. Both `EU-k` and `CF-k` (with $k = 5$) re-train or fine-tune only the linear and last two residual blocks, but the remaining layers of ResNet-18 still retain substantial information about the forget set. `SCRUB` has the closest overall performance to `RURK` when including re-learn time, but unlike `RURK`, it requires extra memory to store a student model, whereas `RURK` supports in-place updates. We defer the performance of other unlearning baselines on small CIFAR-5, and an ablation study on the hyper-parameters (e.g., $\tau$, $v$ and $\lambda$) in `RURK` to Appendix C.

**Residual knowledge.** Figure 2 presents[11] estimates of residual knowledge $\hat{r}_\tau((\mathbf{x}, y))$ under varying perturbation radii $\tau$, computed using Gaussian noise ($p = 2$) with $c = 100$ samples from $\mathcal{B}_p(\mathbf{x}, \tau)$ (cf. Eq. (4)). As expected, `Original` consistently exhibits residual knowledge greater than 1. In small CIFAR-5, although `NTK` performs comparably to `RURK` in Avg. Gap and re-learn time (Table 1), its residual knowledge remains above 1 due to linearization under the neural tangent kernel, which ignores higher-order terms. This highlights residual knowledge as a necessary complement to unlearn accuracy, MIA, and re-learn time. In contrast, `RURK` maintains values near 1 for $\tau < 0.01$ and effectively suppresses them at larger $\tau$. For CIFAR-10, `GD` and `CF-k` retain high residual knowledge similar to `Original`, underscoring the need for access to forget samples, as done by `RURK`. Conversely, `GA` and `SSD` yield residual knowledge below 1 even at $\tau = 0$, indicating over-unlearning. `EU-k` updates only the final layers, leaving residuals in earlier representations. Methods such as `NGD`, `NegGrad+`, `EU-k`, and `SCRUB` follow trends similar to `RURK`, though `RURK` achieves more stable control near $\tau = 0.01$ and stronger suppression beyond that. While `NGD` reduces residual knowledge more effectively than `GD` or `CF-k` through controlled weight noise, it remains less effective than `RURK`. `NegGrad+`, with a similar objective, also retains more residual knowledge, validating the role of Term (ii) in Eq. (6). For residual knowledge under other attacks, see Appendix C.1; for adversarial disagreement and unlearn accuracy of the perturbed forget samples, see Appendix C.5.

---

[11]By Eq. (4), the residual knowledge of `Re-train` is exactly 1 for all $\tau$, so it is omitted from the figure.

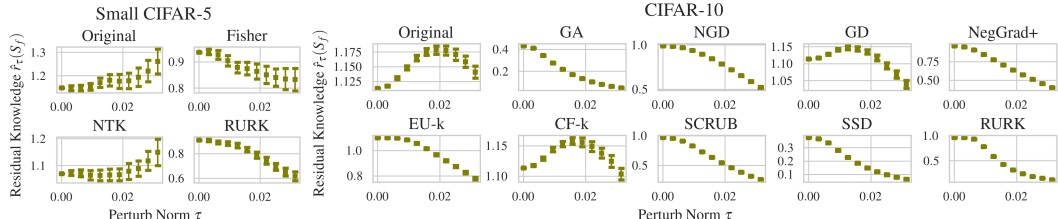

Figure 2: Residual knowledge $\hat{r}_\tau(\mathcal{S}_f)$ of the proposed RURK, Original, and other unlearning methods across two unlearning scenarios, evaluated under varying perturbation norms $\tau$.

| Datasets | Methods | Evaluation Metrics | | | |
|---|---|---|---|---|---|
| | | Retain Acc. (%) | Unlearn Acc. (%) | Test Acc. (%) | Avg. Gap |
| ImageNet-100 | Original | $80.75_{\pm3.06}(5.69)$ | $3.59_{\pm0.19}(9.03)$ | $72.67_{\pm1.18}(4.23)$ | $6.32$ |
| | Re-train | $75.06_{\pm1.54}(0.00)$ | $12.62_{\pm1.40}(0.00)$ | $68.44_{\pm1.51}(0.00)$ | $0.00$ |
| | GD | $82.29_{\pm0.37}(7.23)$ | $6.26_{\pm2.38}(7.13)$ | $74.44_{\pm1.39}(6.01)$ | $6.79$ |
| | NegGrad+ | $81.86_{\pm1.86}(6.80)$ | $40.67_{\pm1.85}(27.28)$ | $73.53_{\pm1.06}(5.10)$ | $13.06$ |
| | SSD | $80.35_{\pm3.00}(5.30)$ | $90.15_{\pm13.92}(76.77)$ | $71.81_{\pm1.05}(3.38)$ | $28.48$ |
| | RURK | $83.09_{\pm0.26}(8.03)$ | $11.62_{\pm0.69}(1.00)$ | $75.19_{\pm0.59}(6.75)$ | $5.26$ |

Figure 3: Performance summary and residual knowledge (following Table 1 and Figure 2) of selected unlearning methods on ImageNet-100.

**Large-Scale ImageNet-100.** In Figure 3, RURK consistently achieves the smallest average performance gap compared to GD and NegGrad+, demonstrating strong scalability. Residual knowledge analysis shows that GD still retains forgotten information due to its lack of explicit control over the forget set, while SSD fails to fully suppress forget-related neurons in large architectures like ResNet-50. Although NegGrad+ mitigates residual knowledge more effectively, its alignment with the re-trained model remains weaker than RURK. Overall, residual knowledge persists beyond small-scale settings, and RURK remains the most effective method for mitigating it in larger, more complex models.

# 6 Final remark

Our study reveals a key limitation of existing unlearning methods: although they erase direct memorization of forget samples, they often fail to remove implicit generalization around them, causing unlearned models to recognize perturbed variants more often than re-trained ones. RURK mitigates this by incorporating perturbed forget samples, disrupting such generalization and reducing residual knowledge while preserving retain-set accuracy.

**Limitations.** Ideally, unlearned and re-trained models should be statistically indistinguishable not only on original inputs but also on all perturbations within $\mathcal{B}_p(\mathbf{x}, \tau)$. However, controlling adversarial disagreement across such perturbations is computationally infeasible—it is as hard as achieving perfect adversarial robustness. Moreover, since the re-trained model is typically unavailable during unlearning, we can only bound one side of the residual knowledge (i.e., $r_\tau(\mathcal{S}_f) \leq 1$; see § 4.2).

**Future directions.** First, our probabilistic analysis in § 3 could be extended to account for hypothesis class complexity and linked to the transferability of adversarial examples (Tramèr et al., 2017). Second, the indistinguishability–robustness trade-off introduced in § 4.2 opens up new directions in both unlearning and adversarial robustness, warranting deeper investigation. Third, our mitigation objective (cf. Eq. (6)) resembles a reverse of adversarial training, suggesting an open question of whether adversarial training or distributionally robust optimization could in fact impede unlearning or amplify residual knowledge. Moreover, in the context of MIA against unlearned models, adversaries could exploit side information—such as the minimal perturbation required for a model to re-recognize a forgotten sample. A smaller perturbation norm may indicate prior inclusion in training, increasing inference success. This perspective opens a promising research direction for future MIA studies in unlearning, as recent work has begun leveraging model variation near training points or perturbation dynamics to strengthen attacks (Jalalzai et al., 2022; Del Grosso et al., 2022; Xue et al., 2025). Finally, extending residual knowledge beyond classification to generative tasks—such as image or text generation—remains an open challenge. In these settings, unlearning often targets concepts rather than samples, yet tasks like image-to-image generation (Fan et al., 2023; Li et al., 2024) show that perturbing partial inputs can still regenerate forgotten content, suggesting analogous residual behaviors. In large language models, the compositional nature of text and differing notions of "forget" and "retain" complicate formalization, though phenomena like relearning and jailbreaking indicate conceptual parallels (Hu et al., 2024; Shumailov et al., 2024; Liu et al., 2025).

**Broader impact.** Residual knowledge introduces new privacy risks. For instance, if a user opts out of a biometric-based payment system (e.g., palm or facial recognition), residual traces may still allow adversaries to craft perturbed inputs that the system accepts—potentially enabling unauthorized access. Such vulnerabilities undermine trust in ML systems and challenge current interpretations of the "Right to be Forgotten."

**Disclaimer.** This paper was prepared for informational purposes by the Global Technology Applied Research center and Artificial Intelligence Research group of JPMorgan Chase & Co. This paper is not a product of the Research Department of JPMorgan Chase & Co. or its affiliates. Neither JPMorgan Chase & Co. nor any of its affiliates makes any explicit or implied representation or warranty and none of them accept any liability in connection with this paper, including, without limitation, with respect to the completeness, accuracy, or reliability of the information contained herein and the potential legal, compliance, tax, or accounting effects thereof. This document is not intended as investment research or investment advice, or as a recommendation, offer, or solicitation for the purchase or sale of any security, financial instrument, financial product or service, or to be used in any way for evaluating the merits of participating in any transaction.

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

# A  Omitted proofs and theoretical results

We first introduce (and prove) the following useful lemmas to facilitate the proofs of the propositions. The first lemma is a variant of the $(\epsilon, \delta)$-indistinguishability in Definition 1.

**Lemma A.1** (Probabilistic indistinguishability)**.** *If $X$ and $Y$ are $(\epsilon, \delta)$-indistinguishable, then with probability at least $1 - 2\delta/(1 - e^{-\epsilon})$ over $z$ drawn from $\mathsf{support}(X) \cup \mathsf{support}(Y)$, we have*

$$e^{-2\epsilon} \Pr[Y = z] \leq \Pr[X = z] \leq e^{2\epsilon} \Pr[Y = z]. \tag{A.1}$$

*Proof.* We first consider the complement event of the right inequality in Eq. (A.1). Define $\mathcal{Z} = \{z | \Pr[X = z] > e^{2\epsilon} \Pr[Y = z]\}$, we directly have $\Pr[X \in \mathcal{Z}] > e^{2\epsilon} \Pr[Y \in \mathcal{Z}]$. By $(\epsilon, \delta)$-indistinguishable, we have

$$\delta \geq \Pr[X \in \mathcal{Z}] - e^{\epsilon} \Pr[Y \in \mathcal{Z}] > (e^{2\epsilon} - e^{\epsilon}) \Pr[Y \in \mathcal{Z}]$$
$$\Rightarrow \Pr[Y \in \mathcal{Z}] < \frac{\delta}{e^{2\epsilon} - e^{\epsilon}} = \frac{\delta}{e^{2\epsilon}(1 - e^{-\epsilon})}. \tag{A.2}$$

Now consider the complement event of the left inequality in Eq. (A.1) and define $\mathcal{Z}' = \{z | \Pr[X = z] < e^{-2\epsilon} \Pr[Y = z]\}$, by similar algebra, we have

$$\delta \geq \Pr[Y \in \mathcal{Z}'] - e^{\epsilon} \Pr[X \in \mathcal{Z}'] > (e^{2\epsilon} - e^{\epsilon}) \Pr[X \in \mathcal{Z}']$$
$$\Rightarrow \Pr[X \in \mathcal{Z}'] < \frac{\delta}{e^{2\epsilon} - e^{\epsilon}}$$
$$\Rightarrow \Pr[Y \in \mathcal{Z}'] \leq e^{\epsilon} \Pr[X \in \mathcal{Z}'] + \delta < e^{\epsilon} \frac{\delta}{e^{2\epsilon} - e^{\epsilon}} + \delta = \frac{\delta}{1 - e^{-\epsilon}}. \tag{A.3}$$

Since $\frac{\delta}{1 - e^{-\epsilon}}$ is always larger than $\frac{\delta}{e^{2\epsilon}(1 - e^{-\epsilon})}$ (by a factor of $e^{2\epsilon}$), by combining Eq. (A.2) and Eq. (A.3), we have

$$\Pr[Y \in \mathcal{Z} \cup \mathcal{Z}'] = \Pr[\{\Pr[X = z] > e^{2\epsilon} \Pr[Y = z]\} \text{ or } \{\Pr[X = z] < e^{-2\epsilon} \Pr[Y = z]\}]$$
$$\leq \frac{\delta}{1 - e^{-\epsilon}}. \tag{A.4}$$

With the same analysis on $\Pr[X \in \mathcal{Z}]$ and $\Pr[X \in \mathcal{Z}']$, we have $\Pr[X \in \mathcal{Z} \cup \mathcal{Z}'] \leq \frac{\delta}{1 - e^{-\epsilon}}$. Finally, putting together the the probability bounds on $\Pr[Y \in \mathcal{Z} \cup \mathcal{Z}']$ and $\Pr[X \in \mathcal{Z} \cup \mathcal{Z}']$, we have

$$\Pr[e^{-2\epsilon} \Pr[Y = z] \leq \Pr[X = z] \leq e^{2\epsilon} \Pr[Y = z]] \leq 1 - \frac{\delta}{1 - e^{-\epsilon}}, \tag{A.5}$$

as desired. $\qquad \square$

The second lemma relates to the isoperimetric inequality (Talagrand, 1995). Before stating it, we define the notion of a $\tau$-expansion of a set. Given a set $\mathcal{C} \subset \mathbb{R}^d$, its $\tau$-expansion with respect to the $\ell_p$-norm is defined as

$$\mathcal{E}(\mathcal{C}, p, \tau) \triangleq \{\mathbf{c} \in \mathcal{R}^d | \|\mathbf{c} - \mathbf{g}\|_p \leq \tau \text{ for some } \mathbf{g} \in \mathcal{C}\}. \tag{A.6}$$

The isoperimetric inequality is used to characterize the normalized surface area of the $\tau$-expansion of a half unit sphere, showing how it grows with the expansion radius $\tau$ and the dimension $d$. This property is leveraged in the condition of Proposition 2, and is formally stated below.

**Lemma A.2** (Milman and Schechtman (1986))**.** *Let $\mathcal{C}$ be the half unit sphere in $\mathbb{R}^d$, i.e., $\mathcal{C} \in \mathbb{S}^{d-1}$ and $\mathsf{surf}(\mathcal{C})/\mathsf{surf}(\mathbb{S}^{d-1}) \geq 1/2$, then with $p = 2$ or $p = \infty$, the $\tau$-expansion $\mathcal{C}$ has the surface area that satisfies*

$$\frac{\mathsf{surf}(\mathcal{E}(\mathcal{C}, p, \tau))}{\mathsf{surf}(\mathbb{S}^{d-1})} \geq 1 - \left(\frac{\pi}{8}\right)^{1/2} \exp\left(-\frac{d-1}{2}\tau^2\right). \tag{A.7}$$

The third lemma utilizes Lemma A.1 to lower bound the probability of disagreement, i.e., the expected value of the disagreement indicator function $k(\mathbf{x})$.

**Lemma A.3** (Lower bound on disagreement probability)**.** *If the unlearned $M(A(\mathcal{S}), \mathcal{S}, \mathcal{S}_f)$ and retrained $A(\mathcal{S}_r)$ models satisfies $(\epsilon, \delta)$-unlearning, then with probability $2\delta/(1 - e^{-\epsilon})$, the probability of disagreement among the two models on a sample $\mathbf{x} \in \mathcal{S}$ is lower bounded by*

$$\mathbb{E}[k(\mathbf{x})] = \mathbb{E}[\mathbb{1}(M(\mathbf{x}) \neq A(\mathbf{x}))] = \Pr[M(\mathbf{x}) \neq A(\mathbf{x})] > 1 - \mathcal{O}\left(e^{-2\epsilon}\right). \quad \text{(A.8)}$$

*Proof.* We prove the lower bound by directly decompose $\Pr[M(\mathbf{x}) \neq A(\mathbf{x})]$, i.e.,

$$\Pr[M(\mathbf{x}) \neq A(\mathbf{x})] = \int\limits_{h \in \mathcal{H}} \Pr\left[m(\mathbf{x}) \neq a(\mathbf{x}) | M = m = h, A = a \neq h\right] \Pr[M \neq h, A = h] dh$$

$$+ \int\limits_{h \in \mathcal{H}} \Pr\left[m(\mathbf{x}) \neq a(\mathbf{x}) | M = m = h, A = a = h\right] \Pr[M = h, A = h] dh$$

(A.9)

The second integral in Eq. (A.9) is zero since $\Pr\left[m(\mathbf{x}) \neq a(\mathbf{x}) | M = m = h, A = a = h\right] = 0$. Moreover, by Lemma A.1, with probability $1 - 2\delta/(1 - e^{-\epsilon})$, we have

$$\Pr[M \neq h, A = h] = \Pr[M \neq h] \Pr[A = h] \leq \Pr[A = h] - e^{-2\epsilon} \Pr[A = h]^2. \quad \text{(A.10)}$$

In other words, with $2\delta/(1 - e^{-\epsilon})$, we have

$$\Pr[M \neq h, A = h] = \Pr[M \neq h] \Pr[A = h] > \Pr[A = h] - e^{-2\epsilon} \Pr[A = h]^2. \quad \text{(A.11)}$$

The combination of Eq. (A.9) and Eq. (A.11) yield

$$\Pr[M(\mathbf{x}) \neq A(\mathbf{x})] = \int\limits_{h \in \mathcal{H}} \Pr\left[m(\mathbf{x}) \neq a(\mathbf{x}) | M = m = h, A = a \neq h\right] \Pr[M \neq h, A = h] dh$$

$$> \int\limits_{h \in \mathcal{H}} \Pr[A = h] - e^{-2\epsilon} \Pr[A = h]^2 dh$$

$$= \int\limits_{h \in \mathcal{H}} \Pr[A = h] dh - e^{-2\epsilon} \int\limits_{h \in \mathcal{H}} \Pr[A = h]^2 dh$$

$$= 1 - \mathcal{O}\left(e^{-2\epsilon}\right),$$

(A.12)

the desired result, as $\int\limits_{h \in \mathcal{H}} \Pr[A = h] dh = 1$ and $\int\limits_{h \in \mathcal{H}} \Pr[A = h]^2 dh$ is a constant. $\qquad \square$

The fourth lemma provides the relation between the adversarial disagreement $k_\tau(\mathbf{x}') = \mathbb{1}(M(\mathbf{x}) \neq A(\mathbf{x}))$ and the residual knowledge $r_\tau((\mathbf{x}, y))$.

**Lemma A.4** (Bounding adversarial disagreement with the residual knowledge.)**.** *The expected value of $k_\tau(\mathbf{x}')$ over the perturbation distribution is upper and lower bounded by*

$$r_\tau((\mathbf{x}, y)) \Pr\left[a(\mathbf{x}') = y\right] (1 - \Pr\left[a(\mathbf{x}') = y\right]) \leq \mathbb{E}\left[k_\tau(\mathbf{x}')\right] \leq 1 - r_\tau((\mathbf{x}, y)) \Pr\left[a(\mathbf{x}') = y\right]^2. \quad \text{(A.13)}$$

*Proof.* We prove the lemma by directly following the definition of adversarial disagreement. First, we have

$$\mathbb{E}\left[k_\tau(\mathbf{x}')\right] = \Pr\left[m(\mathbf{x}') \neq a(\mathbf{x}')\right] = \sum_{l \in \mathcal{Y}} \Pr\left[m(\mathbf{x}') = l, a(\mathbf{x}') \neq l\right]$$

$$= \sum_{l \in \mathcal{Y}} \Pr\left[m(\mathbf{x}') = l\right] (1 - \Pr\left[a(\mathbf{x}') = l\right]) \geq \Pr\left[m(\mathbf{x}') = y\right] (1 - \Pr\left[a(\mathbf{x}') = y\right]),$$

(A.14)

where the final inequality follows by isolating the contribution of the true label $y$. By substituting the definition of residual knowledge from Eq. (4), we obtain

$$\mathbb{E}\left[k_\tau(\mathbf{x}')\right] \geq r_\tau((\mathbf{x}, y)) \Pr\left[a(\mathbf{x}') = y\right] (1 - \Pr\left[a(\mathbf{x}') = y\right]). \quad \text{(A.15)}$$

Similarly, by evaluating $1 - \mathbb{E}\left[k_\tau(\mathbf{x}')\right] = \Pr\left[m(\mathbf{x}') = a(\mathbf{x}')\right]$, we have

$$1 - \mathbb{E}\left[k_\tau(\mathbf{x}')\right] = \Pr\left[m(\mathbf{x}') = a(\mathbf{x}')\right] = \sum_{l \in \mathcal{Y}} \Pr\left[m(\mathbf{x}') = l, a(\mathbf{x}') = l\right]$$

$$\geq \Pr\left[m(\mathbf{x}') = y\right]\Pr\left[a(\mathbf{x}') = y\right] = r_\tau((\mathbf{x}, y))\Pr\left[a(\mathbf{x}') = y\right]^2 \qquad \text{(A.16)}$$

$$\Rightarrow \mathbb{E}\left[k_\tau(\mathbf{x}')\right] \leq 1 - r_\tau((\mathbf{x}, y))\Pr\left[a(\mathbf{x}') = y\right]^2.$$

$\square$

## A.1 Proof of Proposition 1

Suppose the unlearned $M(A(\mathcal{S}), \mathcal{S}, \mathcal{S}_f)$ and re-trained $A(\mathcal{S}_r)$ models are $(\epsilon, \delta)$-indistinguishable, and let $\mathbf{x}$ be a fixed sample. From Lemma A.1, we have that with probability $2\delta/(1 - e^\epsilon)$, $h$ drawn from $\mathsf{support}(M) \cup \mathsf{support}(A)$, we have

$$\Pr[M = h] \leq e^{-2\epsilon}\Pr[A = h] \text{ or } e^{2\epsilon}\Pr[A = h] \leq \Pr[M = h]. \qquad \text{(A.17)}$$

Therefore, for all $\mathcal{X}' \subseteq \mathcal{X}$, by using the right inequality in Eq. (A.17),

$$\Pr[g_\mathbf{x}(m) \in \mathcal{X}'] = \Pr[g_\mathbf{x}(h) \in \mathcal{X}', M = h] = \Pr[g_\mathbf{x}(h) \in \mathcal{X}']\Pr[M = h]$$

$$\geq e^{2\epsilon}\Pr[g_\mathbf{x}(h) \in \mathcal{X}']\Pr[A = h] = e^{2\epsilon}\Pr[g_\mathbf{x}(a) \in \mathcal{X}']. \qquad \text{(A.18)}$$

Similarly, for the other inequality, we have

$$\Pr[g_\mathbf{x}(m) \in \mathcal{X}'] \leq e^{-2\epsilon}\Pr[g_\mathbf{x}(h) \in \mathcal{X}']\Pr[A = h] = e^{-2\epsilon}\Pr[g_\mathbf{x}(a) \in \mathcal{X}']. \qquad \text{(A.19)}$$

Eq. (A.18) and Eq. (A.19) together give the desired result.

## A.2 Proof of Proposition 2

Let $\mathcal{R} = \{\mathbf{x} \in \mathbb{S}^{d-1} | k(\mathbf{x}) = 0\}$ to be the region of the sphere that has agreement between the unlearned and re-trained models. By assumption, we have $\mathsf{surf}(\mathcal{R})/\mathsf{surf}(\mathbb{S}^{d-1}) \leq 1/2$. Accordingly, we define $\bar{\mathcal{R}} = \{\mathbf{x} \in \mathbb{S}^{d-1} | k(\mathbf{x}) = 1\}$ to be the complement of $\mathcal{R}$, denoting the region of the sphere that has disagreement between the unlearned and re-trained models. Since $\mathsf{surf}(\bar{\mathcal{R}})/\mathsf{surf}(\mathbb{S}^{d-1}) \geq 1/2$, its $\tau$-expansion $\mathcal{E}(\bar{\mathcal{R}}, p, \tau)$, by Lemma A.2, is at least as large as the epsilon expansion of a half sphere; that is

$$\frac{\mathsf{surf}(\mathcal{E}(\bar{\mathcal{R}}, p, \tau))}{\mathsf{surf}(\mathbb{S}^{d-1})} \geq 1 - \left(\frac{\pi}{8}\right)^{1/2}\exp\left(-\frac{d-1}{2}\tau^2\right). \qquad \text{(A.20)}$$

Note that the $\tau$-expansion $\mathcal{E}(\bar{\mathcal{R}}, p, \tau)$ represents the set of samples that either has $k(\mathbf{x}) = 1$ or admits a $\mathbf{x}' \in \mathcal{B}_p(\mathbf{x}, \tau)$ such that $k(\mathbf{x}') = 1$. We can therefore define a set $\mathcal{E}^c$ that is the complement of $\mathcal{E}(\bar{\mathcal{R}}, p, \tau)$ with surface area satisfies

$$\frac{\mathsf{surf}(\mathcal{E}^c)}{\mathsf{surf}(\mathbb{S}^{d-1})} \leq \left(\frac{\pi}{8}\right)^{1/2}\exp\left(-\frac{d-1}{2}\tau^2\right). \qquad \text{(A.21)}$$

The probability to draw a sample in $\mathcal{E}^c$ is then upper bounded by

$$\sup_\mathbf{x} \Pr[M(\mathbf{x}) = A(\mathbf{x})] \times \left(\frac{\pi}{8}\right)^{1/2}\exp\left(-\frac{d-1}{2}\tau^2\right). \qquad \text{(A.22)}$$

Using Lemma A.3, we know that with probability $2\delta/(1 - e^{-\epsilon})$,

$$\sup_\mathbf{x} \Pr[M(\mathbf{x}) = A(\mathbf{x})] = \sup_\mathbf{x} 1 - \Pr[M(\mathbf{x}) \neq A(\mathbf{x})] = 1 - \left(1 - \mathcal{O}\left(e^{-2\epsilon}\right)\right) = \mathcal{O}\left(e^{-2\epsilon}\right). \qquad \text{(A.23)}$$

Putting Eq. (A.22) and Eq. (A.23) together, we know that the probability to draw a sample in $\mathcal{E}(\bar{\mathcal{R}}, p, \tau)$ is at least

$$\left(\frac{2\delta}{1 - e^{-\epsilon}}\right) \times \left\{1 - \sup_\mathbf{x}\Pr[M(\mathbf{x}) = A(\mathbf{x})] \times \left(\frac{\pi}{8}\right)^{1/2}\exp\left(-\frac{d-1}{2}\tau^2\right)\right\}$$

$$= \left(\frac{2\delta}{1 - e^{-\epsilon}}\right) \times \left\{1 - \mathcal{O}\left(e^{-2\epsilon}\right) \times \left(\frac{\pi}{8}\right)^{1/2}\exp\left(-\frac{d-1}{2}\tau^2\right)\right\} \qquad \text{(A.24)}$$

$$= \left(\frac{2\delta}{1 - e^{-\epsilon}}\right) \times \left\{1 - \mathcal{O}\left[\left(\frac{\pi}{8}\right)^{1/2}\exp\left(-2\epsilon - \frac{d-1}{2}\tau^2\right)\right]\right\}.$$

# B  Details on the experimental setup

We provide implementation details of `RURK` in § 4.2, and introduce the unlearning baselines in § 5, including their mathematical formulations, operational interpretations, implementation specifics, and GitHub links. Finally, we summarize the dataset descriptions, training setups, and evaluation metrics.

## B.1  Comparison with Pawelczyk et al. (2025)

Pawelczyk et al. (2025) investigate the failure of machine unlearning under data poisoning attacks. In their setup, the training set $S_{\text{train}}$ consists of two disjoint subsets: $S_{\text{clean}}$ and $S_{\text{poison}}$, where $S_{\text{poison}}$ contains maliciously corrupted samples generated via targeted or backdoor poisoning. The original model is trained on both subsets, and the goal of unlearning is to remove the influence of $S_{\text{poison}}$ so that the resulting model resembles one retrained solely on $S_{\text{clean}}$. Ideally, the unlearned model should be "clean" and exhibit no poisoning effects. However, Pawelczyk et al. (2025) find that even after applying state-of-the-art unlearning algorithms, the unlearned model continues to display backdoor behavior—revealed through persistent correlations with the poison pattern—unlike the clean retrained model. They conclude that unlearning cannot fully sanitize a poisoned model, as residual effects of the attack remain embedded in its representations and decision boundary.

Our work, in contrast, examines *clean unlearning* under adversarial perturbations at test time rather than under poisoned data at training time. Both the retain and forget sets in our framework are clean; the original model $M_o$ is trained on $S_{\text{retain}} \cup S_{\text{forget}}$, and the unlearned model $M_u$ aims to forget $S_{\text{forget}}$ so as to be indistinguishable from a retrained model $M_r$ trained only on $S_{\text{retain}}$. We show that even when $M_u$ and $M_r$ are distributionally indistinguishable, they can still diverge in behavior on *perturbed forget samples* $S_{\text{perturbed}}$—samples not seen during training but lying in the local neighborhood of $S_{\text{forget}}$. Specifically, $M_u$ tends to re-recognize these perturbed samples more often than $M_r$, revealing a form of *residual knowledge* that persists despite formal unlearning guarantees.

The key distinction between the two settings lies in the source and nature of the residual information. In Pawelczyk et al. (2025), the unlearned model retains *explicit* knowledge of malicious patterns learned from $S_{\text{poison}}$ (e.g., a backdoor trigger). In our work, the residual information is *implicit*, arising from the generalization structure around the clean forget samples rather than from any poisoned signal. Thus, while Pawelczyk et al. (2025) highlight the limits of unlearning under training-time data corruption, our work reveals that even with entirely clean data, residual generalization can cause unlearned and retrained models to remain distinguishable—posing a distinct privacy risk not attributable to poisoning.

## B.2  Details of `RURK` in § 4.2

We first provide the pseudo-code of `RURK` in the following algorithm box Algorithm 1.

We implement `RURK` using PyTorch (`torch`) (Paszke, 2019), and ensure reproducibility by fixing three random seeds: $[131, 42, 7]$. During optimization, we use a batch size of $128$, the standard cross-entropy loss (`torch.nn.CrossEntropyLoss()`), and the SGD optimizer (`torch.optim.SGD`) with a learning rate of $0.01$, momentum of $0.90$, and weight decay of $5 \times 10^{-4}$. To stabilize training, we apply a cosine annealing learning rate scheduler (`torch.optim.lr_scheduler.CosineAnnealingLR`) and cap the total number of iterations at $200$. Additionally, we clip the gradient norm to $1.0$ using `torch.nn.utils.clip_grad_norm_`.

For both unlearning scenarios (small CIFAR-5 and CIFAR-10), we set the perturbation budget $\tau = 0.03$ and use the TorchAttacks library (Kim, 2020) to identify the vulnerable set $\mathcal{V}((\mathbf{x}, y), \tau)$. To improve efficiency, we define $\mathcal{V}((\mathbf{x}, y), \tau)$ as the entire perturbation ball $\mathcal{B}_p(\mathbf{x}, \tau)$ and set $v = 1$, meaning that only a single perturbed sample is drawn from a multivariate Gaussian distribution centered at $\mathbf{x}$ with standard deviation $\tau$.

We configure the hyper-parameters as follows: for small CIFAR-5, we use $N = 2$ and $\lambda_f = \lambda_a = 0.03$; for CIFAR-10, we set $N = 2$, $\lambda_f = 0.03$, and $\lambda_a = 0.00045$. Since $v = 1$ and the perturbation is generated via Gaussian noise, the inner loop in line 12 of Algorithm 1 executes only once, making the operations in lines 11–15 constant-time. As a result, the computational complexity of `RURK` is comparable to fine-tuning-based methods such as `GA`, `GD`, `NegGrad+`, and `NGD`.

**Algorithm 1:** RURK Implementation

**input** : Original $\mathbf{w} = A(\mathcal{S})$; Forget Set $\mathcal{S}_f$; Retain Set $\mathcal{S}_r$; Loss Function $\ell$.
**output** : Unlearned Model $\mathbf{w}_{\text{RURK}}$
**parameter** : Perturbation Norm $\tau$; Regularization Strength $\lambda_f$, $\lambda_a$; Sample Size $v$; # Epoch $N$

1  RLoader $\leftarrow$ MakeDataLoader($\mathcal{S}_r$);
2  FLoader $\leftarrow$ MakeDataLoader($\mathcal{S}_f$);
3  $\mathbf{w}_{\text{RURK}} \leftarrow \mathbf{w}$;
4  **for** $epoch \leftarrow 1$ **to** $N$ **do**
5     **for** *batchIdx, (retainData, retainTargets) in* RLoader **do**   // Scan over all Batches
6        retainOutput $\leftarrow h_{\mathbf{w}}$(*retainData*);
7        RLoss $\leftarrow$ Loss(*retainOutput, retainTargets*);
8        forgetData, forgetTarget = nextIter(FLoader);
9        forgetOutput $\leftarrow h_{\mathbf{w}}$(*forgetDataData*);
10       FLoss $\leftarrow$ Loss(*forgetOutput, forgetTarget*);
11       vulnerableSet = [];
12       **for** $i \leftarrow 1$ **to** $v$ **do**       // Construct the Vulnerable Set $\mathcal{V}((\mathbf{x}, y), \tau)$
13          vulnerableSet $\leftarrow$ vulnerableSet + findAdv(*forgetData, forgetTarget, $\tau$*);
14       **end**
15       advForgetOutput $\leftarrow h_{\mathbf{w}}$(*vulnerableSet*);
16       AdvFLoss $\leftarrow$ Loss(*advForgetOutput, forgetTarget*);
17       Loss $\leftarrow$ RLoss $-\lambda_f$FLoss $-\lambda_a$AdvFLoss ;     // Entire Loss in Eq. (6)
18       $\mathbf{w}_{\text{RURK}} \leftarrow \mathbf{w}_{\text{RURK}} + \nabla_{\mathbf{w}_{\text{RURK}}}$Loss ;     // SGD Optimization
19    **end**
20 **end**
   **return** : $\mathbf{w}_{\text{RURK}}$

However, if stronger adversaries like FGSM or PGD are used to identify the vulnerable set, the computational complexity increases from $\mathcal{O}(N)$ to $\mathcal{O}(N \times v)$. Note that unlike NGD, we do not inject additional noise into the gradient update steps.

**More discussion on RURK.** The regularization over the forget set in the loss $L(\mathbf{w}, A(\mathcal{S}))$ highlights two key features of our proposed unlearning method, RURK. First, under mild conditions, the unlearning procedure satisfies certified unlearning via Rényi Differential Privacy (RDP), providing formal guarantees—a certificate of unlearning. Second, enhancing robustness against vulnerable perturbations $\mathcal{V}((\mathbf{x}, y), \tau)$ may increase the distinguishability of the unlearned model from the ideal re-trained model.

To support the first point, consider the case where the forget set contains a single sample (i.e., $|\mathcal{S}_f| = 1$), so that the original dataset $\mathcal{S}$ and the retain set $\mathcal{S}_r$ are adjacent. Assume the original model $m$ was trained to satisfy $(\alpha, \epsilon_0)$-RDP. Further, suppose the unlearning procedure minimizes the loss in Eq. (6) using Projected Noisy Gradient Descent (PNGD). As shown by Chien et al. (2024), the Markov chain defined by PNGD updates of the form

$$\mathbf{w}_{t+1} = \Pi_R \left( \mathbf{w}_t - \eta \nabla L(\mathbf{w}_t, A(\mathcal{S})) + \sqrt{2\eta\sigma^2}W_t \right), \text{ with } \mathbf{w}_1 = A(\mathcal{S}) \tag{B.25}$$

where $W_t \sim \mathcal{N}(0, \mathbb{I}_d)$ and $\Pi_R$ denotes projection onto the ball of radius $R$, converges to a stationary distribution if the loss $L(\mathbf{w}, A(\mathcal{S}))$ from Eq. (6) is continuous. The first term of Eq. (6) is continuous by standard assumptions, as the loss $\ell(\cdot, (\mathbf{x}, y))$ is typically continuous in the weights. To show continuity of the second term, consider $g(\mathbf{w}) = \min_{\mathbf{z} \in \mathcal{B}_p(\mathbf{x}, \tau)} \ell(\mathbf{w}, (\mathbf{z}, y))$. Let $\mathbf{z}^*$ be a minimizer; then $g(\mathbf{w}) = \ell(\mathbf{w}, (\mathbf{z}^*, y))$, which is continuous in $\mathbf{w}$ since $\ell$ is. Therefore, the overall loss is continuous, and the PNGD process converges.

Furthermore, when the loss is $L$-smooth and $M$-Lipschitz, the stationary distribution obtained via PNGD satisfies $(\alpha, \epsilon)$-RDP for some $\epsilon$ depending on $L$, $M$, and convexity properties of the loss (Chien et al., 2024). For forget sets with multiple elements, one can sequentially apply this procedure, preserving Rényi unlearning at each step.

To justify the second point, observe that Term (ii) in Eq. (6) impacts the loss landscape's smoothness. Specifically, the term $\kappa(\mathbf{w}, (\mathbf{x}, y))$ tends to increase with the perturbation radius $\tau$, as the adversarial loss over a larger ball is generally higher. This leads to a gradient norm that increases with $\tau$,

implying that the smoothness and Lipschitz constants of the overall loss also grow with $\tau$. According to Theorem 3.2 in Chien et al. (2024), the Rényi distinguishability parameter $\epsilon$ is inversely related to these constants, implying a trade-off: achieving higher robustness (larger $\tau$) increases model distinguishability (larger $\epsilon$).

The assumptions underlying these results are mild and standard in the literature. We assume the loss is $L$-smooth and $M$-Lipschitz, and that the learning dynamics for the original dataset satisfy the Log-Sobolev Inequality (LSI) with constant $C_{\mathrm{LSI}}$ (Gross, 1975). Moreover, the `Original` $A(\mathcal{S})$ can be trained to satisfy $(\alpha, \epsilon_0)$-RDP under LSI using the same PNGD framework, as discussed by Chien et al. (2024).

## B.3 Existing unlearning algorithms

Here, we provide detailed descriptions of the 11 unlearning baseline methods used in § 5, along with links to their corresponding GitHub repositories.

**Certified Removal (CR).** Guo et al. (2020) propose a single-step Newton–Raphson update to remove the influence of forget samples. Assuming that the empirical risk $L(\mathbf{w}, \mathcal{S})$ is twice differentiable with continuous second derivatives, and letting $\mathbf{w}^* = \arg\min_{\mathbf{w} \in \mathcal{W}} L(\mathbf{w}, \mathcal{S}_r)$ denote the empirical risk minimizer over the retain set $\mathcal{S}_r$, the Taylor expansion of the gradient $\nabla L$ around $\mathbf{w}^*$ yields:

$$\nabla L(\mathbf{w}^, \mathcal{S}_r) \approx \nabla L(\mathbf{w}_o, \mathcal{S}_r) + H_{\mathbf{w}_o}(\mathbf{w}^- \mathbf{w}_o), \tag{B.26}$$

where $\mathbf{w}_o = A(\mathcal{S})$ denotes the Original model parameters, and $H_{\mathbf{w}_o} = \nabla^2 L(\mathbf{w}_o, \mathcal{S}_r)$ is the Hessian of the empirical risk over $\mathcal{S}_r$ evaluated at $\mathbf{w}_o$. Re-arranging the terms gives:

$$\mathbf{w}^* \approx \mathbf{w}_o - H_{\mathbf{w}_o}^{-1} \nabla L(\mathbf{w}_o, \mathcal{S}_r), \tag{B.27}$$

since $\nabla L(\mathbf{w}^*, \mathcal{S}_r) = 0$ by optimality. The certified removal method (CR) then defines the unlearned model as

$$M(\mathbf{w}_o, \mathcal{S}, \mathcal{S}_f) = \mathbf{w}_o - \lambda H_{\mathbf{w}_o}^{-1} \nabla L(\mathbf{w}_o, \mathcal{S}_r), \tag{B.28}$$

where $\lambda$ is a step-size parameter. Under the assumption of convexity (ensuring the uniqueness of minimizers) and bounded approximation error, Guo et al. (2020, Theorem 1) show that this procedure guarantees $(\epsilon, \delta)$-certified removal, i.e., $M(\mathbf{w}_o, \mathcal{S}, \mathcal{S}_f) \overset{\epsilon,\delta}{\approx} \mathbf{w}^*$.

In our implementation, we follow the official codebase at https://github.com/facebookresearch/certified-removal. Since CR is designed for binary linear models, we use a pre-trained ResNet-18 as a non-private feature extractor (excluding its final linear layer), and apply the Newton update only to the final layer. Specifically, we train $|\mathcal{Y}|$ one-vs-rest logistic regression classifiers on the extracted features and set $\lambda = 0.1$.

**Fisher Unlearning (Fisher).** Fisher (Golatkar et al., 2020a) is one of the simplest unlearning mechanisms, which removes the influence of forget samples by directly perturbing the model weights with Gaussian noise (cf.§2.1). Unlike standard Gaussian perturbation, the variance of the noise is scaled according to the inverse of the Fisher information matrix. Specifically, the unlearned model is defined as:

$$M(\mathbf{w}, \mathcal{S}, \mathcal{S}_f) = \mathbf{w} + n, \quad n \sim \mathcal{N}(0, \alpha \mathbf{B}^{-1}), \tag{B.29}$$

where $\mathbf{B}$ denotes the Hessian matrix $\nabla^2 L(\mathbf{w}, \mathcal{S}_r)$. However, computing the exact Hessian is computationally intractable even for moderately sized neural networks, and the matrix may not be positive definite. To address this, Golatkar et al. (2020a) approximate the Hessian using the Levenberg–Marquardt algorithm, yielding a semi-positive-definite matrix closely related to the Fisher information matrix (Martens, 2020), which motivates the name of the method. In our implementation, we follow the official codebase and adopt the same hyper-parameters as in https://github.com/AdityaGolatkar/SelectiveForgetting.

**Neural Tangent Kernel Unlearning (NTK).** NTK (Golatkar et al., 2020b) extends the Newton update idea from CR by applying the Neural Tangent Kernel (NTK) linearization of neural networks. The NTK matrix between two datasets $\mathcal{S}_1$ and $\mathcal{S}_2$ is defined as:

$$K(\mathcal{S}_1, \mathcal{S}_2) \triangleq \nabla_{\mathbf{w}} h_{\mathbf{w}}(\mathcal{S}_1) \nabla_{\mathbf{w}} f_{\mathbf{w}}(\mathcal{S}_2)^\top, \tag{B.30}$$

where $h_{\mathbf{w}}$ denotes the network output and $h_{\mathbf{w}}$ may optionally denote a scalar output function (e.g., pre-activation logits). Let $\mathbf{w}^* = \arg\min_{\mathbf{w} \in \mathcal{W}} L(\mathbf{w}, \mathcal{S})$ and $\mathbf{w}_r = \arg\min_{\mathbf{w} \in \mathcal{W}} L(\mathbf{w}, \mathcal{S}_r)$ be the minimizers of the empirical risk over the full and retain sets, respectively. By linearizing the network output around $\mathbf{w}^*$ using the NTK approximation, NTK enables a closed-form update that shifts the model weights from $\mathbf{w}^*$ to an approximation of $\mathbf{w}_r$ via a one-shot adjustment:

$$\mathbf{w}_r \approx M(\mathbf{w}^*, \mathcal{S}, \mathcal{S}_f) = \mathbf{w}^* + \mathbf{P} \nabla_{\mathbf{w}^*} h_{\mathbf{w}^*}(\mathcal{S}_f)^\top \mathbf{M} \mathbf{V}, \tag{B.31}$$

where the projection matrix $\mathbf{P} = \mathbf{I} - \nabla_{\mathbf{w}^*} h_{\mathbf{w}^*}(\mathcal{S}_r)^\top K(\mathcal{S}_r, \mathcal{S}_r)^{-1} \nabla_{\mathbf{w}^*} h_{\mathbf{w}^*}(\mathcal{S}_r)$ ensures that the gradient contributions from the forget set are orthogonalized against those from the retain set. The matrix $\mathbf{M}$ is defined as

$$\mathbf{M} = \left[ K(\mathcal{S}_f, \mathcal{S}_f) - K(\mathcal{S}_r, \mathcal{S}_f)^\top K(\mathcal{S}_r, \mathcal{S}_r)^{-1} K(\mathcal{S}_r, \mathcal{S}_f) \right]^{-1}, \tag{B.32}$$

and the re-weighting matrix $\mathbf{V}$ is given by

$$\mathbf{V} = (y_f - h_{\mathbf{w}^*}(\mathcal{S}_f)) + K(\mathcal{S}_r, \mathcal{S}_f)^\top K(\mathcal{S}_f, \mathcal{S}_f)^{-1}(y_r - h_{\mathbf{w}^*}(\mathcal{S}_r)), \tag{B.33}$$

where $y_f$ and $y_r$ denote the ground truth labels for the forget and retain sets, respectively. We follow the official implementation and use the same settings and hyper-parameters as in `https://github.com/AdityaGolatkar/SelectiveForgetting`.

**Gradient Descent (GD).** GD (Neel et al., 2021) is one of the simplest unlearning algorithms. It continues training the original model `Original` on the retain set $\mathcal{S}_r$ using standard gradient descent. Specifically, the unlearned model $M$ is obtained by iteratively applying the update:

$$\mathbf{w}_{t+1} \leftarrow \mathbf{w}_t - \eta, g_t(\mathbf{w}_t), \quad \text{with} \quad \mathbf{w}_1 = A(\mathcal{S}),$$

where $\eta$ is the step size and $g_t(\mathbf{w}_t)$ is a (mini-batch) stochastic gradient of the empirical loss over $\mathcal{S}_r$, i.e., $\frac{1}{|\mathcal{S}_r|} \sum_{\mathbf{s} \in \mathcal{S}_r} \ell(\mathbf{w}, \mathbf{s})$. The key intuition behind GD is that when the forget set is small (i.e., $|\mathcal{S}_f| \ll |\mathcal{S}|$), the minimizers of the loss functions over $\mathcal{S}$ and $\mathcal{S}_r$ are expected to be close. Therefore, a few steps of gradient descent starting from the original model $\mathbf{w}_1 = A(\mathcal{S})$ can efficiently move the parameters toward a minimizer of the updated training objective. Building on this intuition, Neel et al. (2021) also provide theoretical guarantees for the effectiveness of GD in both convex and certain non-convex settings.

Our implementation follows `https://github.com/ChrisWaites/descent-to-delete`, using the hyper-parameters :

- SGD optimizer with a lr = 1e-2, momentum = 0.9, and weight decay = 1e-4.
- Random seed for the data loader is 7; random seeds for repeated experiments are 131, 42, 7.
- Batch size = 128
- Number of epochs = 10

**Noisy Gradient Descent (NGD).** NGD (Chourasia and Shah, 2023) is a simple extension of GD in which Gaussian noise is added to each gradient update. The unlearned model is obtained by iteratively applying the update:

$$\mathbf{w}_{t+1} \leftarrow \mathbf{w}_t - \eta \left( g_t(\mathbf{w}_t) + \xi_t \right), \quad \text{with} \quad \mathbf{w}_1 = A(\mathcal{S}),$$

where $\eta$ is the step size, $g_t(\mathbf{w}_t)$ denotes a (mini-batch) stochastic gradient of the empirical loss over the retain set $\mathcal{S}_r$, i.e., $\frac{1}{|\mathcal{S}_r|} \sum_{\mathbf{s} \in \mathcal{S}_r} \ell(\mathbf{w}, \mathbf{s})$, and $\xi_t \sim \mathcal{N}(0, \sigma^2)$ is an independent Gaussian noise term added at each iteration. The key distinction between NGD and GD lies in the injection of noise during optimization. This stochasticity not only increases the robustness of the unlearning process but also enables formal privacy guarantees, as demonstrated in the context of certified unlearning via Rényi differential privacy (Chien et al., 2024). Notably, a similar noise-injection mechanism is employed in the DP-SGD algorithm for training models with differential privacy guarantees (Abadi et al., 2016).

Our implementation follows `https://github.com/Graph-COM/Langevin_unlearning` and the same hyper-parameters as GD with

- $(\sigma, \eta) = (0.03, 0.1)$, trained for 10 epochs for the small CIFAR-5 settings with sample unlearning.
- $(\sigma, \eta) = (0.03, 0.1)$, trained for 2 epochs for the CIFAR-10 settings with sample unlearning.

**Gradient Descent (GA).** GA (Graves et al., 2021) aims to remove the influence of the forget set $\mathcal{S}_f$ from a trained model by reversing the learning process through gradient ascent. It stores all gradient updates involving $\mathcal{S}_f$ during the initial training phase, and unlearning is then performed by applying the exact reverse updates—i.e., gradient ascent—using the stored gradients. However, this

approach is highly memory-intensive and becomes impractical for large-scale models. To address this limitation, Jang et al. (2022) proposed a more scalable variant, in which unlearning is achieved by performing mini-batch gradient ascent on the forget set. Specifically, the model is updated by minimizing the negated loss $-L(\mathbf{w}, \mathcal{S}_f)$, effectively implementing ascent steps that counteract the original training influence of $\mathcal{S}_f$.

Our implementation follows `https://github.com/joeljang/knowledge-unlearning`. using the similar hyper-parameters as GD:

- SGD optimizer with a lr = 1e-5, momentum = 0.9, and weight decay = 1e-4, clipping gradient norm = 1.

- Random seed for the data loader is 7; random seeds for repeated experiments are 131, 42, 7.

- Batch size = 128

- Number of epochs = 1 for Small-CIFAR-5 (both sample and class unlearning) and CIFAR-10 with sample unlearning; while 3 for CIFAR-10 with class unlearning

**Negative Gradient Plus (`NegGrad+`).** `NegGrad+` (Kurmanji et al., 2024) fine-tunes the model by simultaneously minimizing the loss on the retain set $\mathcal{S}_r$ and maximizing the loss on the forget set $\mathcal{S}_f$, effectively negating the gradient contributions from the latter. Specifically, the unlearned model is obtained by optimizing the following objective:

$$L(\mathbf{w}, \mathcal{S}_r) - \beta L(\mathbf{w}, \mathcal{S}_f), \tag{B.34}$$

where $\beta$ is a hyper-parameter that controls the strength of unlearning by weighting the loss contribution from $\mathcal{S}_f$. `NegGrad+` shares conceptual similarity with the Gradient Ascent (GA) method, as both perform loss maximization on the forget set to induce forgetting. However, `NegGrad+` is empirically more stable and yields better performance, as it simultaneously enforces loss minimization on the retain set, ensuring that useful knowledge is preserved during unlearning.

Our implementation follows `https://github.com/meghdadk/SCRUB`, using similar hyper-parameters as GD and GA:

- SGD optimizer with a lr = 1e-2, momentum = 0.9, and weight decay = 1e-4, clipping gradient norm = 1.

- $\beta = 0.001$

- Random seed for the data loader is 7; random seeds for repeated experiments are 131, 42, 7.

- Batch size = 128

- Number of epochs = 1 for Small-CIFAR-5 (both sample and class unlearning) and CIFAR-10 with sample unlearning; while 3 for CIFAR-10 with class unlearning

**Exact Unlearning the last $k$ layers (`EU-k`).** `EU-k` (Goel et al., 2022) is a simple, parameter-efficient unlearning method designed for deep learning models, requiring access only to the retain set $\mathcal{S}_r$. Given a parameter $k$, `EU-k` retrains from scratch the last $k$ layers of the neural network—those closest to the output layer—while keeping the earlier layers fixed. By adjusting $k$, `EU-k` provides a tunable trade-off between unlearning effectiveness and computational efficiency.

Our implementation is based on the official repository at `https://github.com/shash42/Evaluating-Inexact-Unlearning`, using the following hyper-parameters:

- SGD optimizer with a lr = 1e-2, momentum = 0.9, and weight decay = 1e-4.

- Last fully-connected layer, and last two residual blocks are reinitialized, and only fine-tune those parameters.

- Random seed for the data loader is 7; random seeds for repeated experiments are 131, 42, 7.

- Batch size = 128

- Number of epochs = 10

**Catastrophically Forgetting the last $k$ layers (CF-k).** CF-k (Goel et al., 2022) builds on the observation that neural networks tend to forget information about data samples encountered early in training—a phenomenon known as catastrophic forgetting (French, 1999). Similar to EU-k, CF-k focuses on the last $k$ layers of the network; however, instead of re-training these layers from scratch, it fine-tunes them starting from the original model parameters $\mathbf{w} = A(\mathcal{S})$, using only the retain set $\mathcal{S}_r$, while keeping all earlier layers frozen.

As with EU-k, our implementation follows the official repository at https://github.com/shash42/Evaluating-Inexact-Unlearning, using the following hyper-parameters:

- SGD optimizer with a lr = 1e-2, momentum = 0.9, and weight decay = 1e-4.
- Last fully-connected layer, and last two residual blocks are fine-tuned.
- Random seed for the data loader is 7; random seeds for repeated experiments are 131, 42, 7.
- Batch size = 128
- Number of epochs = 10

**SCalable Remembering and Unlearning unBound (SCRUB).** SCRUB (Kurmanji et al., 2024) is one of the state-of-the-art unlearning methods for deep learning. It formulates the unlearning task within a student–teacher framework: given a trained teacher model $\mathbf{w}_T$, the goal is to train a student model $\mathbf{w}$ that selectively imitates the teacher. Specifically, the student should retain the teacher's behavior on the retain set $\mathcal{S}_r$ while diverging significantly from it on the forget set $\mathcal{S}_f$, as measured by the KL divergence. To achieve this, SCRUB optimizes a modified knowledge distillation objective (Hinton et al., 2014):

$$\mathbb{E}_{(\mathbf{x},y)\sim\mathcal{S}_r}\left[\alpha, D_{\mathsf{KL}}(h_{\mathbf{w}_T}(\mathbf{x})|h_{\mathbf{w}}(\mathbf{x})) + \gamma, \ell(\mathbf{w},(\mathbf{x},y))\right] - \mathbb{E}_{(\mathbf{x},y)\sim\mathcal{S}_f}\left[D_{\mathsf{KL}}(h_{\mathbf{w}_T}(\mathbf{x})|h_{\mathbf{w}}(\mathbf{x}))\right],$$
(B.35)

where $\alpha$ and $\gamma$ are hyper-parameters that balance knowledge retention and task performance. The first expectation encourages the student to match the teacher on the retain set while also performing well on the classification task, and the second term enforces divergence from the teacher on the forget set.

Our implementation follows https://github.com/meghdadk/SCRUB, with the hyper-parameters:

- SGD optimizer with a lr = 5e-4, momentum = 0.9, and weight decay = 5e-4.
- Random seed for the data loader is 7; random seeds for repeated experiments are 131, 42, 7.
- Batch size = 128
- $\alpha = 0.001$
- $\gamma = 1$ for small CIFAR-5 settings with both sample and class unlearning
- $\gamma = 0.6$ for CIFAR-10 setting with sample unlearning and $\gamma = 75$ for CIFAR-10 with class unlearning

**Selective Synaptic Dampening (SSD).** SSD was introduced by Foster et al. (2024) as a method to unlearn a specific forget set from a neural network without retraining it from scratch. Building on ideas similar to Fisher, but with a more refined approach, SSD selectively dampens weights that exhibit disproportionately high influence—measured via the Fisher information—on the forget set relative to the retain set. Given a model with parameters $\mathbf{w}$, let $\mathbf{F}_r$ and $\mathbf{F}_f$ denote the Fisher information matrices computed over the retain set $\mathcal{S}_r$ and the forget set $\mathcal{S}_f$, respectively. Unlearning is performed by scaling each parameter $\mathbf{w}_i$ according to its relative Fisher sensitivity:

$$\mathbf{w}_i = \begin{cases} \beta\mathbf{w}_i & \text{if } \mathbf{F}_{f,i} > \alpha\mathbf{F}_{r,i}, \\ \mathbf{w}_i & \text{otherwise,} \end{cases}$$
(B.36)

where $\mathbf{F}_{f,i}$ and $\mathbf{F}_{r,i}$ denote the $i$-th diagonal entries of the Fisher matrices for $\mathcal{S}_f$ and $\mathcal{S}_r$, respectively. The hyper-parameter $\alpha$ controls the threshold for selecting influential weights, and the dampening factor $\beta$ is defined as $\beta = \min_i \lambda\mathbf{F}_{r,i}/\mathbf{F}_{f,i}, 1$ for a tunable parameter $\lambda$.

Our implementation is based on the official repository at https://github.com/if-loops/selective-synaptic-dampening, using the following hyper-parameters:

- $\lambda = 1.0$
- $\alpha = 10.0$ for CIFAR-10 and 100.0 for Small-CIFAR-5.

### B.4 Training and evaluation details

Here, we provide details on dataset preparation, unlearning settings, and evaluation procedures.

**Dataset and unlearning settings.** We evaluate unlearning methods in the context of image classification using CIFAR-10 (Krizhevsky et al., 2009) and ResNet-18 (He et al., 2016), focusing on the random sample unlearning setting across two scenarios. The CIFAR-10 dataset contains 60,000 color images of size $32 \times 32$ pixels, evenly distributed across 10 classes: airplane, automobile, bird, cat, deer, dog, frog, horse, ship, and truck. Each class includes 5,000 training and 1,000 test samples. CIFAR-10 is widely used in machine unlearning research—for example, by Kurmanji et al. (2024); Chien et al. (2024); Golatkar et al. (2020a,b); Foster et al. (2024). In the first scenario—small CIFAR-5—we follow the setup of Golatkar et al. (2020a,b) by creating a reduced version of CIFAR-10 with 200 training and 200 test samples from each of the first five classes. From class 0, we randomly select 100 samples (50%) as the forget set $\mathcal{S}_f$. The second scenario uses the full CIFAR-10 dataset. Here, we designate 2,000 samples (50%[12]) from class 0 as the forget set $\mathcal{S}_f$. In addition to sample unlearning, we also consider class unlearning. In this setting, the forget set contains 200 samples from class 0 in the small CIFAR-5 scenario, and 4,000 samples from class 0 in the full CIFAR-10 scenario. All experiments are conducted on an AWS EC2 g5.24xlarge instance.

**Training of `Original` and `Re-train`.** To avoid any external knowledge (e.g., the default ImageNet pre-trained weights from `torchvision` (Marcel and Rodriguez, 2010)), both the `Original` and `Re-train` models are trained entirely from scratch, without loading any pre-trained weights. We use a batch size of 128 and cross-entropy loss (`torch.nn.CrossEntropyLoss()`). For the small CIFAR-5 setting, we first pre-train a ResNet-18 model on the full CIFAR-10 dataset using the SGD optimizer (`torch.optim.SGD(lr=0.4, momentum=0.9, weight_decay=2e-4)`) along with a cosine annealing learning rate scheduler (`torch.optim.lr_scheduler.CosineAnnealingLR(T_max=200)`). We then fine-tune this model on small CIFAR-5 for 25 epochs with a learning rate of 0.01 to obtain the `Original` model reported in Table 1. The `Re-train` model in Table 1 is obtained using the same setup, except the fine-tuning is performed on the retain set $\mathcal{S}_r$ for 100 epochs. For the CIFAR-10 setting, both the `Original` and `Re-train` models are trained from scratch on the full dataset and the retain set, respectively, for 137 epochs using a learning rate of 0.01. For the VGG-11 ablation study, we follow the same training protocol, including the optimizer, scheduler, and learning rate.

**Evaluation details.** The retain, forget, and test accuracies reported in Table 1 are computed by directly evaluating each model on the corresponding retain, forget, and test sets. The unlearning accuracy is defined as $1 -$ forget accuracy. The re-learn time quantifies how easily a model can reacquire knowledge of the forget set. It is defined as the number of fine-tuning epochs required for the model $m$ to satisfy $L(m, \mathcal{S}_f) \leq (1 + \eta)L(\texttt{Original}, \mathcal{S}_f)$, where we set $\eta = 0.05$. This value can be adjusted depending on the desired tolerance.

To evaluate vulnerability to membership inference attacks (MIA), we use a support vector classifier (SVC) implemented via the `scikit-learn` package, with parameters `SVC(C=3, gamma=``auto'', kernel=``rbf'')` (Pedregosa et al., 2011). The SVC is trained to distinguish between samples seen during training and those that were not, based on the model's output likelihood for the correct class label. In the sample unlearning setting, retain samples are labeled as 1 (seen) and test samples as 0 (unseen) during SVC training; forget samples are then evaluated by the SVC, and an ideal unlearned model should cause them to be classified as 0. In the class unlearning setting, retain samples are labeled as 1 and forget samples as 0 for SVC training; evaluation is then conducted on a held-out subset of the forget class. The reported MIA Accuracy in Table 1 corresponds to the SVC's attack failure rate, i.e., the proportion of forget samples classified as unseen.

Both the construction of the vulnerable set $\mathcal{V}((\mathbf{x}, y), \tau)$ and the evaluation of residual knowledge $r_\tau(\mathcal{S}_f)$ involve generating perturbed inputs within the norm ball $\mathcal{B}_p(\mathbf{x}, \tau)$. We implement this using the `torchattacks` package (Kim, 2020). For Gaussian noise (i.e., $p = 2$), we use `torchattacks.GN(model, std=`$\tau$`)`; for FGSM (i.e., $p = \infty$), we use `torchattacks.FGSM(model, eps=`$\tau$`)`; and for PGD, we use `torchattacks.PGD(model,`

---

[12]Each class has 5,000 training samples, with 20% held out for validation.

```
eps=τ, alpha=2/255., steps=pgd_epoch, random_start=True). We apply targeted at-
```
tacks for both FGSM and PGD.

To estimate residual knowledge, we apply these attacks to the unlearned model $m$ to generate per-
turbed inputs $\mathbf{x}'$, and compute both $\Pr[m(\mathbf{x}') = y]$ and $\Pr[a(\mathbf{x}') = y]$ over $c = 100$ independent runs,
using the same perturbation $\mathbf{x}'$ for both models. Constructing the vulnerable set $\mathcal{V}((\mathbf{x}, y), \tau)$ involves
solving the minimization problem $\min_{\mathbf{z} \in \mathcal{B}_p(\mathbf{x}, \tau)} \ell(\mathbf{w}, (\mathbf{z}, y))$. However, since the `torchattacks`
package is designed to solve the adversarial objective $\max_{\mathbf{z} \in \mathcal{B}_p(\mathbf{x}, \tau)} \ell(\mathbf{w}, (\mathbf{z}, y))$, we instead define
a random target label $y' \neq y$ and solve $\max_{\mathbf{z} \in \mathcal{B}_p(\mathbf{x}, \tau)} \ell(\mathbf{w}, (\mathbf{z}, y'))$. To adapt this into our residual
knowledge framework, we flip the sign of the regularization term in Eq. (6), effectively encouraging
the model to associate perturbed variants of forget samples with an incorrect label $y'$. This discour-
ages the model from retaining knowledge of the true label and facilitates unlearning. We repeat this
process $v$ times for each forget sample to construct its corresponding vulnerable set.

## B.5 Details on Figure 1

Figure 1 is based on the UCI Iris dataset (3 classes, 50 samples each, 4 features). For both the
original and re-trained models, we use a simple feedforward neural network with one hidden layer
of 100 neurons. We select 7 forget samples in total from three class for illustration. While this
selection differs from the random sample or class unlearning setup (Lines 297–303), the goal is purely
illustrative—to convey the intuition behind the existence of disagreement and residual knowledge.
The unlearned model is trained via GD.

# C   Additional results and experiments

We include additional experimental results and comparisons, including: (i) comprehensive results on sample unlearning for both the small CIFAR-5 and CIFAR-10 scenarios, covering accuracy metrics and residual knowledge estimates under various adversarial attacks; (ii) results on class unlearning for both small CIFAR-5 and CIFAR-10; (iii) ablation studies on the hyper-parameters of RURK; and (iv) an analysis of the prevalence of residual knowledge across different settings.

## C.1   Complete results on sample unlearning for small CIFAR-5

We present the full version of Table 1 under the small CIFAR-5 setting in Table C.2. Combined with Table 1, the results demonstrate that RURK consistently achieves superior performance—surpassing certified removal methods such as CR, Fisher, and NTK, as well as deep-learning-compatible approaches like SCRUB, NegGrad+, and NGD—by achieving the smallest average gap from the re-trained model Re-train. Notably, the re-learn time of RURK is also comparable to that of Re-train.

We further include the complete residual knowledge curves estimated under different adversarial attacks: Gaussian noise in Figure C.4, FGSM in Figure C.5, and PGD in Figure C.6. For both Gaussian noise and FGSM, most methods exhibit residual knowledge greater than 1—i.e., inheriting information from the Original model—whereas RURK maintains residual knowledge close to 1 for small perturbation norms $\tau$, and reduces it below 1 as $\tau$ increases.

With PGD (10 steps), a stronger attack, we observe that methods such as NTK, NGD, and SCRUB still suffer from residual knowledge. In contrast, methods like GD, GA, NegGrad+, EU-k, CF-k, and SSD maintain residual knowledge near 1 across varying values of $\tau$. In particular, SSD strictly preserves residual knowledge at 1, though this comes at the cost of a larger average gap from Re-train (cf. Table C.2).

Overall, RURK effectively suppresses residual knowledge to values below 1 for small $\tau$, and keeps it close to 1 as $\tau$ increases—striking a favorable balance between preserving accuracy and mitigating residual knowledge.

Table C.2: The complete performance summary of various unlearning methods on small CIFAR-5 (cf. Table 1). Results are reported in the format $a_{\pm b}$, indicating the mean $a$ and standard deviation $b$ over 3 independent trials. The absolute performance gap relative to Re-train is shown in (blue). For methods that fail to recover the forget-set knowledge within 30 training epochs, the re-learn time is reported as ">30".

| Datasets | Methods | Evaluation Metrics | | | | | |
|---|---|---|---|---|---|---|---|
| | | Retain Acc. (%) | Unlearn Acc. (%) | Test Acc. (%) | MIA Acc. (%) | Avg. Gap | Re-learn Time (# Epoch) |
| | Original | $99.93_{\pm0.10}(0.03)$ | $0.00_{\pm0.00}(8.33)$ | $95.37_{\pm0.80}(0.57)$ | $4.67_{\pm3.30}(22.33)$ | 7.82 | - |
| | Re-train | $99.96_{\pm0.05}(0.00)$ | $8.33_{\pm3.30}(0.00)$ | $94.80_{\pm0.85}(0.00)$ | $27.00_{\pm5.66}(0.00)$ | 0.00 | $3.33_{\pm0.47}$ |
| Small CIFAR-5 | CR | $99.56_{\pm0.47}(0.40)$ | $14.00_{\pm5.66}(5.67)$ | $91.80_{\pm0.99}(3.00)$ | $58.17_{\pm0.79}(31.17)$ | 10.06 | - |
| | Fisher | $92.67_{\pm0.63}(7.29)$ | $12.67_{\pm0.94}(4.34)$ | $88.80_{\pm1.98}(6.00)$ | $47.33_{\pm6.13}(20.33)$ | 9.49 | $3.00_{\pm1.41}$ |
| | NTK | $99.93_{\pm0.10}(0.03)$ | $7.00_{\pm0.00}(1.33)$ | $95.37_{\pm0.80}(0.57)$ | $16.00_{\pm4.24}(11.00)$ | 3.23 | $4.67_{\pm0.47}$ |
| | GD | $84.04_{\pm0.42}(15.93)$ | $22.67_{\pm5.19}(14.33)$ | $76.83_{\pm1.04}(17.97)$ | $84.67_{\pm10.84}(27.33)$ | 18.89 | $12.67_{\pm12.97}$ |
| | NGD | $95.07_{\pm1.36}(4.89)$ | $4.67_{\pm0.47}(3.66)$ | $89.33_{\pm2.03}(5.47)$ | $13.33_{\pm2.36}(13.67)$ | 6.92 | $0.67_{\pm0.47}$ |
| | GA | $94.22_{\pm2.83}(5.74)$ | $40.67_{\pm5.19}(32.33)$ | $86.37_{\pm0.24}(8.43)$ | $84.00_{\pm11.31}(26.67)$ | 18.29 | $2.33_{\pm0.47}$ |
| | NegGrad+ | $96.78_{\pm2.04}(3.19)$ | $25.00_{\pm1.41}(16.67)$ | $89.77_{\pm0.66}(5.03)$ | $74.67_{\pm17.91}(17.33)$ | 10.55 | $2.00_{\pm0.00}$ |
| | EU-k | $91.15_{\pm5.92}(8.81)$ | $20.00_{\pm4.24}(11.67)$ | $76.33_{\pm4.05}(18.47)$ | $37.00_{\pm4.24}(20.33)$ | 14.82 | $9.33_{\pm6.85}$ |
| | CF-k | $99.96_{\pm0.05}(0.00)$ | $0.33_{\pm0.47}(8.00)$ | $94.73_{\pm0.75}(0.07)$ | $37.00_{\pm24.04}(20.33)$ | 7.10 | $17.00_{\pm11.43}$ |
| | SCRUB | $99.88_{\pm0.18}(0.08)$ | $1.33_{\pm0.47}(7.00)$ | $94.67_{\pm0.79}(0.13)$ | $18.33_{\pm9.43}(8.67)$ | 3.97 | $1.00_{\pm0.00}$ |
| | SSD | $96.85_{\pm1.20}(3.11)$ | $13.33_{\pm8.01}(5.00)$ | $89.43_{\pm2.88}(5.37)$ | $69.00_{\pm8.49}(11.67)$ | 6.29 | $2.33_{\pm0.47}$ |
| | RURK | $99.52_{\pm0.37}(0.44)$ | $5.67_{\pm2.36}(2.66)$ | $93.83_{\pm0.90}(0.97)$ | $33.33_{\pm12.26}(6.33)$ | 2.60 | $2.00_{\pm0.00}$ |

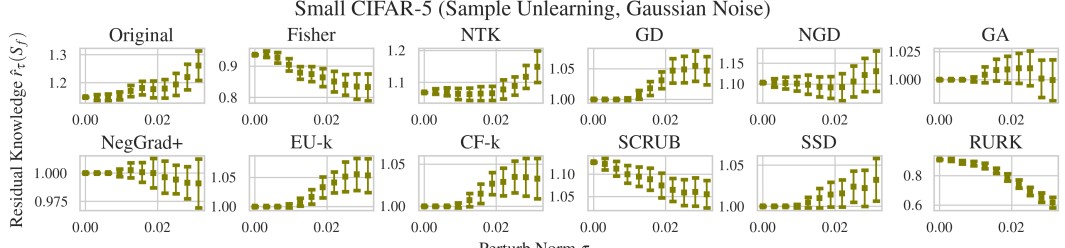

Figure C.4: Residual knowledge $\hat{r}_\tau(\mathcal{S}_f)$ of the proposed `RURK`, `Original`, and other unlearning methods on small CIFAR-5 with sample unlearning, evaluated under varying perturbation norms $\tau$, using Gaussian noise ($p = 2$) to draw $c = 100$ samples from $\mathcal{B}_p(\mathbf{x}, \tau)$.

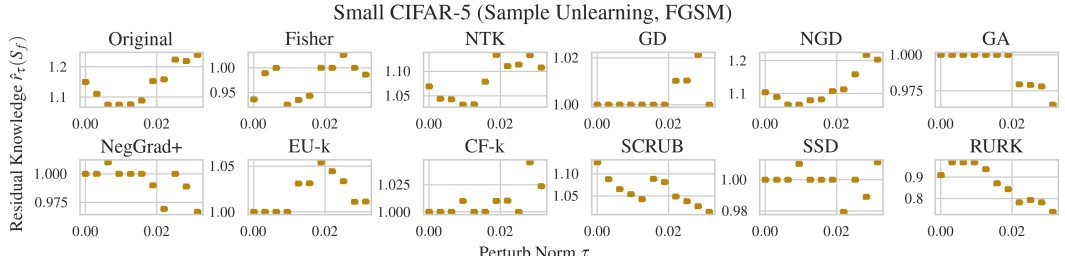

Figure C.5: Residual knowledge $\hat{r}_\tau(\mathcal{S}_f)$ of the proposed `RURK`, `Original`, and other unlearning methods on small CIFAR-5 with sample unlearning, evaluated under varying perturbation norms $\tau$, using targeted FGSM ($p = \infty$) to draw $c = 100$ samples from $\mathcal{B}_p(\mathbf{x}, \tau)$.

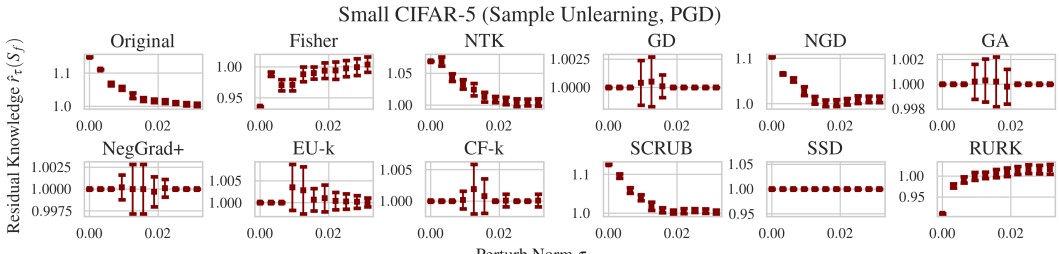

Figure C.6: Residual knowledge $\hat{r}_\tau(\mathcal{S}_f)$ of the proposed `RURK`, `Original`, and other unlearning methods on small CIFAR-5 with sample unlearning, evaluated under varying perturbation norms $\tau$, using targeted PGD ($p = \infty$) to draw $c = 100$ samples from $\mathcal{B}_p(\mathbf{x}, \tau)$.

We provide the residual knowledge estimates under FGSM in Figure C.7 and under PGD in Figure C.8, corresponding to the results presented in Table 1 and Figure 2 of the main text. RURK demonstrates consistent behavior across all adversarial attack types—Gaussian noise, FGSM, and PGD—by maintaining residual knowledge close to 1 for small perturbation radii $\tau$, and effectively suppressing it below 1 as $\tau$ increases.

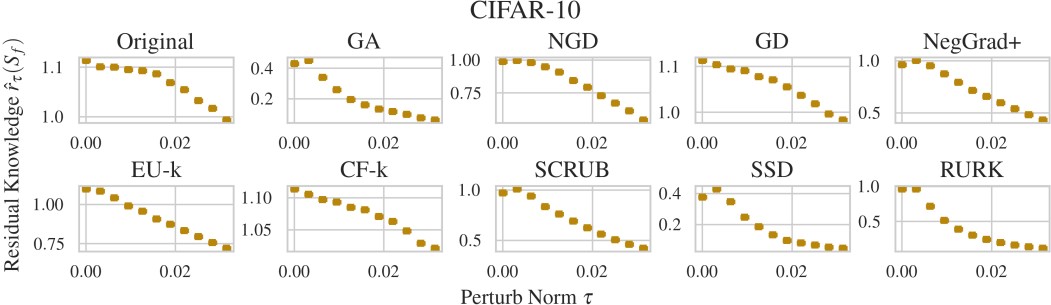

Figure C.7: Residual knowledge $\hat{r}_\tau(\mathcal{S}_f)$ of the proposed RURK, Original, and other unlearning methods on CIFAR-10 with sample unlearning, evaluated under varying perturbation norms $\tau$, using targeted FGSM ($p = \infty$) to draw $c = 100$ samples from $\mathcal{B}_p(\mathbf{x}, \tau)$.

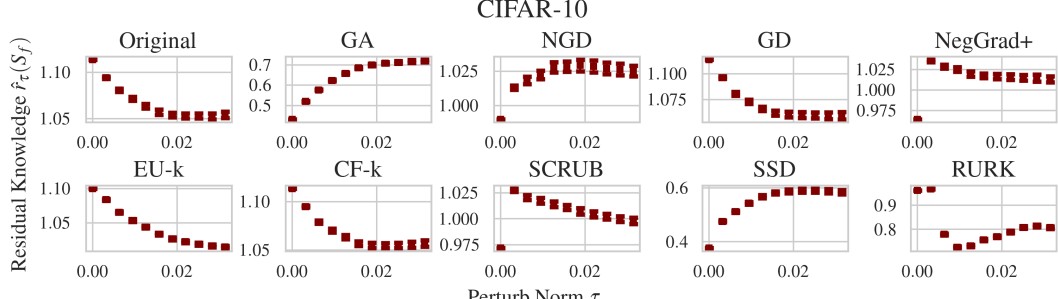

Figure C.8: Residual knowledge $\hat{r}_\tau(\mathcal{S}_f)$ of the proposed RURK, Original, and other unlearning methods on CIFAR-10 with sample unlearning, evaluated under varying perturbation norms $\tau$, using targeted PGD ($p = \infty$) to draw $c = 100$ samples from $\mathcal{B}_p(\mathbf{x}, \tau)$.

## C.2  Class unlearning: Forgetting all samples in a class

In the main text, we focus on the sample unlearning setting, where 50% of the samples from class 0 are unlearned. Here, we provide analogous results for the class unlearning setting, where 100% of the samples from class 0 are removed, on both small CIFAR-5 and CIFAR-10. Note that in class unlearning, the `Re-train` model is never exposed to any samples from class 0. As a result, the unlearning accuracy is 100%—that is, `Re-train` never correctly classifies any forget sample into class 0. In Table C.3, we report the same evaluation metrics as in Table 1, including the average absolute gaps in retain, unlearn, test, and MIA accuracy compared to `Re-train`.

In both small CIFAR-5 and CIFAR-10, `EU-k` achieves the smallest average gap relative to `Re-train`. This result is expected, as class unlearning in this case resembles a transfer learning scenario: from a source domain (small CIFAR-5/CIFAR-10) to a target domain (small CIFAR-4/CIFAR-9), where the forget class is entirely absent. This creates a clear domain shift, allowing `EU-k` to perform well by fine-tuning only the top layers. In contrast, in sample unlearning, both the source and target domains still contain samples from class 0, making transfer learning less effective and leading to poorer performance for `EU-k`.

We also present residual knowledge estimates for the small CIFAR-5 class unlearning scenario under Gaussian noise (Figure C.9), FGSM (Figure C.10), and PGD (Figure C.11). While `RURK` is the second-best performer in terms of average accuracy gap in the small CIFAR-5 setting, it outperforms `EU-k` in suppressing residual knowledge. In the CIFAR-10 class unlearning scenario, `RURK` is the best among popular methods such as `GA`, `SCRUB` and `SSD`.

Table C.3: Performance summary of various unlearning methods for class unlearning. Results are reported in the format $a_{\pm b}$, indicating the mean $a$ and standard deviation $b$ over 3 independent trials. The absolute performance gap relative to `Re-train` is shown in (blue). For methods that fail to recover the forget-set knowledge within 30 training epochs, the re-learn time is reported as ">30".

| Datasets | Methods | Evaluation Metrics | | | | | |
|---|---|---|---|---|---|---|---|
| | | Retain Acc. (%) | Unlearn Acc. (%) | Test Acc. (%) | MIA Acc. (%) | Avg. Gap | Re-learn Time (# Epoch) |
| Small CIFAR-5 | Original | $99.92_{\pm0.12}(0.30)$ | $0.00_{\pm0.00}(100.00)$ | $95.25_{\pm0.88}(1.75)$ | $26.50_{\pm4.95}(73.50)$ | 43.89 | - |
| | Re-train | $99.79_{\pm0.12}(0.17)$ | $100.00_{\pm0.00}(0.00)$ | $93.33_{\pm0.12}(0.00)$ | $100.00_{\pm0.00}(0.00)$ | 0.00 | $1.00_{\pm0.00}$ |
| | Fisher | $93.75_{\pm0.71}(6.04)$ | $14.33_{\pm0.47}(85.67)$ | $89.12_{\pm2.12}(4.20)$ | $54.67_{\pm6.84}(45.33)$ | 35.31 | $3.33_{\pm0.47}$ |
| | NTK | $25.00_{\pm0.00}(74.79)$ | $100.00_{\pm0.24}(0.00)$ | $25.00_{\pm0.00}(68.33)$ | $100.00_{\pm0.00}(0.00)$ | 35.78 | $9.67_{\pm0.47}$ |
| | GD | $67.29_{\pm0.41}(32.50)$ | $93.67_{\pm0.24}(6.33)$ | $62.75_{\pm2.65}(30.58)$ | $89.83_{\pm6.60}(10.17)$ | 19.90 | $31.00_{\pm0.00}$ |
| | NGD | $91.38_{\pm0.00}(8.42)$ | $100.00_{\pm0.00}(0.00)$ | $59.75_{\pm0.00}(33.58)$ | $100.00_{\pm0.00}(0.00)$ | 10.50 | $12.33_{\pm1.89}$ |
| | GA | $98.17_{\pm1.71}(1.62)$ | $33.00_{\pm5.66}(67.00)$ | $92.79_{\pm0.94}(0.54)$ | $64.33_{\pm0.94}(35.67)$ | 26.21 | $2.00_{\pm0.00}$ |
| | NegGrad+ | $99.50_{\pm0.53}(0.29)$ | $19.83_{\pm5.42}(80.17)$ | $93.83_{\pm0.65}(0.50)$ | $58.67_{\pm2.59}(41.33)$ | 30.57 | $2.00_{\pm0.00}$ |
| | EU-k | $99.46_{\pm0.06}(0.33)$ | $100.00_{\pm0.00}(0.00)$ | $84.50_{\pm0.35}(8.83)$ | $100.00_{\pm0.00}(0.00)$ | 2.29 | $31.00_{\pm0.00}$ |
| | CF-k | $99.96_{\pm0.06}(0.17)$ | $42.33_{\pm9.66}(57.67)$ | $95.79_{\pm0.41}(2.46)$ | $97.83_{\pm2.36}(2.17)$ | 15.61 | $31.00_{\pm0.00}$ |
| | SCRUB | $99.96_{\pm0.06}(0.17)$ | $7.33_{\pm5.42}(92.67)$ | $95.5_{\pm0.35}(2.2)$ | $43.67_{\pm1.18}(56.33)$ | 37.84 | $1.00_{\pm0.00}$ |
| | SSD | $98.46_{\pm0.24}(1.33)$ | $7.00_{\pm9.90}(93.00)$ | $93.29_{\pm0.77}(0.04)$ | $42.17_{\pm4.48}(57.83)$ | 38.05 | $2.00_{\pm0.00}$ |
| | RURK | $98.58_{\pm1.65}(1.21)$ | $87.17_{\pm2.59}(12.83)$ | $93.71_{\pm0.65}(0.38)$ | $96.33_{\pm5.19}(3.67)$ | 4.52 | $1.33_{\pm0.47}$ |
| CIFAR-10 | Original | $100.00_{\pm0.00}(0.00)$ | $0.00_{\pm0.00}(100.00)$ | $94.71_{\pm0.06}(0.82)$ | $7.87_{\pm0.29}(92.13)$ | 48.24 | - |
| | Re-train | $100.00_{\pm0.00}(0.00)$ | $100.00_{\pm0.00}(0.00)$ | $93.89_{\pm0.14}(0.00)$ | $100.00_{\pm0.00}(0.00)$ | 0.00 | $12.67_{\pm4.64}$ |
| | GD | $99.98_{\pm0.02}(0.02)$ | $0.09_{\pm0.06}(99.91)$ | $94.33_{\pm0.09}(0.44)$ | $29.27_{\pm0.79}(70.73)$ | 42.78 | $0.67_{\pm0.47}$ |
| | NGD | $99.91_{\pm0.01}(0.09)$ | $20.84_{\pm0.35}(79.16)$ | $94.40_{\pm0.02}(0.51)$ | $96.43_{\pm0.12}(3.57)$ | 20.83 | $8.00_{\pm3.27}$ |
| | GA | $95.51_{\pm0.16}(4.49)$ | $93.60_{\pm0.07}(6.40)$ | $88.60_{\pm0.21}(5.29)$ | $96.47_{\pm0.31}(3.53)$ | 4.93 | $1.00_{\pm0.82}$ |
| | NegGrad+ | $99.93_{\pm0.00}(0.07)$ | $57.19_{\pm0.58}(42.81)$ | $94.30_{\pm0.06}(0.41)$ | $86.80_{\pm0.37}(13.20)$ | 14.12 | $0.67_{\pm0.47}$ |
| | EU-k | $98.38_{\pm0.03}(1.62)$ | $100.00_{\pm0.00}(0.00)$ | $92.15_{\pm0.01}(1.74)$ | $100.00_{\pm0.00}(0.00)$ | 0.84 | $2.67_{\pm0.94}$ |
| | CF-k | $100.00_{\pm0.00}(0.00)$ | $0.06_{\pm0.03}(99.94)$ | $94.57_{\pm0.07}(0.69)$ | $38.43_{\pm1.05}(61.57)$ | 40.55 | $0.00_{\pm0.00}$ |
| | SCRUB | $93.15_{\pm9.26}(6.85)$ | $100.00_{\pm0.00}(0.00)$ | $86.16_{\pm7.88}(7.17)$ | $100.00_{\pm0.00}(0.00)$ | 3.51 | $18.67_{\pm7.41}$ |
| | SSD | $100.00_{\pm0.00}(0.00)$ | $65.27_{\pm1.43}(34.73)$ | $95.09_{\pm0.02}(1.20)$ | $100.00_{\pm0.00}(0.00)$ | 8.98 | $5.67_{\pm1.25}$ |
| | RURK | $99.80_{\pm0.04}(0.20)$ | $95.52_{\pm0.86}(4.48)$ | $93.93_{\pm0.06}(0.04)$ | $97.90_{\pm0.36}(2.10)$ | 1.70 | $1.00_{\pm0.00}$ |

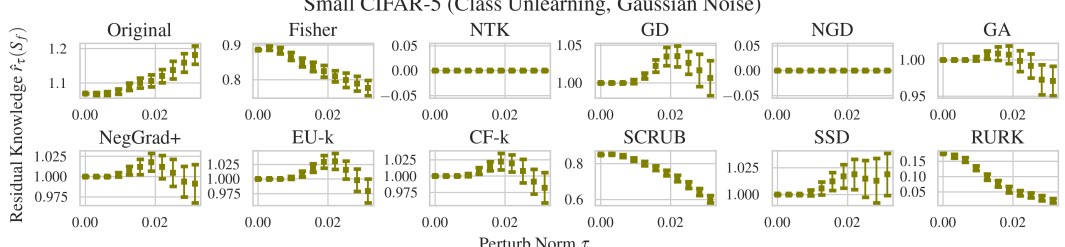

Figure C.9: Residual knowledge $\hat{r}_\tau(\mathcal{S}_f)$ of the proposed RURK, Original, and other unlearning methods on small CIFAR-5 with class unlearning, evaluated under varying perturbation norms $\tau$, using Gaussian noise ($p = 2$) to draw $c = 100$ samples from $\mathcal{B}_p(\mathbf{x}, \tau)$.

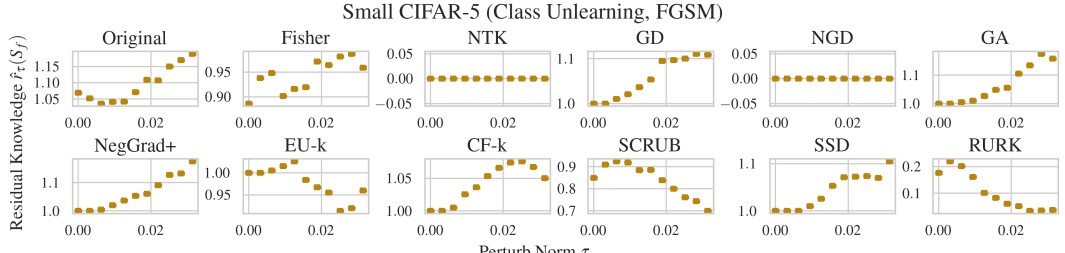

Figure C.10: Residual knowledge $\hat{r}_\tau(\mathcal{S}_f)$ of the proposed RURK, Original, and other unlearning methods on small CIFAR-5 with class unlearning, evaluated under varying perturbation norms $\tau$, using targeted FGSM ($p = \infty$) to draw $c = 100$ samples from $\mathcal{B}_p(\mathbf{x}, \tau)$.

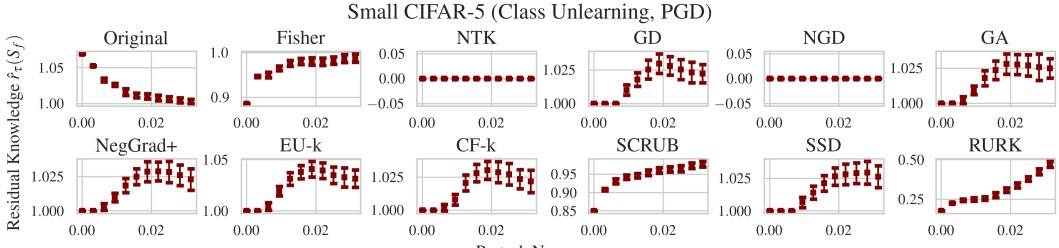

Figure C.11: Residual knowledge $\hat{r}_\tau(\mathcal{S}_f)$ of the proposed RURK, Original, and other unlearning methods on small CIFAR-5 with class unlearning, evaluated under varying perturbation norms $\tau$, using targeted PGD ($p = \infty$) to draw $c = 100$ samples from $\mathcal{B}_p(\mathbf{x}, \tau)$.

## C.3 Ablation studies on `RURK`

In this section, we conduct ablation studies on the key hyper-parameters of `RURK`, including the perturbation radius $\tau$, regularization strength $\lambda$, sample size $v$, types of adversarial attacks, and alternative model architectures. We focus on the sample unlearning setting in the small CIFAR-5 scenario, with all results summarized in Table C.4, Table C.5, Figure C.12, and Figure C.13.

**Different perturbation norms $\tau$.** The perturbation norm $\tau$ determines the radius of the perturbation ball used to evaluate residual knowledge. We set $\tau = 0.03 \approx 8/255$, which aligns with common practice for the maximum adversarial perturbation. To assess the sensitivity of `RURK` to this parameter, we conduct an ablation study over $\tau \in \{0.00, 0.01, 0.02, 0.03, 0.04, 0.10\}$. As shown in Table C.4, when $\tau = 0.00$ or $0.01$, `RURK` fails to unlearn the forget samples from `Original`. However, as $\tau$ increases, the unlearning accuracy improves monotonically, while the test accuracy declines—as expected due to the increased regularization. Notably, `RURK` achieves the best average gap when $\tau = 0.03$. Since $\tau$ is the key hyper-parameter controlling the trade-off between unlearning efficacy and utility, we further visualize the estimated residual knowledge under varying $\tau$ in Figure C.12. The figure shows that increasing $\tau$ improves robustness against residual knowledge under stronger attacks. For instance, under Gaussian noise and FGSM, the residual knowledge remains near 1 for small $\tau$. In contrast, PGD—a stronger adversary—can still uncover residual knowledge at low $\tau$ values, but its effectiveness diminishes significantly when $\tau = 0.1$, indicating improved robustness.

**Different regularization strengths $\lambda$.** The regularization strength $\lambda$ controls the weighting of samples in the vulnerable set within the loss function defined in Eq. (6). We sweep over $\lambda \in \{0.00, 0.01, 0.02, 0.03, 0.04, 0.10\}$ and observe that the trend in the average gap closely mirrors that of $\tau$. This similarity is expected, as both $\tau$ and $\lambda$ jointly influence the extent to which residual knowledge is removed during unlearning.

**Other methods to search for the vulnerable set.** In the main text, we search for samples in the vulnerable set $\mathcal{V}((\mathbf{x}, y), \tau)$ using Gaussian noise. Here, we extend the evaluation to include stronger adversarial methods, such as FGSM and PGD, with varying numbers of attack steps (indicated by the number following PGD in Table C.4). We observe that FGSM—a targeted attack—achieves MIA accuracy comparable to that of `Re-train`. Likewise, PGD with 5 steps also yields MIA accuracy close to `Re-train`, indicating effective removal of forget-set information. These results suggest that targeted attacks (FGSM, PGD) are more effective than untargeted ones like Gaussian noise in eliminating residual knowledge, albeit at the cost of increased unlearning time.

**Different samples size $v$.** The sample size $v$ in Eq.(6) determines how extensively the vulnerable set $\mathcal{V}((\mathbf{x}, y), \tau)$ is explored through sampling. In high-dimensional settings, $\mathcal{V}((\mathbf{x}, y), \tau)$ may contain a large number of perturbed variants. However, due to computational constraints, it is impractical to include all possible samples during training. Instead, we randomly draw $v$ samples from $\mathcal{V}((\mathbf{x}, y), \tau)$ for each forget sample. We experiment with $v \in \{1, 2, 3, 4\}$ and observe that the average gap achieved by `RURK` remains relatively stable across these values. This robustness is expected, as the loss term $\kappa(\mathbf{w}, (\mathbf{x}, y))$ in Eq.(6) is computed as an average over the $v$ sampled perturbations.

**Other learning structures.** Thus far, our evaluation of `RURK` has focused on models trained with ResNet-18. To assess its generality across architectures, we present results on small CIFAR-5 using VGG-11 (Simonyan and Zisserman, 2014), as shown in Table C.5. Compared to `NGD`, `RURK` continues to achieve the smallest average gap. Figure C.13 further illustrates the residual knowledge comparison between `RURK` and `NGD`, demonstrating that `RURK` maintains lower and more stable residual knowledge across perturbations. These results confirm that `RURK` effectively removes forget-set information and controls residual knowledge across different network architectures, including both plain convolutional networks (VGG-11) and those with residual connections (ResNet-18).

Table C.4: Ablation study of `RURK` on small CIFAR-5 with sample unlearning. Results are reported in the format $a_{\pm b}$, indicating the mean $a$ and standard deviation $b$ over 3 independent trials. The absolute performance gap relative to `Re-train` is shown in (blue). For methods that fail to recover the forget-set knowledge within 30 training epochs, the re-learn time is reported as ">30". We bold the results with hyper-parameters the same in the main text.

| Other Parameters | Methods | Evaluation Metrics | | | | | |
|---|---|---|---|---|---|---|---|
| | | Retain Acc. (%) | Unlearn Acc. (%) | Test Acc. (%) | MIA Acc. (%) | Avg. Gap | Re-learn Time (# Epoch) |
| - | Original | $99.93_{\pm0.10}(0.03)$ | $0.00_{\pm0.00}(8.33)$ | $95.37_{\pm0.80}(0.57)$ | $4.67_{\pm3.30}(22.33)$ | 7.82 | - |
| | Re-train | $99.96_{\pm0.05}(0.00)$ | $8.33_{\pm3.30}(0.00)$ | $94.80_{\pm0.85}(0.00)$ | $27.00_{\pm5.66}(0.00)$ | 0.00 | $3.33_{\pm0.47}$ |
| Gaussian $\lambda = 0.03$ $v = 1$ # epoch = 1 | RURK ($\tau = 0.00$) | $99.93_{\pm0.10}(0.03)$ | $0.00_{\pm0.00}(8.33)$ | $95.10_{\pm0.99}(0.30)$ | $14.33_{\pm8.01}(12.67)$ | 5.33 | $1.00_{\pm0.00}$ |
| | RURK ($\tau = 0.01$) | $99.93_{\pm0.10}(0.03)$ | $0.00_{\pm0.00}(8.33)$ | $94.63_{\pm1.04}(0.17)$ | $15.67_{\pm8.96}(11.33)$ | 4.97 | $1.00_{\pm0.00}$ |
| | RURK ($\tau = 0.02$) | $99.89_{\pm0.16}(0.07)$ | $2.33_{\pm0.47}(6.00)$ | $94.27_{\pm1.08}(0.53)$ | $22.00_{\pm9.90}(5.00)$ | 2.90 | $1.00_{\pm0.00}$ |
| | **RURK ($\tau = 0.03$)** | $\mathbf{99.52_{\pm0.37}(0.44)}$ | $\mathbf{5.67_{\pm2.36}(2.66)}$ | $\mathbf{93.83_{\pm0.90}(0.97)}$ | $\mathbf{33.33_{\pm12.26}(6.33)}$ | **2.60** | $\mathbf{2.00_{\pm0.00}}$ |
| | RURK ($\tau = 0.04$) | $98.96_{\pm0.37}(1.00)$ | $13.33_{\pm0.47}(5.00)$ | $92.37_{\pm1.23}(2.43)$ | $38.67_{\pm14.61}(11.67)$ | 5.03 | $1.67_{\pm0.47}$ |
| | RURK ($\tau = 0.10$) | $96.89_{\pm0.16}(3.07)$ | $29.67_{\pm4.71}(21.34)$ | $87.80_{\pm2.55}(7.00)$ | $54.00_{\pm15.56}(27.00)$ | 14.60 | $1.67_{\pm0.47}$ |
| Gaussian $\tau = 0.03$ $v = 1$ # epoch = 1 | RURK ($\lambda = 0.00$) | $99.96_{\pm0.05}(0.00)$ | $0.00_{\pm0.00}(8.33)$ | $95.00_{\pm1.13}(0.20)$ | $11.33_{\pm6.60}(15.67)$ | 6.05 | $1.00_{\pm0.00}$ |
| | RURK ($\lambda = 0.01$) | $99.93_{\pm0.10}(0.03)$ | $0.00_{\pm0.00}(8.33)$ | $94.60_{\pm0.99}(0.20)$ | $15.67_{\pm8.96}(11.33)$ | 4.97 | $1.00_{\pm0.00}$ |
| | RURK ($\lambda = 0.02$) | $99.89_{\pm0.16}(0.07)$ | $2.67_{\pm0.94}(5.66)$ | $94.17_{\pm1.08}(0.63)$ | $22.67_{\pm10.37}(4.33)$ | 2.68 | $1.00_{\pm0.00}$ |
| | **RURK ($\lambda = 0.03$)** | $\mathbf{99.52_{\pm0.37}(0.44)}$ | $\mathbf{5.67_{\pm2.36}(2.66)}$ | $\mathbf{93.83_{\pm0.90}(0.97)}$ | $\mathbf{33.33_{\pm12.26}(6.33)}$ | **2.60** | $\mathbf{2.00_{\pm0.00}}$ |
| | RURK ($\lambda = 0.04$) | $98.89_{\pm0.47}(1.07)$ | $13.33_{\pm0.47}(5.00)$ | $92.50_{\pm0.85}(2.30)$ | $39.67_{\pm13.20}(12.67)$ | 5.26 | $1.67_{\pm0.47}$ |
| | RURK ($\lambda = 0.10$) | $85.04_{\pm2.62}(14.92)$ | $76.67_{\pm6.60}(68.34)$ | $76.87_{\pm0.05}(17.93)$ | $87.00_{\pm4.24}(60.00)$ | 40.30 | $2.00_{\pm0.00}$ |
| $\lambda = 0.03$ $\tau = 0.03$ $v = 1$ # epoch = 1 | RURK (**Gaussian**) | $\mathbf{99.52_{\pm0.37}(0.44)}$ | $\mathbf{5.67_{\pm2.36}(2.66)}$ | $\mathbf{93.83_{\pm0.90}(0.97)}$ | $\mathbf{33.33_{\pm12.26}(6.33)}$ | **2.60** | $\mathbf{2.00_{\pm0.00}}$ |
| | RURK (FGSM) | $99.70_{\pm0.10}(0.26)$ | $4.33_{\pm0.47}(4.00)$ | $94.17_{\pm1.08}(0.63)$ | $28.67_{\pm11.79}(1.67)$ | 1.64 | $1.00_{\pm0.00}$ |
| | RURK (PGD 1) | $98.81_{\pm0.10}(1.15)$ | $12.33_{\pm0.47}(4.00)$ | $92.53_{\pm1.32}(2.27)$ | $46.67_{\pm14.61}(19.67)$ | 6.77 | $1.67_{\pm0.47}$ |
| | RURK (PGD 5) | $99.85_{\pm0.05}(0.11)$ | $2.00_{\pm0.00}(6.33)$ | $94.07_{\pm1.23}(0.73)$ | $27.33_{\pm13.67}(0.33)$ | 1.88 | $1.00_{\pm0.00}$ |
| | RURK (PGD 10) | $99.93_{\pm0.10}(0.03)$ | $0.67_{\pm0.47}(7.66)$ | $94.57_{\pm1.08}(0.23)$ | $20.00_{\pm11.31}(7.00)$ | 3.73 | $1.00_{\pm0.00}$ |
| Gaussian $\tau = 0.03$ $\lambda = 0.03$ # epoch = 1 | RURK (**v = 1**) | $\mathbf{99.52_{\pm0.37}(0.44)}$ | $\mathbf{5.67_{\pm2.36}(2.66)}$ | $\mathbf{93.83_{\pm0.90}(0.97)}$ | $\mathbf{33.33_{\pm12.26}(6.33)}$ | **2.60** | $\mathbf{2.00_{\pm0.00}}$ |
| | RURK ($v = 2$) | $99.52_{\pm0.37}(0.44)$ | $4.67_{\pm0.94}(3.66)$ | $93.87_{\pm0.94}(0.93)$ | $33.33_{\pm12.26}(6.33)$ | 2.84 | $1.33_{\pm0.47}$ |
| | RURK ($v = 3$) | $99.52_{\pm0.37}(0.44)$ | $5.00_{\pm1.41}(3.33)$ | $93.87_{\pm0.94}(0.93)$ | $33.33_{\pm12.26}(6.33)$ | 2.76 | $1.33_{\pm0.47}$ |
| | RURK ($v = 4$) | $99.52_{\pm0.37}(0.44)$ | $5.67_{\pm2.36}(2.66)$ | $93.83_{\pm0.90}(0.97)$ | $33.33_{\pm12.26}(6.33)$ | 2.60 | $1.00_{\pm0.00}$ |

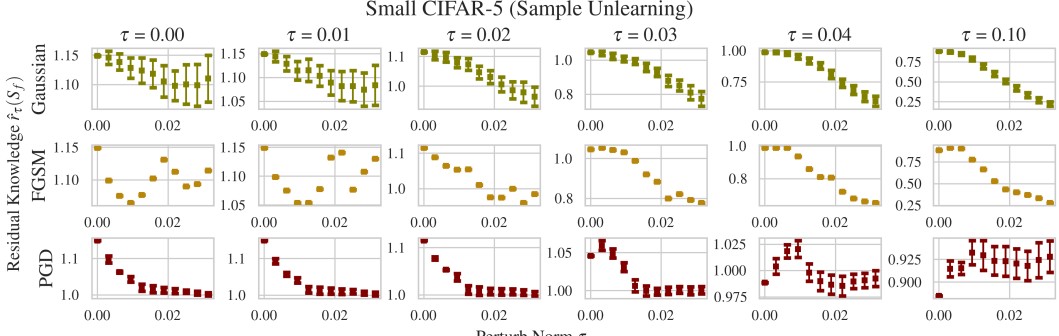

Figure C.12: Residual knowledge $\hat{r}_\tau(\mathcal{S}_f)$ of the proposed `RURK`, `Original`, and other unlearning methods on small CIFAR-5 with sample unlearning, evaluated under varying perturbation norms $\tau$, using untargeted Gaussian noise ($p = 2$), targeted FGSM ($p = \infty$), and targeted PGD ($p = \infty$) to draw $c = 100$ samples from $\mathcal{B}_p(\mathbf{x}, \tau)$.

Table C.5: The performance summary of various unlearning methods on small CIFAR-5 with VGG-11. Results are reported in the format $a_{\pm b}$, indicating the mean $a$ and standard deviation $b$ over 3 independent trials. The absolute performance gap relative to `Re-train` is shown in (blue). For methods that fail to recover the forget-set knowledge within 30 training epochs, the re-learn time is reported as ">30".

| Datasets | Methods | Evaluation Metrics | | | | | |
| --- | --- | --- | --- | --- | --- | --- | --- |
| | | Retain Acc. (%) | Unlearn Acc. (%) | Test Acc. (%) | MIA Acc. (%) | Avg. Gap | Re-learn Time (# Epoch) |
| Small CIFAR-5 | Original | $100.00_{\pm0.00}(2.56)$ | $0.00_{\pm0.00}(15.00)$ | $94.10_{\pm0.00}(2.50)$ | $0.00_{\pm0.00}(16.00)$ | 9.01 | - |
| | Re-train | $97.44_{\pm0.00}(0.00)$ | $15.00_{\pm0.00}(0.00)$ | $91.60_{\pm0.00}(0.00)$ | $16.00_{\pm0.00}(0.00)$ | 0.00 | $4.33_{\pm0.47}$ |
| | NGD | $95.00_{\pm0.00}(2.44)$ | $3.00_{\pm0.00}(12.00)$ | $91.40_{\pm0.00}(0.20)$ | $1.00_{\pm0.00}(15.00)$ | 7.41 | $1.00_{\pm0.00}$ |
| | RURK | $99.33_{\pm0.00}(1.89)$ | $7.00_{\pm0.00}(8.00)$ | $92.70_{\pm0.00}(1.10)$ | $22.00_{\pm0.00}(6.00)$ | 4.25 | $2.00_{\pm0.00}$ |

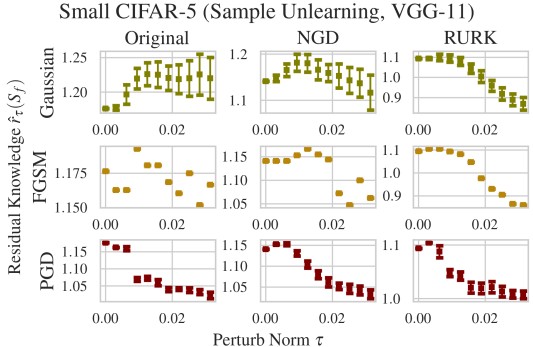

Figure C.13: Residual knowledge $\hat{r}_\tau(\mathcal{S}_f)$ of the proposed `RURK`, `Original`, and `NGD` on small CIFAR-5 with sample unlearning and VGG-11 structure, evaluated under varying perturbation norms $\tau$, using untargeted Gaussian noise ($p = 2$), targeted FGSM ($p = \infty$), and targeted PGD ($p = \infty$) to draw $c = 100$ samples from $\mathcal{B}_p(\mathbf{x}, \tau)$.

## C.4 The prevalence of residual knowledge

In Figure 2, we report the estimated residual knowledge over the entire forget set $\hat{r}_\tau(\mathcal{S}_f)$. We also provide per-sample estimates $\hat{r}_\tau((\mathbf{x}, y))$ for each $(\mathbf{x}, y) \in \mathcal{S}_f$, as defined in Eq. (4). Table C.6 summarizes the proportion of forget samples exhibiting residual knowledge greater than one—i.e., those still recognizable after unlearning—under different perturbation norms $\tau$, evaluated on small CIFAR-5 (with `NTK`) and CIFAR-10 (with `NGD`). Remarkably, even under imperceptibly small perturbations (e.g., $\tau = 0.013$), about 11% of the forget samples in small CIFAR-5 and more than 8% (approximately 160 samples) in CIFAR-10 still exhibit residual knowledge above one. These findings highlight that residual knowledge is not only prevalent but also poses a significant privacy risk, emphasizing the need for stronger certification and mitigation techniques in machine unlearning.

Table C.6: Percentage of forget samples that have residual knowledge $\hat{r}_\tau((\mathbf{x}, y))$ large than 1.

| Datasets | Methods | Gaussian Perturbation Norm $\tau$ | | | | | | | | | | |
| --- | --- | --- | --- | --- | --- | --- | --- | --- | --- | --- | --- | --- |
| | | 0.000 | 0.003 | 0.006 | 0.009 | 0.013 | 0.016 | 0.019 | 0.022 | 0.025 | 0.028 | 0.031 |
| Small CIFAR-5 | NTK | $0.00_{\pm0.00}$ | $3.00_{\pm1.71}$ | $5.00_{\pm2.18}$ | $6.00_{\pm2.37}$ | $11.00_{\pm3.13}$ | $14.00_{\pm3.47}$ | $26.00_{\pm4.39}$ | $34.00_{\pm4.74}$ | $39.00_{\pm4.88}$ | $45.00_{\pm4.97}$ | $48.00_{\pm5.00}$ |
| CIFAR-10 | NGD | $0.00_{\pm0.00}$ | $1.85_{\pm1.35}$ | $5.25_{\pm2.23}$ | $7.25_{\pm2.59}$ | $8.50_{\pm2.79}$ | $8.55_{\pm2.80}$ | $9.30_{\pm2.90}$ | $8.90_{\pm2.85}$ | $8.10_{\pm2.73}$ | $7.10_{\pm2.57}$ | $6.95_{\pm2.54}$ |

## C.5 Disagreement and unlearn accuracy on perturbed forget samples

We provide the adversarial disagreement (§ 4) and the unlearn accuracy on perturbed forget samples for selected baselines on both Small CIFAR-5 and CIFAR-10 (cf. Table 1) in the following four tables.

In Table C.7, RURK achieves the lowest disagreement among all baselines when the perturbation norm is small (e.g., below 0.0125), consistent with the stable residual knowledge curves shown in Figure 2. As the perturbation magnitude increases, however, RURK reduces residual knowledge more aggressively, leading to higher disagreement—a trade-off that arises naturally in multi-class settings, where low disagreement does not necessarily imply residual knowledge close to 1 as in binary cases. Similarly, in Table C.8, RURK attains unlearn accuracy on perturbed forget samples closest to that of the re-trained model, particularly for small perturbations, indicating reduced distinguishability between the two models around the forget region. It is worth noting that RURK is designed to control residual knowledge rather than disagreement directly, as managing disagreement across multiple classes is inherently more complex.

Table C.7: Disagreement of the perturbed forget samples on the Small CIFAR-5 and CIFAR-10 datasets over varying perturbation norms $\tau$.

| Datasets | Methods | Gaussian Perturbation Norm $\tau$ | | | | | | | | | | |
|---|---|---|---|---|---|---|---|---|---|---|---|---|
| | | 0.0000 | 0.0031 | 0.0063 | 0.0094 | 0.0125 | 0.0157 | 0.0188 | 0.0220 | 0.0251 | 0.0282 | 0.0314 |
| Small CIFAR-5 | Original | 0.1300 | 0.1273 | 0.1277 | 0.1332 | 0.1450 | 0.1471 | 0.1450 | 0.1472 | 0.1573 | 0.1665 | 0.1836 |
| | Re-train | 0.0000 | 0.0000 | 0.0000 | 0.0000 | 0.0000 | 0.0000 | 0.0000 | 0.0000 | 0.0000 | 0.0000 | 0.0000 |
| | Fisher | 0.1400 | 0.1345 | 0.1249 | 0.1188 | 0.1129 | 0.1181 | 0.1316 | 0.1532 | 0.1811 | 0.2054 | 0.2214 |
| | NTK | 0.0900 | 0.0893 | 0.0884 | 0.0873 | 0.0805 | 0.0807 | 0.0753 | 0.0797 | 0.0903 | 0.1084 | 0.1285 |
| | RURK | 0.0600 | 0.0435 | 0.0397 | 0.0448 | 0.0506 | 0.0713 | 0.0880 | 0.1247 | 0.1510 | 0.1658 | 0.1765 |
| CIFAR-10 | Original | 0.1020 | 0.1048 | 0.1148 | 0.1297 | 0.1462 | 0.1626 | 0.1755 | 0.1968 | 0.2136 | 0.2431 | 0.2593 |
| | Re-train | 0.0000 | 0.0000 | 0.0000 | 0.0000 | 0.0000 | 0.0000 | 0.0000 | 0.0000 | 0.0000 | 0.0000 | 0.0000 |
| | NGD | 0.1840 | 0.1873 | 0.2209 | 0.3391 | 0.4920 | 0.6069 | 0.6837 | 0.7347 | 0.7699 | 0.7906 | 0.8005 |
| | RURK | 0.1515 | 0.1549 | 0.1701 | 0.1996 | 0.2395 | 0.2898 | 0.3355 | 0.3855 | 0.4353 | 0.4802 | 0.5310 |

Table C.8: Unlearn accuracy of the perturbed forget samples on the Small CIFAR-5 and CIFAR-10 datasets over varying perturbation norms $\tau$.

| Datasets | Methods | Gaussian Perturbation Norm $\tau$ | | | | | | | | | | |
|---|---|---|---|---|---|---|---|---|---|---|---|---|
| | | 0.0000 | 0.0031 | 0.0063 | 0.0094 | 0.0125 | 0.0157 | 0.0188 | 0.0220 | 0.0251 | 0.0282 | 0.0314 |
| Small CIFAR-5 | Original | 0.0000 | 0.0000 | 0.0000 | 0.0004 | 0.0053 | 0.0205 | 0.0442 | 0.0660 | 0.0913 | 0.1161 | 0.1364 |
| | Re-train | 0.1300 | 0.1273 | 0.1277 | 0.1336 | 0.1502 | 0.1665 | 0.1855 | 0.2082 | 0.2417 | 0.2746 | 0.3119 |
| | Fisher | 0.0900 | 0.0938 | 0.1056 | 0.1265 | 0.1566 | 0.2013 | 0.2473 | 0.3078 | 0.3632 | 0.4149 | 0.4675 |
| | NTK | 0.0700 | 0.0652 | 0.0681 | 0.0797 | 0.0986 | 0.1141 | 0.1333 | 0.1462 | 0.1700 | 0.1891 | 0.2120 |
| | RURK | 0.1400 | 0.1521 | 0.1625 | 0.1736 | 0.1879 | 0.2110 | 0.2436 | 0.2846 | 0.3258 | 0.3713 | 0.4124 |
| CIFAR-10 | Original | 0.0000 | 0.0000 | 0.0000 | 0.0010 | 0.0061 | 0.0178 | 0.0351 | 0.0599 | 0.0896 | 0.1253 | 0.1660 |
| | Re-train | 0.1020 | 0.1048 | 0.1148 | 0.1301 | 0.1478 | 0.1666 | 0.1834 | 0.2046 | 0.2245 | 0.2478 | 0.2670 |
| | NGD | 0.1360 | 0.1396 | 0.1754 | 0.3227 | 0.5014 | 0.6437 | 0.7423 | 0.8070 | 0.8536 | 0.8862 | 0.9088 |
| | RURK | 0.1110 | 0.1187 | 0.1424 | 0.1845 | 0.2330 | 0.2930 | 0.3634 | 0.4256 | 0.4935 | 0.5581 | 0.6193 |

