# OpenReview forum: "The Unseen Threat: Residual Knowledge in Machine Unlearning under Perturbed Samples"
_NeurIPS.cc/2025/Conference — NeurIPS 2025 poster_

### Official Review · Reviewer_eKwF · 2025-06-23

**Clarity:** 3
**Significance:** 3
**Originality:** 3
**Rating:** 5
**Confidence:** 4

**Summary:**

Machine unlearning is an approach to remove specific data without re-training the model. While its effectiveness has been proven, this paper theoretically proves that the unlearned model may suffer from perturbed forget samples, raising a new risk of unlearning. With the DP framework, they introduce model distinguishability and sample disagreement, depending on which they formulate the vulnerability as residual knowledge and propose RURK to constrain the model's ability to recognize perturbed forget samples. Extensive results show the effectiveness of their method.

**Questions:**

- I may miss something, but I didn't find any illustration on how to get Fig.1. I think it would be better to give a brief introduction.
- For Proposition 2:
    - First, the assumption that the sample space is a sphere is too strict and not very practical.
    - Second, when $\epsilon=\delta=0$, why does Eq.3 become zero, as the first term $\frac{2\delta}{1-e^{\epsilon}}$ is $\frac{0}{0}$?
- As $r_{\tau}$ is a relative ratio, is it enough to measure the residual knowledge inside the unlearned model $m$? For example, if $\Pr[m(x')=y] = N \Pr[a(x')=y] = N\xi$ where $\xi \to 0^+$ and $N$ is a large constant. Thus, $r_{\tau}=N$ can be large but $\Pr[m(x')=y]$ is small. In that case, can we say the unlearned model has a large amount of residual knowledge since both models are unconfident and unlikely to classify the label correctly?

**Ethical Concerns:**

["NO or VERY MINOR ethics concerns only"]

**Final Justification:**

After multiple rounds of discussion, I acknowledge the contribution and novelty of this paper and find it quite interesting. As privacy risks become increasingly important, unlearning has emerged as a critical technique to address them. However, the effect of unlearning under perturbation remains underexplored. This paper provides a theoretical analysis showing that, even when unlearned and re-trained models are indistinguishable, they may still disagree on adversarial samples. The authors introduce the concept of residual knowledge as a special case to quantify this disagreement and propose the RURK method to mitigate it. I believe this work will draw the community’s attention to the privacy risks of unlearning in the presence of adversarial examples.

Despite its strengths, the paper’s main limitation lies in its narrow scope. It focuses solely on (small) classification models, overlooking emerging risks in generative models such as diffusion models. Extending the analysis to broader architectures and tasks would significantly enhance the impact and generality of the results.

In light of our discussions, I am glad to raise my score to 5 with higher confidence.

**Limitations:**

yes

**Quality:**

3

**Strengths And Weaknesses:**

Strength:

- The paper is well written and easy to follow.
- The problem is well motivated. While most prior work focuses on the forget performance of unlearned samples, that of perturbed forget samples is not well studied, though interesting and important.
- The experiments are comprehensive, including 11 baselines, 5 metrics, and 2 datasets to evaluate their effectiveness in multiple dimensions.

Weakness:
- The size of the models and datasets used in this paper is too small. As the unlearning of generative models is receiving more and more attention and is a critical part in unlearning, adding experiments on diffusion models can enhance this paper's impact, though I don't see a direct or easy application of the proposed method on diffusion models.
- The assumptions of some theories are not that practical or maybe there are some mistakes. See questions.
- The relative ratio $r_{\tau}$ seems insufficient to measure the residual knowledge. See questions
- The introduction of residual knowledge is confusing to me. As Sec.3 has proven that an unlearned model can still correctly predict the perturbed forget samples, it is enough to elicit the RURK method, which unlearns the perturbed forget samples. Why do we need residual knowledge and $r_{\tau}$? Does the RURK loss reduce $r_{\tau}$? Adding more discussion would enhance the understanding of this paper.

---

> ### Author Rebuttal · Authors · 2025-07-31
>
> We thank the reviewer for their time and effort. We are pleased that the reviewer found the paper well-written, well-motivated, and supported by comprehensive experiments. Below, we systematically address each of the weaknesses and questions raised in the review.
>
> For Weakness 1, we understand the reviewer’s concern. Indeed, for a diffusion model (generative tasks), unlearning often focuses on concept unlearning, where the aim is to forget an entire class or concept represented by a label. In these scenarios, the notion of residual knowledge, as defined in our paper, does not directly apply, since the input is not a sample but a label or prompt. However, there are notable exceptions. For instance, in image-to-image (I2I) generation (as shown in Fan et al. (2023, Figure 6) and Li et al. (2024, Figure 1)), the input is a partial image (e.g., an unmasked region) rather than a label. In such cases, residual knowledge may still manifest: perturbations to the input region could lead the model to regenerate forgotten content. However, we have conducted larger-scale experiments on the ImageNet-100 dataset—a subset of ImageNet-1k containing 100 classes and approximately 1,300 images per class (for a total of 130k images)—using ResNet-50. In this setting, we select class 0 as the forget class and perform 50\% random sample unlearning, evaluating the results as summarized in Table 1 below. Due to time constraints, we include a subset of competitive baselines for comparison. As shown in the results, RURK consistently achieves the smallest average gap compared to GD and NegGrad+, confirming its effectiveness even at scale. Additionally, we provide the tabular version of the residual knowledge corresponding to Figure 2. The results reveal that GD still exhibits residual knowledge $> 1$, since it only fine-tunes on retain samples and lacks explicit control over forget samples or residual knowledge. Similarly, SSD, when applied to a large architecture like ResNet-50, struggles to identify and suppress all neurons associated with the forget information, leading to suboptimal mitigation. While NegGrad+ does not suffer from residual knowledge in the same way, its residual knowledge remains farther from 1 compared to RURK, indicating less alignment with the re-trained model. These findings support that the phenomenon of residual knowledge is not limited to small-scale settings, and that RURK remains effective in controlling it, even on larger and more complex datasets. We will incorporate and refine this new empirical evidence in the revised version.
>
>  Metrics   | Retain Acc.      | Unlearn Acc.     | Test Acc.        | Avg. Gap
> ---------- | ---------------- | ---------------- | ---------------- | ----------------
> Original   | 80.75±3.06(5.69) | 3.59±0.19(9.03)  | 72.67±1.18(4.23) | 6.32
> Re-train   | 75.06±1.54(0.00) | 12.62±1.40(0.00) | 68.44±1.51(0.00) | 0.00
> GD         | 82.29±0.37(7.23) | 6.26±2.38(7.13)  | 74.44±1.39(6.01) | 6.79
> NegGrad+   | 81.86±1.86(6.80) | 40.67±1.85(27.28)| 73.53±1.06(5.10) | 13.06
> SSD        | 80.35±3.00(5.30) | 90.15±13.92(76.77)|  71.81±1.05(3.38)| 28.48
> RURK       | 83.09±0.26(8.03) | 11.62±0.69(1.00) | 75.19±0.59(6.75) | 5.26
>
>
> $\tau$     | 0.0000 | 0.0031 | 0.0063 | 0.0094 | 0.0125 | 0.0157 | 0.0188 | 0.0220 | 0.0251 | 0.0282 | 0.0314
> -------- | ------ | ------ | ------ | ------ | ------ | ------ | ------ | ------ | ------ | ------ | ------
> Original | 1.1000 | 1.0996 | 1.1018 | 1.1011 | 1.1077 | 1.1107 | 1.1174 | 1.1288 | 1.1303 | 1.1421 | 1.1553
> GD       | 1.0656 | 1.0624 | 1.0647 | 1.0650 | 1.0606 | 1.0575 | 1.0595 | 1.0610 | 1.0642 | 1.0665 | 1.0654
> NegGrad+ | 0.6915 | 0.6861 | 0.6791 | 0.6723 | 0.6701 | 0.6624 | 0.6588 | 0.6556 | 0.6469 | 0.6351 | 0.6163
> SSD      | 1.1046 | 1.1032 | 1.1067 | 1.1095 | 1.1133 | 1.1161 | 1.1233 | 1.1291 | 1.1367 | 1.1446 | 1.1514
> RURK     | 1.0000 | 0.9977 | 0.9954 | 0.9898 | 0.9846 | 0.9763 | 0.9624 | 0.9339 | 0.9108 | 0.8858 | 0.8633
>
>
> For Weakness 2, please refer to our response to Question 2 below.
>
> For Weaknesses 3, 4 and Question 3, we appreciate the thoughtful comments regarding the residual knowledge $r_\tau$, and will expand on this discussion in the revised manuscript. The reason we introduce residual knowledge $r_\tau$ is to bring in the re-trained model $a$, as the ideal outcome of unlearning is that the unlearned model $m$ and $a$ behave consistently on both the training samples and their perturbed counterparts (see Lines 183–186). The residual knowledge captures the tendency of the unlearned model $m$ to re-recognize slightly perturbed forget samples compared to the re-trained model $a$, revealing a potential privacy risk (see Lines 33–39). Therefore, $r_\tau$ cannot be defined using the unlearned model $m$ alone. As explained in Lines 263–266, RURK is designed to reduce residual knowledge by directly minimizing $\Pr[m(x') = y]$, avoiding reliance on the re-trained model during the unlearning process. The example raised by the reviewer illustrates exactly the phenomenon that $r_\tau$ is intended to capture: even if $\Pr[m(x') = y]$ is small, when divided by a smaller $\Pr[a(x') = y]$, the ratio $r_\tau > 1$, indicating that the unlearned model is still more prone to re-identify the forget sample than the re-trained model. This is the precise privacy risk we aim to define and mitigate in this work.
>
>
>
> For Question 1, Figure 1 provides a pictorial illustration of residual knowledge, which is later supported by the quantitative results in Table 1 and Figure 2. The figure is intended to convey that, after successful unlearning, the unlearned and re-trained models should make consistent decisions on all samples. However, due to slight deviations in the model parameters or decision boundaries (as captured by $(\epsilon, \delta)$-indistinguishability), the models may disagree on perturbed samples near the decision boundary. This discrepancy illustrates the presence of residual knowledge. We will incorporate this clarification into the revised caption of Figure 1.
>
>
> For Question 2, we appreciate the reviewer’s careful attention to detail and will incorporate additional explanation in the revision. (1): The assumption underlying the isoperimetric inequality---that samples lie on the surface of a unit sphere---is indeed a strong one. However, the underlying phenomenon of the isoperimetric inequality (if a subset occupies a small volume, its boundary must be relatively large) can be extended beyond the unit sphere. In fact, this principle generalizes to the entire unit cube (not just its surface), as shown in Proposition 2.8 of [R1]. The unit cube is a more realistic assumption, as images are often scaled such that pixel values lie in $[0, 1]$. That said, Proposition 2 is still meaningful: it illustrates that disagreement on perturbed samples can arise even under the stricter assumption of a unit sphere (which is a smaller space of samples, compared with the unit cube), making the result even more compelling when considering looser assumptions like the unit cube. We will extend Proposition 2 with the unit cube assumption and include it in Line 195 and in the Appendix. (2): The reviewer is also correct that the expression in question is indeterminate unless a specific relationship between $\epsilon$ and $\delta$ is defined. To address this, we can make the relationship explicit by setting $\delta = k\epsilon$ for some constant $k$. This allows us to apply L’Hôpital’s Rule to evaluate the limit: $\lim_{\epsilon \to 0} \frac{\frac{d}{d\epsilon}(2k\epsilon)}{\frac{d}{d\epsilon}(1-\exp(-\epsilon))} =  \lim_{\epsilon \to 0} \frac{2k}{\exp(-\epsilon)} = 0$. We will include this clarification in  Line 206.
>
> Thank you again for your thoughtful feedback. We're happy to provide any additional information or address further questions. If you feel that we’ve sufficiently addressed your main concerns, we kindly ask you to consider updating your score accordingly.
>
>
> [R1] Ledoux, M., 2001. The concentration of measure phenomenon (No. 89). American Mathematical Soc.

---

> > ### Comment · Reviewer_eKwF · 2025-08-01
> >
> > I appreciate the careful rebuttal and additional experiments from the author. However, I still have some questions:
> >
> > - **W4**: I still cannot get the point of why the authors introduced "residual knowledge". If my understanding is correct, the whole logic of this paper is: adversarial samples can be more distinguishable (sec.3) -> train a model on perturbed forget samples (sec.4.2). I am unclear why the authors introduce the residual knowledge in Sec.4.1, as neither the theories (sec.3) nor the loss function (sec.4.2) involves it. To make my statement clearer, the residual knowledge can be used as a metric to evaluate the performance of unlearned models or compare the difference between unlearned and re-trained models, but not as a motivation for the proposed RURK. The observation in Fig.1 and the theories in Sec.3 are sufficient to motivate RURK IMO, and I don't know how to bridge residual knowledge with either Sec.3 or Sec. 4.2.
> > - **Q1**: Sorry for the confusion. My question here is that there is no explanation about the experimental details. I would like to know more about the experimental setting to produce Fig.1, such as data (size), model, etc.

---

> > > ### Author Response · Authors · 2025-08-01
> > >
> > > We appreciate the reviewer’s follow-up questions and the effort to enable early interaction.
> > >
> > > **Point W4**: In Section 3, we show that adversarial examples can be more indistinguishable. However, in Section 3.2, we only demonstrate that disagreement between the re-trained and unlearned models "exists" (e.g., Figure 1), but did not provide a method to quantify how severe it is. Section 4 addresses this by defining a quantitative metric, the residual knowledge—a specific case of disagreement where the unlearned model outputs the correct label while the re-trained model does not.
> > >
> > > We focus on this special case for two reasons: (i) quantifying disagreement in multi-class settings is complex (Lines 224–227), and (ii) this form of disagreement (i.e., the residual knowledge) has clearer privacy implications—it captures the chance that the unlearned model re-recognizes a perturbed forget sample that the re-trained model does not (Lines 242–245). Thus, residual knowledge can be viewed as a special case of this disagreement (please also refer to Lines 252–258), and motivate our design of RURK in Section 4.2. We agree with the reviewer that one could directly use the disagreement to study distinguishability; however, our work emphasizes the privacy risk. We will include this clarification in the revision—thank you for prompting it.
> > >
> > > **Point Q1**: Thank you for the clarification. Figure 1 is based on the UCI Iris dataset (3 classes, 50 samples each, 4 features). For both the original and re-trained models, we use a simple feedforward neural network with one hidden layer of 100 neurons. We select 7 forget samples in total from three class for illustration. While this selection differs from the random sample or class unlearning setup (Lines 297–303), the goal is purely illustrative—to convey the intuition behind the existence of disagreement and residual knowledge. The unlearned model is trained via GD (fine-tuning on retain samples; see Lines 314–315).
> > >
> > > We hope these clarifications address your concerns. As we’ve seen your acknowledgment posted, we still kindly ask the reviewer to consider updating the score if you feel our responses have sufficiently resolved your points. Thank you again!

---

> > > > ### Comment · Reviewer_eKwF · 2025-08-03
> > > >
> > > > Thanks for your response.
> > > >
> > > > **Q1**: Thanks for your explanation. I think it would be better to include it in the appendix.
> > > >
> > > > **W4**: Thanks for your response. It is now clear to justify the introduction of residual knowledge.
> > > >
> > > > **W1**: Thanks for the authors' response. However, I think the critical difference between RN50 and diffusion models is the model architecture. Though the author uses ImageNet as a larger dataset to evaluate their method, it does not mean that their method can be (easily) applied to diffusion models. I understand that diffusion models are not the main focus of this paper, but anyway, this will limit the impact of this paper.
> > > >
> > > > **New point (Q4)**: I noticed that [A] (also cited by the author) shows that existing unlearning methods cannot remove poisoned data. My questions are:
> > > > 1. Can the conclusions/theories in this paper be used to explain the phenomenon in [A]? What is the difference between the **observations** in [A] and this paper? Note that I am not asking about the difference in principles or explanations. Since these two papers are very similar in their observations, a more in-depth discussion about them would enhance this paper.
> > > > 2. If their observations are the same, can I understand that the unique dimension of this paper beyond [A] is that this paper compares the difference between an unlearned and re-trained model, i.e., disagreement (as in line 138-140)?
> > > > 3. I am now thinking this paper is using the established indistinguishability to study the unlearning effect on perturbed samples, or generalizing the indistinguishability to perturbed samples. Please let me know if my understanding is correct.
> > > >
> > > > [A] Pawelczyk, Martin, et al. "Machine unlearning fails to remove data poisoning attacks." arXiv preprint arXiv:2406.17216 (2024).

---

> > > > > ### Author Response · Authors · 2025-08-04
> > > > >
> > > > > We appreciate the reviewer’s follow-up questions and continued interaction!
> > > > >
> > > > > **Points Q1 and W4**: We’re grateful for the constructive discussion, which has helped improve the clarity of our work. We will incorporate these points into the revision.
> > > > >
> > > > > **Point W1**: We agree with the reviewer that ResNet-50 and diffusion models have very different architectures, and we also appreciate the reviewer’s understanding that diffusion models—or generative models more broadly—are beyond the scope of this paper. As mentioned in our earlier response to W1, defining residual knowledge for generative tasks can differ significantly from classification, except in image-to-image generation. Studying the role of residual knowledge in unlearning for generative models, including diffusion models and language models, is one of our prioritized future directions.
> > > > >
> > > > > **New Point Q4**: We thank the reviewer for raising this thoughtful connection to [A]. We first clarify the setting and findings in [A] (see also Section 4.4 of [A]). Following their notation, let $S_{\text{train}}$ be the training set, composed of two disjoint subsets: $S_{\text{clean}}$ and $S_{\text{poison}}$, where $S_{\text{clean}}$ is clean and $S_{\text{poison}}$ is corrupted via a targeted or backdoor data poisoning attack (Sections 4.1–4.3 in [A]). The original model is trained on $S_{\text{train}}$, with $S_{\text{poison}}$ designated as the forget set and $S_{\text{clean}}$ as the retain set.
> > > > >
> > > > > Ideally, an unlearned model (obtained by removing $S_{\text{poison}}$ from the original) should resemble the re-trained model trained only on $S_{\text{clean}}$, and thus be free of poisoning effects (measured via correlation). However, the authors of [A] observe that the unlearned model still exhibits effects of data poisoning—unlike the clean re-trained model. In other words, [A] concludes that a corrupted model cannot be fully sanitized by simply unlearning the poisoned samples.
> > > > >
> > > > > This is fundamentally different from our setting. We do not involve any corrupted data during training. In our case, both the retain and forget samples are clean. Our result shows that even when the re-trained and unlearned models are indistinguishable in distribution, they can still behave differently on perturbed forget samples—exactly as the reviewer noted in point (3). To summarize: [A] trains on corrupted data and unlearns the poisoned subset, while our work trains on clean data and studies indistinguishability under (adversarial) perturbations applied at test time.
> > > > >
> > > > > While our conclusion may not directly explain the results in [A], there may be useful connections. For instance, as mentioned in our Future Directions (Lines 384–387), it would be interesting to study whether training the original model with adversarial robustness techniques would make residual knowledge more or less severe, i.e., harder or easier to remove the residual knowledge.
> > > > >
> > > > > We will include this discussion in the revision as the reviewer suggested. We hope these clarifications address your questions!
> > > > >
> > > > > [A] Pawelczyk, Martin, et al. "Machine unlearning fails to remove data poisoning attacks." arXiv preprint arXiv:2406.17216 (2024).

---

> ### Comment · Reviewer_eKwF · 2025-08-05
>
> Thanks for your response, but I would like to continue the discussion on **Q4**.
> Let me first show my understanding of your paper (which I will use [Y] to represent) and the reference (which I still use [A] to represent). I will then attempt to bridge the two papers based on my understanding.
>
> Let $S_{train}$ be the training set, consisting of $S_{retain}$ and $S_{forget}$. Here $S_{retain}$ is the set to remember, while $S_{forget}$ is the set to forget. Let $S_{poison}$ and $S_{perturb}$ be the poisoned data and perturbed data of $S_{forget}$, respectively.
>
> - My understanding of [Y]: An unlearned model, which tries to forget $S_{forget}$ from a model trained on $S_{retain}$ and $S_{forget}$, can still remember $S_{perturb}$.
>
> - My understanding of [A]: An unlearned model, which tries to forget $S_{poison}$ from a model trained on $S_{retain}$ and $S_{poison}$, can still remember (backdoor trigger of) $S_{poison}$.
>
> Let me make a hypothesis based on my understanding of [Y]: An unlearned model, which tries to forget $S_{perturb}$ from a model trained on $S_{retain}$ and $S_{perturb}$, can still remember $S_{forget}$.
>
> If this hypothesis is true, I think both papers are observing the same phenomenon: There is some essential knowledge that cannot be erased from the original model during unlearning. In [Y], such knowledge is the recognition of an image, because removing/adding noise does not affect the model's classification on (perturbed) forget samples. In [A], such knowledge is the recognition of backdoor patterns, where we can consider the images themselves as the noise/perturbation.
>
> My questions are:
> 1. Is my hypothesis true or not? I understand that proving it theoretically or conducting empirical experiments may be non-trivial, but from my view, it is a reasonable and straightforward hypothesis.
> 2. If my hypothesis is true, is my understanding of the connection between [A] and [Y] correct as well?
> 3. If the answer to the 2nd question is yes, is there any further connection between [A] and [Y]? What is the contribution of [Y] beyond [A]?

---

> > ### Comment · Reviewer_eKwF · 2025-08-05
> >
> > Moreover, this paper studies the residual knowledge of unlearning. I think it would be better to discuss it in your paper:
> >
> > Xuan, Hao, and Xingyu Li. "Verifying Robust Unlearning: Probing Residual Knowledge in Unlearned Models." arXiv preprint arXiv:2504.14798 (2025).

---

> > > ### Author Response · Authors · 2025-08-06
> > >
> > > We appreciate the reviewer’s follow-up comments and will respond to both comments together here. We follow the reviewer’s notations on [Y], [A] and the datasets.
> > >
> > > We would first like to offer a more detailed clarification:
> > >
> > >  - [Y]: The original model $M_o$ is trained on both $S_\text{retain}$ and $S_\text{forget}$. The unlearned model $M_u$ is intended to forget $S_\text{forget}$, making it indistinguishable from a re-trained model $M_r$ that is trained only on $S_\text{retain}$. However, $M_u$ and $M_r$ are distinguishable on $S_\text{perturbed}$, which introduces a new privacy risk.
> > >
> > >  - [A]: The original model $M_o$ is trained on both $S_\text{retain}$ and $S_\text{poison}$. The unlearned model $M_u$ is expected to forget $S_\text{poison}$, but it may still retain information such as the backdoor trigger embedded in $S_\text{poison}$.
> > >
> > > In other words, in [A], the original model $M_o$ learns (i) the forget samples directly, and (ii) implicitly the poison trace. Existing unlearning methods may effectively remove (i), but not (ii).
> > >
> > > By contrast, in [Y], the original model never sees $S_\text{perturbed}$ during training—it only learns from $S_\text{forget}$. Nonetheless, we discover that $M_u$ and $M_r$ remain distinguishable on $S_\text{perturbed}$. That is, unlike [A], we do not need the original model to learn both (i) and (ii) to observe distinguishability.
> > >
> > > Therefore, for **Point 1**, the hypothesis proposed by the reviewer may not be accurate. It should instead be: **An unlearned model, which forgets $S_\text{forget}$ from a model trained on $S_\text{retain}$ and $S_\text{forget}$, can still re-recognize $S_\text{perturbed}$ more often than a re-trained model.** It is not that the model "remembers $S_\text{perturbed}$", since the original model has never encountered $S_\text{perturbed}$ to begin with.
> > >
> > > Regarding **Points 2 and 3**: In [A], the original model explicitly learns the poison pattern from $S_\text{poison}$ (i.e., point (ii) above). In contrast, the model in [Y] never learns $S_\text{perturbed}$ at all. Instead, it is the implicit generalizability from $S_\text{forget}$ that remains in the unlearned model and cannot be fully removed by existing unlearning methods. This makes $M_u$ more likely to recognize $S_\text{perturbed}$ compared to $M_r$.
> > >
> > > To be more precise, some essential knowledge is indeed retained in the unlearned model in both [A] and [Y]. However, the nature of this knowledge differs: in [A], it is the explicitly learned poison pattern from $S_\text{poison}$; in [Y], it is the implicit generalizability learned from $S_\text{forget}$. Hence, [A] is better framed as an attack on unlearning algorithms using poisoned data, whereas [Y] highlights a new privacy risk under clean training data. We hope this clarification resolves any confusion, and we will include this explanation in the revision.
> > >
> > > We also thank the reviewer for pointing out the new reference "Verifying Robust Unlearning: Probing Residual Knowledge in Unlearned Models" ([M], released on arXiv on 2025/04/21). We would like to highlight a key difference between [Y] and [M]:
> > > [M] designs an attack against the unlearned model such that, after the attack, the output (prediction or generation) on forget samples resembles that of the **original model**—see Proposition 3 or Definition 4 in [M].
> > > In contrast, [Y] evaluates the distinguishability between the **unlearned and re-trained** models, and proposes RURK accordingly. We hope this distinction is clear and will definitely include [M] in the revised Related Work section.
> > >
> > > Once again, we thank the reviewer for their thoughtful feedback and engaging discussion, which have helped improve the clarity and depth of our manuscript.

---

> > > > ### Comment · Reviewer_eKwF · 2025-08-07
> > > >
> > > > Thanks for your response. I now acknowledge the contribution and novelty of this paper. I have only one question left:
> > > >
> > > > - Since Figure 1 illustrates examples of disagreement, it would be more informative to report the disagreement ratio or unlearning accuracy on **adversarial** samples w/ and w/o RURK. Table 1 currently only includes unlearning accuracy on unperturbed forget samples. This question aims to answer the significance of the observation and how well RURK performs.

---

> > > > > ### Author Response · Authors · 2025-08-08
> > > > >
> > > > > We appreciate that the reviewer acknowledges the contribution and novelty of this work, and thank them for the suggestion to better connect Figure 1 with the theoretical results in Section 3 and the empirical findings in Section 5.
> > > > >
> > > > > Due to time constraints, we provide the **disagreement** (Lines 221–222) and the **unlearn accuracy** on perturbed forget samples for selected baselines on both Small CIFAR-5 and CIFAR-10 (Table 1) in the following four tables:
> > > > >
> > > > >
> > > > > ### Table R1: Disagreement of the perturbed forget samples on the Small CIFAR-5 Dataset over varying perturbation norms
> > > > > Disa./$\tau$   | 0.0000 | 0.0031 | 0.0063 | 0.0094 | 0.0125 | 0.0157 | 0.0188 | 0.0220 | 0.0251 | 0.0282 | 0.0314
> > > > > --       | --     | --     | --     | --     | --     | --     | --     | --     | --     | --     | --
> > > > > Original | 0.1300 | 0.1273 | 0.1277 | 0.1332 | 0.1450 | 0.1471 | 0.1450 | 0.1472 | 0.1573 | 0.1665 | 0.1836
> > > > > Fisher   | 0.1400 | 0.1345 | 0.1249 | 0.1188 | 0.1129 | 0.1181 | 0.1316 | 0.1532 | 0.1811 | 0.2054 | 0.2214
> > > > > NTK      | 0.0900 | 0.0893 | 0.0884 | 0.0873 | 0.0805 | 0.0807 | 0.0753 | 0.0797 | 0.0903 | 0.1084 | 0.1285
> > > > > RURK     | 0.0600 | 0.0435 | 0.0397 | 0.0448 | 0.0506 | 0.0713 | 0.0880 | 0.1247 | 0.1510 | 0.1658 | 0.1765
> > > > >
> > > > > ### Table R2: Unlearn Acc. of the perturbed forget samples on the Small CIFAR-5 Dataset over varying perturbation norms
> > > > > UA/$\tau$   | 0.0000 | 0.0031 | 0.0063 | 0.0094 | 0.0125 | 0.0157 | 0.0188 | 0.0220 | 0.0251 | 0.0282 | 0.0314
> > > > > --       | --     | --     | --     | --     | --     | --     | --     | --     | --     | --     | --
> > > > > Original | 0.0000 | 0.0000 | 0.0000 | 0.0004 | 0.0053 | 0.0205 | 0.0442 | 0.0660 | 0.0913 | 0.1161 | 0.1364
> > > > > Re-train | 0.1300 | 0.1273 | 0.1277 | 0.1336 | 0.1502 | 0.1665 | 0.1855 | 0.2082 | 0.2417 | 0.2746 | 0.3119
> > > > > Fisher   | 0.0900 | 0.0938 | 0.1056 | 0.1265 | 0.1566 | 0.2013 | 0.2473 | 0.3078 | 0.3632 | 0.4149 | 0.4675
> > > > > NTK      | 0.0700 | 0.0652 | 0.0681 | 0.0797 | 0.0986 | 0.1141 | 0.1333 | 0.1462 | 0.1700 | 0.1891 | 0.2120
> > > > > RURK     | 0.1400 | 0.1521 | 0.1625 | 0.1736 | 0.1879 | 0.2110 | 0.2436 | 0.2846 | 0.3258 | 0.3713 | 0.4124
> > > > >
> > > > > ### Table R3: Disagreement of the perturbed forget samples on the CIFAR-10 Dataset over varying perturbation norms
> > > > > Disa./$\tau$   | 0.0000 | 0.0031 | 0.0063 | 0.0094 | 0.0125 | 0.0157 | 0.0188 | 0.0220 | 0.0251 | 0.0282 | 0.0314
> > > > > --       | --     | --     | --     | --     | --     | --     | --     | --     | --     | --     | --
> > > > > Original | 0.1020 | 0.1048 | 0.1148 | 0.1297 | 0.1462 | 0.1626 | 0.1755 | 0.1968 | 0.2136 | 0.2431 | 0.2593
> > > > > NGD      | 0.1840 | 0.1873 | 0.2209 | 0.3391 | 0.4920 | 0.6069 | 0.6837 | 0.7347 | 0.7699 | 0.7906 | 0.8005
> > > > > RURK     | 0.1515 | 0.1549 | 0.1701 | 0.1996 | 0.2395 | 0.2898 | 0.3355 | 0.3855 | 0.4353 | 0.4802 | 0.5310
> > > > >
> > > > > ### Table R4: Unlearn Acc. of the perturbed forget samples on the CIFAR-10 Dataset over varying perturbation norms
> > > > > UA/$\tau$   | 0.0000 | 0.0031 | 0.0063 | 0.0094 | 0.0125 | 0.0157 | 0.0188 | 0.0220 | 0.0251 | 0.0282 | 0.0314
> > > > > --       | --     | --     | --     | --     | --     | --     | --     | --     | --     | --     | --
> > > > > Original | 0.0000 | 0.0000 | 0.0000 | 0.0010 | 0.0061 | 0.0178 | 0.0351 | 0.0599 | 0.0896 | 0.1253 | 0.1660
> > > > > Re-train | 0.1020 | 0.1048 | 0.1148 | 0.1301 | 0.1478 | 0.1666 | 0.1834 | 0.2046 | 0.2245 | 0.2478 | 0.2670
> > > > > NGD      | 0.1360 | 0.1396 | 0.1754 | 0.3227 | 0.5014 | 0.6437 | 0.7423 | 0.8070 | 0.8536 | 0.8862 | 0.9088
> > > > > RURK     | 0.1110 | 0.1187 | 0.1424 | 0.1845 | 0.2330 | 0.2930 | 0.3634 | 0.4256 | 0.4935 | 0.5581 | 0.6193
> > > > >
> > > > > In Tables R1 and R3, RURK achieves the lowest disagreement among all baselines when the perturbation norm $\tau$ is small (e.g., below 0.0125), which is consistent with the stable residual knowledge curves shown in Figure 2. However, as $\tau$ increases, RURK begins to reduce residual knowledge more aggressively, which leads to higher disagreement. This trade-off arises because we are in a multi-class setting—in contrast to binary classification, where low disagreement directly corresponds to residual knowledge close to 1.
> > > > >
> > > > > In Tables R2 and R4, RURK also attains an Unlearn Accuracy for perturbed forget samples that is closest to that of the re-trained model, especially for small $\tau$. This indicates reduced distinguishability around the forget samples between the unlearned and re-trained models, which directly supports the motivation of our work.
> > > > >
> > > > > It is important to note that RURK is designed to control residual knowledge, not disagreement directly, since managing disagreement in multi-class settings is inherently more complex (Lines 224–227). We will incorporate these new results into the revised Table 1 and Figure 2 along with appropriate explanations.
> > > > >
> > > > > We thank the reviewer again for their insightful suggestion, which has helped us improve the clarity and completeness of this work.

---

> > > > > > ### Comment · Reviewer_eKwF · 2025-08-09
> > > > > >
> > > > > > Thanks for your response!
> > > > > >
> > > > > > After multiple rounds of discussion, I acknowledge the contribution and novelty of this paper and find it quite interesting. As privacy risks become increasingly important, unlearning has emerged as a critical technique to address them. However, the effect of unlearning under perturbation remains underexplored. This paper provides a theoretical analysis showing that, even when unlearned and re-trained models are indistinguishable, they may still disagree on adversarial samples. The authors introduce the concept of residual knowledge as a special case to quantify this disagreement and propose the RURK method to mitigate it. I believe this work will draw the community’s attention to the privacy risks of unlearning in the presence of adversarial examples. Additional experiments show the significance of such a risk and the effectiveness of their method.
> > > > > >
> > > > > > Despite its strengths, the paper’s main limitation lies in its narrow scope. It focuses solely on (small) classification models, overlooking emerging risks in generative models such as diffusion models. Extending the analysis to broader architectures and tasks would significantly enhance the impact and generality of the results.
> > > > > >
> > > > > > In light of our discussions, I am glad to raise my score to 5 with higher confidence. The main reason for the lack of a higher score is the limited scope. I also encourage the authors to consider the following suggestions for the final version:
> > > > > >
> > > > > > - Summarize and incorporate the reviewers' comments into the paper.
> > > > > > - Add a conclusion section and include further discussion on reference [A].
> > > > > >
> > > > > > Again, I believe this is an interesting and important work that will benefit the community, thus voting for a clear acceptance.

---

> > > > > > > ### Author Response · Authors · 2025-08-09
> > > > > > >
> > > > > > > We sincerely appreciate the reviewer’s patience and continuous efforts in helping us improve this manuscript. We will incorporate the reviewer’s comments into the revision, including a further discussion of reference [A] and the addition of a conclusion. Thank you again!

---

### Official Review · Reviewer_2RCn · 2025-06-25

**Clarity:** 3
**Significance:** 3
**Originality:** 3
**Rating:** 5
**Confidence:** 2

**Summary:**

The authors examine what they see as an overlooked threat in the field of machine unlearning: though the samples themselves may appear to be unlearned (as gauged by classification accuracy on these samples), perturbations of the data in $\mathcal{S}_f$ are still classified with high accuracy by the model.  The authors propose a method of unlearning designed to reduce this threat, and they provide a novel notion of residual knowledge.

**Questions:**

- Why is your unlearn accuracy 0.00 for the original method, and why are the error bars +/- 0?  This seems like a fishy estimate of the error bars given that the unlearn set is randomly selected from the data.
- The motivation for the definition of unlearning comes from differential privacy, where the randomness comes from the optimization process.  However, in your definition of residual knowledge, the randomness comes instead from sampling inside of $B_r(p)$ where $p\in \mathcal{S}_f$.  How do you reconcile these two definitions of randomness?  It's not entirely clear to me that this is the "right" definition of residual knowledge.  Can you motivate this a bit better?
- As a follow-up to my previous question, in your definition of residual knowledge, you write that randomness comes from sampling inside of the ball around the forget point.  However, you also mention that you are adversarially perturbing.  How are you sampling from $B_r(p)$?  If you are adversarially perturbing, then this doesn't seem like a very realistic threat.
- I find it a little bit concerning that RURK causes larger adversarial perturbations to do worse against the unlearned model.  Can you explain this behavior?
- Are you able to convert your insight into a membership inference attack against unlearned models?  This seems like a good practical application.
- What is the implication of this research for generative models?  If you don't already have the training data you can't perturb it, so I'm not entirely sure how it helps you extract data or do something harmful if perturbations reveal membership in the training data.

**Ethical Concerns:**

["NO or VERY MINOR ethics concerns only"]

**Final Justification:**

The reviewers satisfied my concerns in their rebuttal.

**Limitations:**

- See the questions about definitions and adversarial perturbations.
-The unlearning accuracy seems way too high for RURK-- you only need to match the test data's accuracy.
- The experiments only look at classifiers on one dataset (counting CIFAR-5 and CIFAR-10 as the same), making them effectively toy examples relative to the scale of modern machine learning.  This is, of course, very difficult to fix during a review period.
- I think that it would be nice to see an experiment with randomly labeled data or a mislabeled subset as the unlearning dataset.  In the current experiments, it's not clear to me whether perturbations around the training data recover the original labels because of residual knowledge or because you've unlearned too heavily exactly at the points in $\mathcal{S}_r$, but once you move away from $\mathcal{S}_r$ the classifier should function as normal.

**Paper Formatting Concerns:**

The formatting looks fine.

**Quality:**

4

**Strengths And Weaknesses:**

Strengths:
- In general, the idea that unlearning doesn't handle perturbations of the samples themselves is really interesting and indeed overlooked.  This was a good insight.
- The authors do a good job defining their terms with mathematical precision, and providing mathematical support for their ideas.  This is well done.
- The paper is clearly written and proper care is taken.

---

> ### Author Rebuttal · Authors · 2025-07-31
>
> Thank you for your time and thoughtful review. We’re glad you recognized the overlooked challenge of handling perturbations in unlearning and appreciated the mathematical formulation. We clarify limitations and address your questions in detail.
>
> For Question 1, the original model, trained on both retain and forget samples, achieves 100\% accuracy on the forget set—this is the forget accuracy. However, the Unlearn Accuracy in Table 1 is defined as 1 - forget accuracy (Footnote 6 Page 8), following the evaluation in Fan et al. (2023, Line 330). We ensure the original model performs well to allow a fair comparison; otherwise, it would be unclear if unlearning occurred. This setup leads to consistent 100\% forget accuracy (i.e., 0\% unlearn accuracy) and thus zero std. Forget samples are randomly selected, and results are averaged over trials with different model initializations.
>
> For Questions 2 and 3, we appreciate the reviewer for highlighting this distinction. Our framework involves two sources of randomness: (1) from the unlearning algorithm (e.g., model initialization), and (2) from perturbations of forget samples. Section 3 focuses on the first to show that disagreement between unlearned and re-trained models on perturbed forget samples is inevitable (Prop. 2). But showing existence is not enough—we also need to quantify this disagreement in practice. Thus, in Section 4, we fix a realization of models $m$ and $a$ (sampling once from the first randomness) and evaluate disagreement under the second, by perturbing forget samples (Lines 221–222, Footnote 4). This motivates our definition of residual knowledge (Lines 229–235) and the design of RURK to reduce it. While it is possible to define residual knowledge using both randomness, doing so would require characterizing the distribution of $m$ or $a$—an intractable task without strong assumptions (Guo et al., 2020; Chourasia and Shah, 2023; Chien et al., 2024)—making it impractical for methods like RURK. To capture the second randomness, we perturb forget samples using various methods (Lines 361–374), including Gaussian sampling from the $\mathcal{B}_p(\tau)$ ball and adversarial methods like PGD, which can be viewed as non-isotropic sampling. We will revise the draft to clarify this distinction.
>
> For Question 4, assumed referring to Fig 2, which evaluates residual knowledge—how consistently a perturbed forget sample is treated by the unlearned model relative to the re-trained model. Ideally, an unlearned model should produce a flat line at $y = 1$ across different $\tau$ (x-axis), indicating full alignment with the re-trained model. For instance, in the bottom-right panel (RURK), we observe residual knowledge close to 1 for all $\tau \leq 0.01$, showing greater robustness than other baselines (Lines 366–372). Together with Table 1, these results highlight RURK’s overall strength: minimal accuracy gaps, while effectively reducing residual knowledge. We will clarify this in the revised discussion of Figure 2 in Section 5.
>
> For Question 5, this connection is interesting. In MIA against an unlearned model, an adversary may exploit side information—such as the minimal perturbation needed for the model to re-recognize a forget sample. A small perturbation norm may suggest prior inclusion in the training set, increasing the chance of successful inference. This presents a compelling direction for future MIA research in unlearning. Emerging work has begun leveraging such side information—e.g., model variation near training points [R1, R2] or perturbation dynamics [R3]—to strengthen attacks. We will expand the discussion in the revision to motivate further exploration.
>
> For Question 6, we thank the reviewer for highlighting the distinction. In classification, unlearning targets individual samples (random or all samples in a class)—enabling direct perturbation to assess residual knowledge. In contrast, generative tasks often focus on concept unlearning, where inputs are labels or prompts, making our definition less directly applicable. However, exceptions exist. In image-to-image generation (e.g., Fan et al. 2023; Li et al. 2024), inputs are partial images, where perturbations may still trigger forgotten content. Language generation adds further complexity due to its compositional nature (Lines 92–93), and forget/retain definitions differ significantly (Line 307), making residual knowledge harder to formalize. Our focus is to introduce and analyze residual knowledge in a controlled classification setting, where evaluation is clearer and more comparable—we will clarify this in Section 2. That said, prior work [R4–R6] shows LLMs can still leak sensitive information post-unlearning, through relearning or jailbreaking—behaviors conceptually related to residual knowledge, which we aim to explore in future work.
>
> For Limitation 1, thanks for raising this point. We did not tune RURK’s unlearn accuracy to match the re-trained model. Instead, we ran unlearning until the RURK loss in Eq. (6) converged, allowing a fair comparison across methods. Under this setup, RURK achieves the second-smallest unlearn accuracy gap relative to the re-train baseline (highlighted in blue in Table 1), closely following NGD. This indicates comparable performance. Importantly, in random sample unlearning, absolute unlearn accuracy is less meaningful than the gap from the re-trained model—consistent with the evaluation protocol in Fan et al. (2023).
>
> For Limitation 2, we conducted larger-scale experiments on the ImageNet-100 dataset (100 classes, ~130k images) using ResNet-50. We unlearn 50\% of class 0 via random sampling and report results in the updated Table 1 below. Due to time constraints, we include a subset of strong baselines. RURK consistently achieves the smallest average gap relative to the re-trained model, outperforming GD and NegGrad+, and confirming its scalability. We also provide a tabular version of the residual knowledge results from Figure 2. GD shows residual knowledge $> 1$, as it only fine-tunes on retain samples without explicitly mitigating forget signals. SSD struggles with large models like ResNet-50, failing to fully suppress neurons encoding forget information. While NegGrad+ avoids the same failure mode, its residual knowledge remains farther from 1 than RURK, indicating weaker alignment with the re-trained model. These findings confirm that residual knowledge persists at scale and that RURK effectively mitigates it. We will incorporate and refine this evidence in the revised manuscript.
>
>  Metrics   | Retain Acc.      | Unlearn Acc.     | Test Acc.        | Avg. Gap
> -- | -- | -- | -- | --
> Original   | 80.75±3.06(5.69) | 3.59±0.19(9.03)  | 72.67±1.18(4.23) | 6.32
> Re-train   | 75.06±1.54(0.00) | 12.62±1.40(0.00) | 68.44±1.51(0.00) | 0.00
> GD         | 82.29±0.37(7.23) | 6.26±2.38(7.13)  | 74.44±1.39(6.01) | 6.79
> NegGrad+   | 81.86±1.86(6.80) | 40.67±1.85(27.28)| 73.53±1.06(5.10) | 13.06
> SSD        | 80.35±3.00(5.30) | 90.15±13.92(76.77)|  71.81±1.05(3.38)| 28.48
> RURK       | 83.09±0.26(8.03) | 11.62±0.69(1.00) | 75.19±0.59(6.75) | 5.26
>
> $\tau$     | 0.0000 | 0.0031 | 0.0063 | 0.0094 | 0.0125 | 0.0157 | 0.0188 | 0.0220 | 0.0251 | 0.0282 | 0.0314
> -- | -- | -- | -- | -- | -- | -- | -- | -- | -- | -- | --
> Original | 1.1000 | 1.0996 | 1.1018 | 1.1011 | 1.1077 | 1.1107 | 1.1174 | 1.1288 | 1.1303 | 1.1421 | 1.1553
> GD       | 1.0656 | 1.0624 | 1.0647 | 1.0650 | 1.0606 | 1.0575 | 1.0595 | 1.0610 | 1.0642 | 1.0665 | 1.0654
> NegGrad+ | 0.6915 | 0.6861 | 0.6791 | 0.6723 | 0.6701 | 0.6624 | 0.6588 | 0.6556 | 0.6469 | 0.6351 | 0.6163
> SSD      | 1.1046 | 1.1032 | 1.1067 | 1.1095 | 1.1133 | 1.1161 | 1.1233 | 1.1291 | 1.1367 | 1.1446 | 1.1514
> RURK     | 1.0000 | 0.9977 | 0.9954 | 0.9898 | 0.9846 | 0.9763 | 0.9624 | 0.9339 | 0.9108 | 0.8858 | 0.8633
>
> For Limitation 3, we will clarify this point in the revision. The approach of obfuscating forget sample labels—via randomization or mislabeling—is effectively captured by the GA method in Table 1. GA pushes predictions away from the original labels, as described in Graves et al. (2021, Page 3), and is already included in our evaluation. Table 1 compares standard unlearning metrics (Lines 117–126, 328–344) to ensure unlearning is not overly aggressive. Even with strong unlearning, a well-behaved model should not re-identify small perturbations of forget samples—unlike a re-trained model, which naturally fails to recognize them. A properly unlearned model should produce uncertain outputs, leading to low residual knowledge—precisely what our metric captures.
>
> Thank you again! We’re happy to provide any additional information. If you feel that we’ve adequately addressed your main concerns, we would greatly appreciate your consideration in updating the score accordingly.
>
> [R1] Jalalzai, H., Kadoche, E., Leluc, R. and Plassier, V., 2022. Membership inference attacks via adversarial examples. arXiv:2207.13572.
>
> [R2] Del Grosso, G., Jalalzai, H., Pichler, G., Palamidessi, C. and Piantanida, P., 2022. Leveraging adversarial examples to quantify membership information leakage. In Proceedings of CVPR.
>
> [R3] Xue, J., Sun, Z., Ye, H., Luo, L., Chang, X., Tsang, I. and Dai, G., 2025. Privacy Leaks by Adversaries: Adversarial Iterations for Membership Inference Attack. arXiv:2506.02711.
>
> [R4] Hu, S., Fu, Y., Wu, S. and Smith, V., 2024. Jogging the memory of unlearned models through targeted relearning attacks. In ICML 2024 Workshop on Foundation Models in the Wild.
>
> [R5] Shumailov, I., Hayes, J., Triantafillou, E., Ortiz-Jimenez, G., Papernot, N., Jagielski, M., Yona, I., Howard, H. and Bagdasaryan, E., 2024. Ununlearning: Unlearning is not sufficient for content regulation in advanced generative ai. arXiv:2407.00106.
>
> [R6] Liu, S., Yao, Y., Jia, J., Casper, S., Baracaldo, N., Hase, P., Yao, Y., Liu, C.Y., Xu, X., Li, H. and Varshney, K.R., 2025. Rethinking machine unlearning for large language models. Nature Machine Intelligence.

---

### Official Review · Reviewer_sUpr · 2025-07-02

**Clarity:** 3
**Significance:** 2
**Originality:** 3
**Rating:** 4
**Confidence:** 4

**Summary:**

This paper investigates a new privacy risk in machine unlearning, termed residual knowledge, where an unlearned model retains latent traces of forget samples, observable through small adversarial perturbations, despite formal statistical indistinguishability from a re-trained model. The authors provide theoretical arguments, including isoperimetric inequality-based proofs, suggesting that disagreement between unlearned and re-trained models under perturbations is inevitable in high-dimensional settings. They introduce a metric to quantify residual knowledge and propose RURK, a fine-tuning strategy that penalizes correct classification of perturbed forget samples, to mitigate this risk. The paper presents empirical results on CIFAR-5 and CIFAR-10, evaluating RURK against several baseline unlearning methods.

**Questions:**

- Can the authors provide a more detailed justification for the use of isoperimetric inequalities and high-dimensional geometric reasoning, particularly regarding how these assumptions map to practical deep learning models and real-world data distributions?

- How do the authors envision applying RURK in deployment settings where the forget set is no longer accessible? Can they propose a realistic variant that does not require the forget set during mitigation?

- Can the authors provide concrete runtime and resource overhead measurements for RURK, including comparisons to existing methods, especially for larger models or datasets?

- Would the authors consider extending the experimental evaluation beyond CIFAR to larger-scale or non-vision tasks to validate both the existence and mitigation of residual knowledge in more realistic settings?

- Is there a reliable, model-agnostic way to estimate residual knowledge in practice without requiring access to a re-trained model? The paper's current proxy approach lacks rigorous evaluation in this regard.

**Ethical Concerns:**

["NO or VERY MINOR ethics concerns only"]

**Final Justification:**

My confidence of the paper is strengthened by the addition emperical results provided by the authors.

**Limitations:**

Partially. While the authors acknowledge theoretical and computational limitations, they understate critical practical concerns, including reliance on forget set access, scalability of RURK, and the narrow experimental scope.

**Quality:**

2

**Strengths And Weaknesses:**

# Strengths:

- The paper formalizes residual knowledge as a distinct privacy concern in unlearning, providing a new perspective on the limitations of current guarantees.

- The theoretical treatment leverages established tools from geometric probability to support the inevitability of local disagreement under adversarial perturbations.

- The proposed RURK method is simple in formulation and integrates with existing gradient-based unlearning pipelines.

- Extensive experiments are conducted on two unlearning scenarios (CIFAR-5, CIFAR-10) with 11 competitive baseline methods, covering multiple evaluation dimensions including accuracy, MIA, and re-learn time.



# Weaknesses:

- The theoretical results rely heavily on the isoperimetric inequality on unit spheres, assuming normalized data in high-dimensional Euclidean space. However, in practical deep learning settings, especially with complex data distributions and learned feature spaces, the applicability of this geometric abstraction is questionable. The authors do not justify how these assumptions transfer to real neural network representations.

- The formal inevitability of disagreement is interesting theoretically but lacks quantitative grounding in realistic settings. The paper provides no evidence on the actual magnitude of this residual knowledge in large, real-world models beyond small CIFAR benchmarks, making the practical severity unclear.

- The empirical evaluation is narrow, focusing only on vision tasks with modest-scale datasets (CIFAR-5 and CIFAR-10) and small-to-mid-size models (ResNet-18, VGG). There is no experimental evidence on modern large-scale architectures or other modalities like NLP or diffusion models, where both unlearning and adversarial perturbations exhibit different behaviors.

- The proposed RURK method requires direct access to the forget set during fine-tuning to generate adversarial perturbations for mitigation. This undermines practical applicability, especially in privacy-sensitive scenarios where the forget set may be permanently erased after an unlearning request. The paper does not address this constraint meaningfully.

- The notion of residual knowledge is formalized based on comparing outputs of unlearned and re-trained models on perturbed forget samples. However, in practice, access to a fully re-trained model is typically unavailable after deployment. The authors mention a practical proxy but fail to rigorously evaluate how well this proxy reflects true residual knowledge in realistic scenarios.

- The computational cost of RURK, especially regarding the inner loop required to generate vulnerable perturbations during fine-tuning, is not quantified. Given that adversarial training is already known to be computationally expensive, the feasibility of RURK for large models or datasets remains unclear.

- The analysis of baseline methods in Table 1 is surface level. While raw performance numbers are reported, the authors miss an opportunity to investigate why specific baselines (e.g., Fisher, NTK, GA) fail in terms of residual knowledge, leaving the reader with limited insights into the underlying mechanisms.

- Some claims about the generality of the findings (e.g., in line 13, that residual knowledge is "prevalent across existing unlearning methods") are overstated given the narrow experimental scope and lack of evidence beyond CIFAR-5 and CIFAR-10.

---

> ### Author Rebuttal · Authors · 2025-07-31
>
> We thank the reviewer for their time and constructive feedback. We address the weaknesses and questions point-by-point below.
>
> For Weakness 1 and Question 1, we appreciate the reviewer’s observation and will further clarify in the revision. While assuming samples lie on the unit sphere is strong, the core idea of the isoperimetric inequality—that a small-volume set must have a large boundary—extends to more realistic settings. Specifically, Prop 2.8 in [R1] generalizes this principle to the unit cube, which better reflects image data where pixel values are normalized in $[0, 1]$. Nonetheless, Prop. 2 remains meaningful: it shows disagreement on perturbed samples even under the stricter unit sphere assumption, making the result compelling under broader assumptions like the unit cube. We will extend Propo 2 accordingly in the revision.
>
> For Weakness 2, Weakness 8, and Question 4, we conducted larger-scale experiments on the ImageNet-100 dataset (100 classes, ~130k images) using ResNet-50. We unlearn 50\% of class 0 via random sampling and report results in the updated Table 1 below. Due to time constraints, we include a subset of strong baselines. RURK consistently achieves the smallest average gap relative to the re-trained model, outperforming GD and NegGrad+, and confirming its scalability. We also provide a tabular version of the residual knowledge results from Figure 2. GD shows residual knowledge $> 1$, as it only fine-tunes on retain samples without explicitly mitigating forget signals. SSD struggles with large models like ResNet-50, failing to fully suppress neurons encoding forget information. While NegGrad+ avoids the same failure mode, its residual knowledge remains farther from 1 than RURK, indicating weaker alignment with the re-trained model. These findings confirm that residual knowledge persists at scale and that RURK effectively mitigates it. We will incorporate and refine this evidence in the revised manuscript.
>
>  Metrics   | Retain Acc.      | Unlearn Acc.     | Test Acc.        | Avg. Gap
> -- | -- | -- | -- | --
> Original   | 80.75±3.06(5.69) | 3.59±0.19(9.03)  | 72.67±1.18(4.23) | 6.32
> Re-train   | 75.06±1.54(0.00) | 12.62±1.40(0.00) | 68.44±1.51(0.00) | 0.00
> GD         | 82.29±0.37(7.23) | 6.26±2.38(7.13)  | 74.44±1.39(6.01) | 6.79
> NegGrad+   | 81.86±1.86(6.80) | 40.67±1.85(27.28)| 73.53±1.06(5.10) | 13.06
> SSD        | 80.35±3.00(5.30) | 90.15±13.92(76.77)|  71.81±1.05(3.38)| 28.48
> RURK       | 83.09±0.26(8.03) | 11.62±0.69(1.00) | 75.19±0.59(6.75) | 5.26
>
> $\tau$     | 0.0000 | 0.0031 | 0.0063 | 0.0094 | 0.0125 | 0.0157 | 0.0188 | 0.0220 | 0.0251 | 0.0282 | 0.0314
> -- | -- | -- | -- | -- | -- | -- | -- | -- | -- | -- | --
> Original | 1.1000 | 1.0996 | 1.1018 | 1.1011 | 1.1077 | 1.1107 | 1.1174 | 1.1288 | 1.1303 | 1.1421 | 1.1553
> GD       | 1.0656 | 1.0624 | 1.0647 | 1.0650 | 1.0606 | 1.0575 | 1.0595 | 1.0610 | 1.0642 | 1.0665 | 1.0654
> NegGrad+ | 0.6915 | 0.6861 | 0.6791 | 0.6723 | 0.6701 | 0.6624 | 0.6588 | 0.6556 | 0.6469 | 0.6351 | 0.6163
> SSD      | 1.1046 | 1.1032 | 1.1067 | 1.1095 | 1.1133 | 1.1161 | 1.1233 | 1.1291 | 1.1367 | 1.1446 | 1.1514
> RURK     | 1.0000 | 0.9977 | 0.9954 | 0.9898 | 0.9846 | 0.9763 | 0.9624 | 0.9339 | 0.9108 | 0.8858 | 0.8633
>
>
> For Weakness 3, please refer to the response on large-scale classification experiments. We also thank the reviewer for drawing attention to generative tasks. In such settings—e.g., image or text generation—unlearning typically targets entire concepts or classes via labels or prompts, making our definition of residual knowledge less directly applicable. However, exceptions exist. In image-to-image (I2I) generation (e.g., Fan et al., 2023; Li et al., 2024), inputs are partial images, and perturbing them can still trigger forgotten content. Language generation is even more complex due to its compositional nature (Lines 92–93) and differing forget/retain semantics (Line 307), complicating the definition and evaluation of residual knowledge. Our goal in this paper is to introduce and study residual knowledge as a privacy risk in classification, where evaluation is clearer and more controlled. We will clarify this scope in Section 2. That said, prior work [R2–R4] shows that LLMs can still leak forgotten content through context relearning or jailbreaking—phenomena conceptually related to residual knowledge, which we aim to investigate in future work.
>
> For Weakness 4 and Question 2, we appreciate the reviewer for raising this point. In classification unlearning, access to forget samples is standard practice in recent work—including the baselines we use (Lines 307–327, Appendix B.2), such as Fan et al. (2023), Kurmanji et al. (2024), and Graves et al. (2021). This assumption is also practical: if a user requests deletion, they must either provide the sample or ask the provider to retrieve it. Unlike generative unlearning (which targets entire concepts), classification unlearning becomes infeasible if neither forget nor retain samples are accessible. Moreover, both our residual knowledge definition (Eq. (4)) and RURK objective (Eq. (6)) rely on forget samples. Fine-tuning on retain samples alone cannot ensure residual knowledge is removed. Finally, RURK is an integrated unlearning process—not a post hoc correction—explicitly designed to eliminate residual knowledge at the time of unlearning. We will clarify these points in revised Sections 2 and 4.
>
> For Weakness 5 and Question 5, we appreciate the reviewer’s concern and offer the following clarification. The notions of $(\epsilon, \delta)$-unlearning and residual knowledge (Lines 108–116) are theoretical constructs that require the unlearned model to closely match a re-trained model—either in weight distribution or output behavior on perturbed forget samples. RURK, however, avoids dependence on the re-trained model during unlearning. Rather than forcing residual knowledge to be exactly 1, RURK constrains it to be ≤1, since values above 1 signal a stronger privacy risk. This design choice enables practical unlearning without retraining, as explained in Lines 261–266 and supported by Figure 2. We acknowledge this trade-off in Section 6. Computing the proxy term (ii) in Eq. (6) is analogous to using empirical adversarial examples to estimate adversarial loss. We will highlight this rationale and our design decisions more explicitly in the revised Section 4.
>
> For Weakness 6 and Question 3, we thank the reviewer for pointing out the lack of clarity. In the main text, we set $v=1$ (i.e., one sample from the vulnerable set using Gaussian noise) in Term (ii) of Eq. (6). RURK’s overhead grows linearly with $v$, but since sampling is done in constant time, its overall complexity remains comparable to other gradient-based methods like GD, NGD, and NegGrad+ (Lines 347–349). Appendix C.3 provides an ablation on $v$. For the larger-scale experiment (on an NVIDIA 8xA10g node) shown above, RURK completes in 42.95 minutes, compared to 1920.48 minutes for full retraining. We will add a more detailed runtime comparison in the revision.
>
> For Weakness 7, we thank the reviewer for prompting a more comprehensive discussion. In addition to Lines 345–374, we offer the following clarifications. Figure 2 uses different y-axis scales across subfigures. NTK exhibits residual knowledge because it linearizes the model via the neural tangent kernel and ignores higher-order output terms. Fisher aggressively adds Gaussian noise to model weights to erase forget information but harms utility (test accuracy). Fine-tuning-only methods like GD and CF-k show high residual knowledge, underscoring the need to access forget samples—as RURK does. NGD, though also fine-tuning only on retain samples, adds controlled weight noise (similar to Fisher), reducing residual knowledge more effectively than GD and CF-k, but still less than RURK. GA fine-tunes on forget samples using reversed gradients, over-erasing forget information and deviating from the re-trained model. EU-k updates only the last layers on forget samples, leaving residuals in earlier representations. SSD selectively dampens weights and behaves similarly to GA due to over-removal. SCRUB performs comparably to RURK but is more expensive due to its reliance on a teacher model. NegGrad+ has a similar objective to RURK but retains more residual knowledge, validating the effectiveness of Term (ii) in Eq. (6). We will incorporate these points in the revision.
>
> Thanks again! We are happy to address any additional questions. If you think that we’ve adequately addressed the limitations, we would greatly appreciate your consideration in updating the score accordingly.
>
> [R1] Ledoux, M., 2001. The concentration of measure phenomenon (No. 89).
> American Mathematical Soc.
>
> [R2] Hu, S., Fu, Y., Wu, S. and Smith, V., 2024. Jogging the memory of unlearned models through targeted relearning attacks. In ICML 2024 Workshop on Foundation Models in the Wild.
>
> [R3] Shumailov, I., Hayes, J., Triantafillou, E., Ortiz-Jimenez, G., Papernot, N., Jagielski, M., Yona, I., Howard, H. and Bagdasaryan, E., 2024. Ununlearning: Unlearning is not sufficient for content regulation in advanced generative ai. arXiv preprint arXiv:2407.00106.
>
> [R4] Liu, S., Yao, Y., Jia, J., Casper, S., Baracaldo, N., Hase, P., Yao, Y., Liu, C.Y., Xu, X., Li, H. and Varshney, K.R., 2025. Rethinking machine unlearning for large language models. Nature Machine Intelligence, pp.1-14.

---

> ### Comment · Reviewer_sUpr · 2025-08-08
>
> Thank you for your detailed rebuttal and additional experiments which address some concerns from my review. The results strengthen the case for RURK, but the issues with forget set dependency, computational cost, and non-vision applicability are there. I will review other reviewers feedback and if I am satisfied, I will update my recommendation accordingly. Please include new results and address remaining gaps in the revision.

---

> > ### Author Response · Authors · 2025-08-08
> >
> > We appreciate the reviewer’s follow-up feedback and are glad to hear that the new empirical results on ImageNet help strengthen the case for RURK. We will ensure that these new results, the accompanying discussion and clarifications during the rebuttal, and the remaining gaps are included and addressed in the revision. If you find our responses to other reviewer's feedback satisfactory, we kindly invite you to consider updating your score accordingly. Thank you again!

---

### Official Review · Reviewer_LG56 · 2025-07-03

**Clarity:** 3
**Significance:** 2
**Originality:** 3
**Rating:** 4
**Confidence:** 4

**Summary:**

This manuscript investigates a novel privacy risk in machine unlearning, termed residual knowledge, where slightly perturbed forget samples may still be recognized by unlearned models. The authors provide both theoretical analysis and empirical evidence to demonstrate that this phenomenon arises inevitablely in high-dimensional settings. To mitigate this issue, the authors propose RURK, a fine-tuning-based method that penalizes the model’s ability to re-recognize perturbed forget samples. The authors conduct experiments on small-scale datasets to demonstrate the effectiveness of the proposed RURK

**Questions:**

- Could the authors provide more results on larger-scale datasets
- Will the threat of residual knowledge also arise in large language models? I would appreciate it if the authors could include some empirical results on unlearning in LLMs.
- Is the adversarial attack used during unlearning (i.e., Eq. 6) the same as the one used to evaluate residual knowledge threat? If so, could the authors provide more results under different types of adversarial attacks?

**Ethical Concerns:**

["NO or VERY MINOR ethics concerns only"]

**Final Justification:**

My major concerns have been addressed after the rebuttal. I maintain my initial positive score.

**Limitations:**

Yes

**Quality:**

3

**Strengths And Weaknesses:**

**Strengths**
- This manuscript is well-written with clear logic and easy to follow.
- The authors identify an interesting phenomenon (residual knowledge) in machine unlearning: slight perturbations of forget samples may still be correctly recognized by the unlearned model. This reveals a novel privacy risk for machine unlearning.
- Both empirical evidence and theoretical insights are provided to support the analysis of model disagreement under sample perturbations.
- The proposed method RURK, which incorporates adversarial perturbations into the unlearning process, is simple but reasonable to solve the identified issue.

**Weaknesses**
- The experimental evaluation is limited to small-scale datasets (small CIFAR-5 and CIFAR10) and backbones (ResNet18).
- In experimental tables (e.g., Table 1), it would be helpful to indicate explicitly whether an increasing or decreasing value represents better performance.
- The proposed RURK method does not appear to have significant improvements over certain existing baselines such as SSD, even in terms of mitigating the residual knowledge threat (e.g., as shown in Figure 2).
- This manuscript lacks a conclusion section.

---

> ### Author Rebuttal · Authors · 2025-07-31
>
> We thank the reviewer for their time and thoughtful feedback. We’re glad to hear that the reviewer found the paper well-written and shares our interest in the newly identified privacy risk of residual knowledge. Below, we address each of the reviewer’s comments and suggestions in detail.
>
> For Weakness 1, we understand the reviewer’s concern and have conducted larger-scale experiments on the ImageNet-100 dataset—a subset of ImageNet-1k containing 100 classes and approximately 1,300 images per class (for a total of 130k images)—using ResNet-50. In this setting, we select class 0 as the forget class and perform 50\% random sample unlearning, evaluating the results as summarized in Table 1 below. Due to time constraints, we include a subset of competitive baselines for comparison. As shown in the results, RURK consistently achieves the smallest average gap compared to GD and NegGrad+, confirming its effectiveness even at scale. Additionally, we provide the tabular version of the residual knowledge corresponding to Figure 2. The results reveal that GD still exhibits residual knowledge $> 1$, since it only fine-tunes on retain samples and lacks explicit control over forget samples or residual knowledge. Similarly, SSD, when applied to a large architecture like ResNet-50, struggles to identify and suppress all neurons associated with the forget information, leading to suboptimal mitigation. While NegGrad+ does not suffer from residual knowledge in the same way, its residual knowledge remains farther from 1 compared to RURK, indicating less alignment with the re-trained model. These findings support that the phenomenon of residual knowledge is not limited to small-scale settings, and that RURK remains effective in controlling it, even on larger and more complex datasets. We will incorporate and refine this new empirical evidence in the revised version.
>
>  Metrics   | Retain Acc.      | Unlearn Acc.     | Test Acc.        | Avg. Gap
> ---------- | ---------------- | ---------------- | ---------------- | ----------------
> Original   | 80.75±3.06(5.69) | 3.59±0.19(9.03)  | 72.67±1.18(4.23) | 6.32
> Re-train   | 75.06±1.54(0.00) | 12.62±1.40(0.00) | 68.44±1.51(0.00) | 0.00
> GD         | 82.29±0.37(7.23) | 6.26±2.38(7.13)  | 74.44±1.39(6.01) | 6.79
> NegGrad+   | 81.86±1.86(6.80) | 40.67±1.85(27.28)| 73.53±1.06(5.10) | 13.06
> SSD        | 80.35±3.00(5.30) | 90.15±13.92(76.77)|  71.81±1.05(3.38)| 28.48
> RURK       | 83.09±0.26(8.03) | 11.62±0.69(1.00) | 75.19±0.59(6.75) | 5.26
>
>
> $\tau$     | 0.0000 | 0.0031 | 0.0063 | 0.0094 | 0.0125 | 0.0157 | 0.0188 | 0.0220 | 0.0251 | 0.0282 | 0.0314
> -------- | ------ | ------ | ------ | ------ | ------ | ------ | ------ | ------ | ------ | ------ | ------
> Original | 1.1000 | 1.0996 | 1.1018 | 1.1011 | 1.1077 | 1.1107 | 1.1174 | 1.1288 | 1.1303 | 1.1421 | 1.1553
> GD       | 1.0656 | 1.0624 | 1.0647 | 1.0650 | 1.0606 | 1.0575 | 1.0595 | 1.0610 | 1.0642 | 1.0665 | 1.0654
> NegGrad+ | 0.6915 | 0.6861 | 0.6791 | 0.6723 | 0.6701 | 0.6624 | 0.6588 | 0.6556 | 0.6469 | 0.6351 | 0.6163
> SSD      | 1.1046 | 1.1032 | 1.1067 | 1.1095 | 1.1133 | 1.1161 | 1.1233 | 1.1291 | 1.1367 | 1.1446 | 1.1514
> RURK     | 1.0000 | 0.9977 | 0.9954 | 0.9898 | 0.9846 | 0.9763 | 0.9624 | 0.9339 | 0.9108 | 0.8858 | 0.8633
>
>
>
> For Weakness 2, we appreciate the reviewer’s insight and will clarify this point in the revised version. In Table 1, because we focus on the random-sample unlearning setting, the increasing or decreasing values of the metrics do not inherently indicate better performance. Rather, the unlearning performance is considered better when the metrics are closer to those of the re-train baseline, which serves as the gold standard. This is why we report the absolute gaps (in blue) between each method and the re-train side-by-side; the smaller the gap, the better the performance. In class unlearning, (results provided in Appendix Table C.2 and referenced in  Line 306) the criteria is the same, as the re-trained model is still the gold baseline to compare with.
>
> For Weakness 3, we appreciate the reviewer’s feedback in helping us clarify this point. We would like to emphasize that the performance of the methods is best understood by jointly considering Table 1 and Figure 2. This is because we are concerned not only with standard metrics such as retain and unlearn accuracy (reported in Table 1), but also with residual knowledge (illustrated in Figure 2). As discussed in the main text (Line 354), SSD tends to aggressively erase forget-set information, resulting in high unlearn accuracy but at the cost of test accuracy dropping below 90\%. Furthermore, although SSD and RURK exhibit similar trends in Figure 2, RURK is more resistant to residual knowledge, as indicated by its longer plateau. Additionally, RURK offers a tunable mechanism for controlling residual knowledge via the parameter $\tau$ of the ball $\mathcal{B}_p(x, \tau)$ in Eq. (6). We included an ablation study of the hyper-parameters of RURK in Appendix C.3, Table C.4 and Figure C.11---parameter $\tau$ effectively reflects on the increase of unlearning accuracy and the plateau of the residual knowledge curves.
>
> For Weakness 4, due to space constraints, we chose to omit a summary of the work and instead prioritized discussions on limitations, future directions, and broader impact in Section 6 (Final Remark). However, we are happy to include a conventional concluding paragraph in the revised version.
>
> For Question 1, please refer to our response in Weakness 1.
>
> For Question 2, we appreciate the reviewer’s suggestion. Indeed, unlearning has became an active area of research in LLMs (Lines 92-93), and it presents significantly greater complexity due to the inherent nature of language. In particular, the unlearning process---including the definitions of forget/retain samples and the evaluation metrics---differs considerably between LLMs and classification settings (e.g., the baseline methods discussed in Line 307). In this paper, our primary goal is to highlight a new privacy risk: residual knowledge. To do so, we focus on classification settings, which allow for clearer and more controlled evaluation. We will make this focus more explicit in the revised version. That said, several existing works [R1–R3] have demonstrated that sensitive information can still be reverse-engineered from LLMs even after unlearning—either through relearning (using different text corpora with similar contexts) or via jailbreaking techniques. While these observations are not directly equivalent, they are conceptually related to the residual knowledge phenomenon we investigate in the classification setting. We are excited to further explore this connection in future work.
>
> For Question 3, yes---Figure 2 presents results where RURK is trained using a Gaussian noise attack, and the residual knowledge is also evaluated with respect to the same Gaussian noise, ensuring a fair comparison. We also include results for other adversarial attacks, such as FGSM and PGD, where both RURK and the residual knowledge evaluation are based on the same respective attack. These additional results are provided in Appendix Figure C.11 and referenced in Lines 373–374.
>
> Thanks again for the comment! We are happy to provide more information or answer any follow-up questions. Please consider raising our score if you think we have addressed your main concerns.
>
> [R1] Hu, S., Fu, Y., Wu, S. and Smith, V., 2024. Jogging the memory of unlearned models through targeted relearning attacks. In ICML 2024 Workshop on Foundation Models in the Wild.
>
> [R2] Shumailov, I., Hayes, J., Triantafillou, E., Ortiz-Jimenez, G., Papernot, N., Jagielski, M., Yona, I., Howard, H. and Bagdasaryan, E., 2024. Ununlearning: Unlearning is not sufficient for content regulation in advanced generative ai. arXiv preprint arXiv:2407.00106.
>
> [R3] Liu, S., Yao, Y., Jia, J., Casper, S., Baracaldo, N., Hase, P., Yao, Y., Liu, C.Y., Xu, X., Li, H. and Varshney, K.R., 2025. Rethinking machine unlearning for large language models. Nature Machine Intelligence, pp.1-14.

---

> > ### Comment · Reviewer_LG56 · 2025-08-05
> >
> > Thank you for the authors’ rebuttal. My major concerns have been addressed, and I will maintain my initial positive score.

---

### Decision · Program_Chairs · 2025-09-17

**Decision:**

Accept (poster)

**Comment:**

This paper points out an important gap in machine unlearning, even when models “forget” samples, small perturbations can still make them recognizable, which the authors call residual knowledge. They back this up with both theory and experiments, and propose RURK to reduce the issue. Reviewers liked the clarity, novelty, and the combination of proofs + experiments, and the rebuttal added useful larger-scale results (ImageNet-100, ResNet-50) and runtime clarifications. The main weaknesses are that experiments are still mostly small-scale classification tasks, and the method assumes access to forget samples, which may not always be realistic. Still, overall, the contribution is solid and raises awareness of a real privacy risk, so the consensus leans toward acceptance.